# Auditing Privacy Mechanisms via Label Inference Attacks

**Róbert István Busa-Fekete**
Google Research NY
busarobi@google.com

**Travis Dick**
Google Research NY
tdick@google.com

**Claudio Gentile**
Google Research NY
cgentile@google.com

**Andrés Muñoz Medina**
Google Research NY
ammedina@google.com

**Adam Smith**
Boston University
& Google DeepMind
ads22@bu.edu

**Marika Swanberg**
Boston University
& Google Research NY
marikas@bu.edu

## Abstract

We propose reconstruction advantage measures to audit label privatization mechanisms. A reconstruction advantage measure quantifies the increase in an attacker's ability to infer the true label of an unlabeled example when provided with a *private* version of the labels in a dataset (e.g., aggregate of labels from different users or noisy labels output by randomized response), compared to an attacker that only observes the feature vectors, but may have prior knowledge of the correlation between features and labels. We consider two such auditing measures: one additive, and one multiplicative. These incorporate previous approaches taken in the literature on empirical auditing and differential privacy. The measures allow us to place a variety of proposed privatization schemes—some differentially private, some not—on the same footing. We analyze these measures theoretically under a distributional model which encapsulates reasonable adversarial settings. We also quantify their behavior empirically on real and simulated prediction tasks. Across a range of experimental settings, we find that differentially private schemes dominate or match the privacy-utility tradeoff of more heuristic approaches.

## 1 Introduction

Data sharing and processing provides an undeniable utility to individuals and the society at large. The modern data ecosystem has played a significant role in scientific advancement, economic growth, and technologies that benefit individuals in their daily lives. Yet, the collection, processing, and sharing of sensitive information can lead to real privacy harms. Understanding the theoretical and empirical risks associated with different types of data disclosure is a rich area of research and topic of discussion. The privacy community has studied a number of empirical measures of disclosure risks against several types of attacks, including: inference attacks [29], reconstruction attacks [5], re-identification attacks [14] and label inference attacks [13, 31].

While differential privacy gives rigorous guarantees against these attacks by worst-case adversaries, it is natural to wonder whether non-DP (deterministic) mechanisms give any empirical privacy protections. This line of inquiry necessitates empirical measures of privacy that take adversarial uncertainty into account. At a fundamental level, whether one is studying reconstruction or membership inference, nearly every privacy disclosure metric comes down to *quantifying the relationship between an adversary's prior and posterior knowledge* before and after a data release.

In this work, we consider a number of ways to summarize this prior-posterior relationship, and we apply our proposed metrics to a few Privacy Enhancing Technologies (PETs). We consider two

38th Conference on Neural Information Processing Systems (NeurIPS 2024).

families of metrics: As a coarse measure, we consider the expected *additive* difference between the prior and posterior success rates; while this does not capture unlikely high-disclosure events, it gives a sense for the risk posed to an average individual. To better capture such events, we then consider the finer-grained *multiplicative* difference between the adversary's prior and posterior. These empirical measures of privacy risk allow us to relate and compare the risks of both aggregation-based and differential privacy-based PETs, and hence compare their privacy-accuracy trade-off curves: *What accuracy does one get for a given level of privacy protection?* Together, these metrics paint a more balanced and nuanced picture of the disclosure risks and accuracy trade-offs posed by aggregation and differentially private mechanisms.

We will focus on the simplest possible scenario for information sharing: the case where a user wishes to disclose a single bit of information (for instance a binary label in a classification problem). While this problem is simple to present, it already highlights a lot of the difficulties in providing measures of privacy that are meaningful for both noise-based and aggregate-based tools. Most notably, it already requires handling the correlations—known to an attacker, but unknown to the mechanism—between the visible features and the hidden, sensitive labels. Similar to [31], we incorporate adversarial knowledge by a prior, and measure the difference between the prior and the adversary's posterior distribution conditional on observed aggregates.

Understanding binary labels already has practical implications. For instance, with Chrome's proposed conversion reporting API [1], the event of a user converting after clicking on an online ad — buying a product, signing up for a newsletter, installing an app, etc. — or not is considered sensitive, and therefore is reported only with some noise. However, once reported, ad tech providers can use features associated with an ad click (impression information, publisher information, etc.) to train models that can predict future conversions.

**Our contribution.** We make several contributions: (i) We introduce additive reconstruction advantage measures as ways to quantify the amount of leakage associated with a generic (not necessarily differentially private) privacy mechanism. These are variants and extensions of the Expected Attack Utility (EAU) and advantage contained in [31]. (ii) We quantify such measures for randomized response and random label aggregation under different correlation assumptions with public data. (iii) We consider a more demanding multiplicative reconstruction advantage measure (in the spirit of predecessors of differential privacy [17, 10]), and again quantify this advantage for randomized response and random label aggregation. (iv) We conduct a series of experiments on benchmark and synthetic datasets measuring the privacy-utility tradeoff of a number of basic mechanisms, including randomized response, label aggregation and label aggregation plus Laplace or geometric noise. Remarkably, these experiments show that learning with aggregate labels tends to be strictly harder in practice than learning with randomly perturbed labels. That is, differentially private schemes dominate or match the privacy-utility tradeoff of label aggregate schemes for all or most privacy levels and for both types of advantage measure, a conclusion we have not seen clearly spelled out in prior work on privacy. Even measured "on their own turf", deterministic aggregation—which lack provable guarantees like differential privacy—do not provide a significant advantage.

**Prior Work.** Our measures of privacy risk relate closely to previous work on membership inference and auditing, as well as to variants of differential privacy that assume adversarial uncertainty. We discuss these in Section 3.3, after defining our measures.

**Reproducibility.** For the sake of full reproducibility of our experimental setting and results, our code is available at the link `https://github.com/google-research/google-research/tree/master/auditing_privacy_via_lia`.

## 2 Preliminaries

For a natural number $n$, let $[n] = \{i \in \mathbb{N} : i \leq n\}$. Let $\mathcal{X}$ denote a feature (or instance) space and $\mathcal{Y}$ be a binary ($\mathcal{Y} = \{0, 1\}$) or multiclass ($\mathcal{Y} = [c]$) label space. We assume the existence of a joint distribution $\mathcal{D}$ on $\mathcal{X} \times \mathcal{Y}$, encoding the correlation between the input features and the labels. The marginal distribution over $\mathcal{X}$ will be denoted by $\mathcal{D}_{\mathcal{X}}$, and the conditional distribution of $y$ given $x$ will be denoted by $\mathcal{D}_{\mathcal{Y}|x}$. For distributions over binary labels, we denote by $\eta(x) = \mathbb{P}_{y \sim \mathcal{D}_{\mathcal{Y}|x}}(y = 1|x)$ the probability of drawing label 1 conditioned on feature vector $x$. In the multiclass case, we will use $\eta(x, a) = \mathbb{P}_{y \sim \mathcal{D}_{\mathcal{Y}|x}}(y = a|x)$, for $a \in \mathcal{Y} = [c]$. We define a dataset $S = \langle (x_1, y_1), \ldots, (x_m, y_m) \rangle$ as a sequence of pairs $(x_i, y_i)$, each one drawn i.i.d. from $\mathcal{D}$. We use $\mathbf{x} = (x_1, \ldots, x_m)$ to denote the features in $S$ and $\mathbf{y} = (y_1, \ldots, y_m)$ to denote the corresponding labels. We will sometimes abuse the

notation in the math, and write for brevity $(\mathbf{x}, \mathbf{y})$ instead of $(x_1, y_1), \ldots, (x_m, y_m)$. Consistent with this notation abuse, we will often use $\mathcal{X}^m \times \mathcal{Y}^m$ and $(\mathcal{X} \times \mathcal{Y})^m$ interchangeably—no ambiguity will arise.

In order to unify the study of privacy enhancing technologies, for the rest of the paper we model PETs as (possibly randomized) functions $\mathcal{M} : (\mathcal{X} \times \mathcal{Y})^m \to \mathcal{Z}$. These functions map a collection of $m$ labeled examples to a privacy protected representation in the domain $\mathcal{Z}$. The domain $\mathcal{Z}$ depends on the PET but, throughout the paper, it will be clear from the context.

In our setting, we consider PETs that protect labels and release features in the clear (so, strictly speaking, the output of a PET should also include $\mathbf{x}$ itself; we leave implicit in our notation). Despite this, our reconstruction advantage measures can be applied to any PET, we recall below two standard (and very basic) PETs that have been proposed for various Ads metrics APIs: one which satisfies label differential privacy, and one which relies on random aggregation. These are the two PETs on which we will also be able to give theoretical guarantees. We formulate them for binary classification.

**Definition 2.1.** *Given $\epsilon \geq 0$, we say that a (randomized) algorithm $A$ that takes as input $S$ is $\epsilon$-Label Differentially Private ($\epsilon$-Label DP) if for any two datasets $S$ and $S'$ that differ in the label of a single sample we have $\mathbb{P}(A(S) \in B) \leq e^{\epsilon} \, \mathbb{P}(A(S') \in B)$, where $B$ is any subset of the output space of $A$.*

Randomized Response (RR) is a classical [30] way of achieving $\epsilon$-Label DP. In the binary classification case, RR with privacy parameter $\pi = 1/(1 + e^{\epsilon})$ simply works by randomly flipping each label $y_j$ in the dataset with independent probability $\pi$ before revealing it to the learning algorithm. For a fixed $\epsilon > 0$, RR corresponds to the function $\mathcal{M}_{\mathrm{RR}}(\mathbf{x}, \mathbf{y}) = \tilde{\mathbf{y}} = (\tilde{y}_1, \ldots, \tilde{y}_m) \in \{0, 1\}^m$, where $\tilde{y}_i = 1 - y_i$ with probability $\pi = 1/(1 + e^{\epsilon})$ and equal to $y_i$ with probability $1 - \pi$, independently for each $i \in [m]$.

A completely different label privacy mechanism, still very utilized in practice (see, e.g., the Privacy Sandbox API in Google [1] and the Apple SKAN initiative [2]) is one based on (random) label aggregation, whereby the dataset $S$ gets partitioned uniformly at random into *bags* of a given size $k$, $S = \langle (x_{11}, y_{11}), \ldots, (x_{1k}, y_{1k}), \ldots, (x_{n1}, y_{n1}), \ldots, (x_{nk}, y_{nk}) \rangle$, and only feature vectors and the fraction of positive labels in each bag are revealed to the attacker/learning algorithm. In other words, the attacker has access to $S$ via a collection $\{(\mathcal{B}_i, \alpha_i), \ i \in [n]\}$ of $n$ labeled bags of size $k$, with $m = nk$, where $\mathcal{B}_i = \{x_{ij} \colon j \in [k]\}$, $\alpha_i = \frac{1}{k} \sum_{j=1}^{k} y_{ij}$ is the label proportion (fraction of labels "1") in the $i$-th bag, and all the involved samples $(x_{ij}, y_{ij})$ are drawn i.i.d. from $\mathcal{D}$. Thus, the attacker receives information about the $m$ labels $y_{ij}$ of the $m$ instances $x_{ij}$ from dataset $S$ only in the aggregate form determined by the $n$ label proportions $\alpha_i$. Note, however, that the feature vectors $x_{ij}$ are individually observed. From an attacker viewpoint, this setting is sometimes called Learning from Label Proportions (LLP). Hence, LLP corresponds to the function $\mathcal{M}_{\mathrm{LLP}}(\mathbf{x}, \mathbf{y}) = (\alpha_1, \ldots, \alpha_n) \in [0, 1]^n$.

**Learning from privatized labels.** Randomized response is especially appealing from a practical point of view, since privatized data with label flipping can be handled by many prominent learning algorithms as is, with some tuning of their hyper-parameters. Often the theoretical guarantees of these learners in terms of sample complexity are only deteriorated by some constant that depends on the label noise level, see, for example [27]. To improve accuracy, we debias the gradients in a post-processing step similar to Equation (7) in [19]; we discuss the debiasing details in Appendix C.

On the other hand, a simple and very well-known method for learning from aggregate labels is the one that of [33] call Empirical Proportion Risk Minimization. In fact, different versions of this algorithm are discussed in the literature without a clear reference to its origin. In [12], the authors simply call this algorithm the Proportion Matching algorithm (PROPMATCH), and we shall adopt their terminology here. Given a loss function $\ell : \mathbb{R} \times \mathbb{R} \to \mathbb{R}^{+}$, a hypothesis set of functions $\mathcal{H} \subset \mathbb{R}^{\mathcal{X}}$, mapping $\mathcal{X}$ to a (convex) prediction space $\widehat{\mathcal{Y}} \subseteq \mathbb{R}$, and a collection $\{(\mathcal{B}_i, \alpha_i), \ i \in [n]\}$ of $n$ labeled bags of size $k$, PROPMATCH minimizes the empirical *proportion matching loss*, i.e., it solves the following optimization problem: $\min_{h \in \mathcal{H}} \sum_{i=1}^{n} \ell \left( \frac{1}{k} \sum_{j=1}^{k} h(x_{ij}), \alpha_i \right)$. It is known that the above method is consistent (see for instance [12]). That is, training a model this way – under some mild conditions on $\ell$ and $\mathcal{H}$ and for a large enough sample – results in learning a model $h$ that minimizes the expected *event level* loss $\mathbb{E}_{(x,y) \sim \mathcal{D}}[\ell(h(x), y)]$. Finally, for our baselines, it will be helpful to consider a PET that reveals no label information at all: $\mathcal{M}_{\perp}(\mathbf{x}, \mathbf{y}) = \perp$.

# 3 Auditing Large-Scale Label Inference

In this section, we propose a number of *auditing metrics* to measure the risk of large-scale label reconstruction. Unlike DP auditing techniques which focus on worst-case guarantees, we will focus here on distributional guarantees. In doing so, we extend the (additive) reconstruction advantage definition introduced in [31] to the LLP setting and propose meaningful variants of it.

Our reconstruction advantage measures can be applied to virtually any PET for auditing purposes. For concreteness, we will also provide analytical bounds on such measures when applied to PETs like RR and label aggregation. On one hand, these bounds help illustrate the precise dependence on the data distribution $\mathcal{D}$. On the other, they will pave the way for our experimental findings in Section 4.

A reconstruction advantage metric is grounded in the following natural privacy question: *How much does releasing the output of a PET increase the risk of label inference compared to not releasing anything?* This corresponds naturally to measuring an attacker's *prior* over a target person's label compared to the attacker's *posterior* after viewing the mechanism output.

We model the attacker (often called 'adversary' later on) as a function $\mathcal{A} : \mathcal{X}^m \times \mathcal{Z} \to \mathcal{Y}^m$ that maps the features $\mathbf{x}$ and the output of a PET $\mathcal{M}(\mathbf{x}, \mathbf{y})$ to a vector of predicted labels, one for each example. We compare this attacker's success to the the *uninformed* or *prior* attacker that gets $\mathbf{x}$ and the PET that reveals no label information $\mathcal{M}_\perp(\mathbf{x}, \mathbf{y}) = \perp$. To measure the efficacy of an attacker compared to its prior, we define a number of *attack utility* variants.

The *Expected Attack Utility* of adversary $\mathcal{A}$ using information from PET $\mathcal{M}$ on a collection of $m$ examples drawn i.i.d. from a distribution $\mathcal{D}$ is defined as:

$$\mathrm{EAU}(\mathcal{A}, \mathcal{M}, \mathcal{D}) = \mathop{\mathbb{P}}_{(\mathbf{x},\mathbf{y})\sim\mathcal{D}^m,\, i\sim\mathrm{Uniform}([m]),\, \text{coins of }\mathcal{M}} \big( \mathcal{A}(\mathbf{x}, \mathcal{M}(\mathbf{x}, \mathbf{y}))_i = y_i \big) ,$$

where $\mathcal{A}(\cdot, \cdot)_i$ is the $i$-th component of vector $\mathcal{A}(\cdot, \cdot)$. In words, the expected attack utility of adversary $\mathcal{A}$ is the probability that $\mathcal{A}$ correctly guesses the label of a randomly chosen example when provided the features and the output of $\mathcal{M}$. Equivalently, this is the expected fraction of the $m$ examples that the adversary predicts the correct label for. The adversary's success rate may depend on the distribution over features and labels. For example, if labels are entirely determined by features, then our metric should reflect that privatized labels (for any mechanism $\mathcal{M}$) reveal no additional information about the true labels. To control for the information that features inherently reveal about labels, we assume that the adversary has knowledge of the data distribution $\mathcal{D}$ over $\mathcal{X} \times \mathcal{Y}$, either completely (e.g., Definition 3.1) or approximately through learning on disjoint data (as in our experiments in Section 4).

Further, we define the *Individual Expected Attack Utility* on input data $x_i$ as

$$\mathrm{IEAU}_i(\mathcal{A}, \mathcal{M}, \mathcal{D}, x_i) = \mathop{\mathbb{P}}_{y_i\sim\mathcal{D}_{\mathcal{Y}|x_i}(\mathbf{x}^{(-i)},\mathbf{y}^{(-i)})\sim\mathcal{D}^{m-1},\, \text{coins of }\mathcal{M}} \left( \mathcal{A}(\mathbf{x}, \mathcal{M}(\mathbf{x}, \mathbf{y}))_i = y_i \,\Big|\, x_i \right) ,$$

where $\mathbf{x}^{(-i)}$ is $\mathbf{x}$ with the $i$-th item $x_i$ dropped, and likewise for $\mathbf{y}^{(-i)}$. The quantity $\mathrm{IEAU}_i(\mathcal{A}, \mathcal{M}, \mathcal{D}, x_i)$ emphasizes the (expected) attack utility on a specific piece of data $x_i$ when the associated label $y_i$ is drawn from the conditional distribution $\mathcal{D}_{\mathcal{Y}|x_i}$. For instance, $\mathrm{IEAU}_i(\mathcal{A}, \mathcal{M}_{\mathrm{LLP}}, \mathcal{D}, x_i)$ measures, for a given $x_i$, the chance that an attacker is able to reconstruct the associated label $y_i$, if $y_i$ generated from $\mathcal{D}_{\mathcal{Y}|x_i}$ and, by virtue of mechanism $\mathcal{M}_{\mathrm{LLP}}$, $y_i$ becomes part of a bag (of some size $k$), only the bag label proportion $\alpha$ being revealed to the attacker.

In order to measure the *increase* in risk incurred by releasing the output of a PET, we consider the attack utility of an optimal adversary in two scenarios: one in which the adversary gets the features, $\mathbf{x}$, together with the output of the PET, $\mathcal{M}(\mathbf{x}, \mathbf{y})$, and an alternate setting where the adversary gets only the features (which is equivalent to using $\mathcal{M}_\perp$). We call the difference in attack utility between the informed and uninformed adversary the *attack advantage*. Intuitively, the attack advantage measures the label reidentification risk that can be attributed to the PET rather than to correlations between the features $\mathbf{x}$ and labels $\mathbf{y}$ which are inherent in the distribution $\mathcal{D}$. Since we have given two notions of attack utility above, we have two corresponding notions of attack advantage. Below is an *additive* version, based on utility *differences*, later on (Section 3.2) we will also consider a *multiplicative* version, based on utility *ratios*.

**Definition 3.1.** *Given a PET $\mathcal{M}$ for a set of $m$ examples drawn from a data distribution $\mathcal{D}$, the (additive)* Expected attack Advantage *is defined as*

$$\mathrm{EAdv}(\mathcal{M}, \mathcal{D}) = \sup_{\mathcal{A}_{informed}} \mathrm{EAU}(\mathcal{A}_{informed}, \mathcal{M}, \mathcal{D}) \quad - \sup_{\mathcal{A}_{uninformed}} \mathrm{EAU}(\mathcal{A}_{uninformed}, \mathcal{M}_\perp, \mathcal{D}).$$

*Similarly, when the $i$-th item $x_i$ is kept frozen, the* Individual Expected attack Advantage *is defined as*

$$\mathrm{IEAdv}(\mathcal{M}, \mathcal{D}, x_i) = \sup_{\mathcal{A}_{informed}} \mathrm{IEAU}(\mathcal{A}_{informed}, \mathcal{M}, \mathcal{D}, x_i) - \sup_{\mathcal{A}_{uninformed}} \mathrm{IEAU}(\mathcal{A}_{uninformed}, \mathcal{M}_\perp, \mathcal{D}, x_i).$$

Note that RR is generally described in terms of its behavior on a single example $(x, y)$, rather than a collection of $m$ i.i.d. samples. Likewise, label aggregation operates on each bag of size $k << m$ independently. However, since the data is i.i.d., it is easy to see that the attack advantage for RR is independent of the number of examples $m$, while the attack advantage for (random) label aggregation will only depend on $k$, rather than $m$.

Appendix A.5 contains further variants of attack advantage, like one that accounts for the *tail* of the attack advantage distribution, as well as associated bounds. These can be stretched to the point where the feature vector $\mathbf{x} = (x_1, \ldots, x_m)$ is arbitrary, and only the properties of the condition distribution $\mathcal{D}_{\mathcal{Y}|x}$ is factored in.

## 3.1 Bounding the Additive Attack Advantage

As a warm up, we begin by studying the additive expected attack advantage for LLP when the labels are independent of the features. Note that, since the features $\mathbf{x}$ do not play any role in this simplified setting, the notions of advantage coincide. We recall again that all results involving $\mathcal{M}_{\mathrm{LLP}}$ (Theorems 3.2, 3.3 and 3.4 below) deal with a *single* bag when computing advantage measures (hence we will write $\alpha$ instead of $\alpha_i$). This is because distribution $\mathcal{D}$ is known to the adversary, and the bags the adversary observes are independent of one another, hence there is no extra advantage from operating on an entire dataset of bags at once. All proofs are given in the appendix.

**Theorem 3.2.** *Fix a data distribution $\mathcal{D}$, let $p = \mathbb{P}_{(x,y)\sim\mathcal{D}}(y = 1)$, and fix an arbitrary threshold $\beta \in [0, 1/2]$. If labels are independent of features (i.e., $\mathcal{D}$ is a product of distributions over $\mathcal{X}$ and $\mathcal{Y}$), then for all bag sizes $k \geq 1$ we have:*

$$\mathrm{EAdv}(\mathcal{M}_{LLP}, \mathcal{D}) = \min\{p, 1-p\} - \mathbb{E}_\alpha[\min\{\alpha, 1-\alpha\}] \leq \begin{cases} \sqrt{\frac{p(1-p)}{k}} & \text{if } p \in [0, 1] \\ e^{-\Omega(\beta^2 k)} & \text{if } |p - 1/2| \geq \beta, \end{cases}$$

*where $\Omega(\cdot)$ hides constants independent of $\beta$ and $k$.*

A couple of remarks are in order. First, observe that, as expected, the advantage $\mathrm{EAdv}(\mathcal{M}_{\mathrm{LLP}}, \mathcal{D})$ is always non-negative. This can be easily derived by noting that $\mathbb{E}[\alpha] = p$ and then applying Jensen's inequality to the concave function $x \mapsto \min\{x, 1-x\}$, for $x \in [0, 1]$. Second, despite being non-negative, Theorem 3.2 also proves the desirable property that $\mathrm{EAdv}(\mathcal{M}_{\mathrm{LLP}}, \mathcal{D})$ goes to zero as the bag size $k$ increases. The convergence rate is in general of the form $1/\sqrt{k}$, but it becomes *negative exponential* in $k$ when $p$ is bounded away from 1/2.

We now investigate general distributions over features and labels and consider in turn $\mathrm{EAdv}$ and $\mathrm{IEAdv}$.

**Theorem 3.3.** *Let $\mathcal{D}$ be an arbitrary distribution on $\mathcal{X} \times \mathcal{Y}$, $p = \mathbb{E}[\eta(x)]$, and $\mu = \mathbb{E}[\eta(x)(1-\eta(x))]$. Then, for all bag sizes $k \geq 2$ we have:*

$$\mathrm{EAdv}(\mathcal{M}_{LLP}, \mathcal{D}) = \widetilde{O}\left( \frac{\mu^{1/4}(p(1-p))^{1/4}}{\sqrt{k}} + \frac{\mu^{1/4}}{k} \right),$$

*where $\widetilde{O}$ hides logarithmic factors in $k$.*

Hence, also in this more general case of LLP, the advantage converges to zero as the bag size $k$ grows large. Compared to the rate in Theorem 3.2, we are only losing the $\log k$ factors implicit in the $\widetilde{O}$ notation. This is because, when applied to the scenario where labels and features are independent, $\eta(x) = p$ is constant with $x$, so that $\mu = p(1-p)$, and the first term becomes $\sqrt{\frac{p(1-p)}{k}}$, while the second one reads $\frac{(p(1-p))^{1/4}}{k}$, which is lower order when $k$ is large. We strongly believe that the tighter gap-dependent analysis we carried out for Theorem 3.2 extends to the more general scenario of Theorem 3.3, but we leave this as an open question.

The corresponding bound for the individual expected attack advantage is given next.

**Theorem 3.4.** *Under the same assumptions and notation as in Theorem 3.3, we have, for $\mu > 0$, and $k \geq \frac{2}{\mu} \log(1/\mu)$,*

$$\text{IEAdv}(\mathcal{M}_{LLP}, \mathcal{D}, x_i) = \mu_i \widetilde{O}\left( (p(1-p))^{1/4} \sqrt{\frac{\mu}{k} + \frac{1}{\mu^{3/2} k}} + \sqrt{\frac{\mu}{k^2} + \frac{1}{\mu^{3/2} k^2}} \right),$$

*where $\mu_i = \eta(x_i)(1 - \eta(x_i))$, and $\widetilde{O}$ hides logarithmic factors in $k$.*

In the bounds of Theorems 3.2, 3.3 and 3.4 the dependence on the data distribution $\mathcal{D}$ is encoded in $p(1-p)$ and $\mu$; as $p(1-p)$ and/or $\mu$ gets smaller we should naturally expect a smaller advantage, as the adversary is facing an easier label prediction problem. Appendix A.5 contains further distribution-dependent results, like a bound on the advantage that, conditioned on $\mathbf{x} = (x_1, \ldots, x_m)$, is of the form $\frac{1}{k} \sum_{i=1}^{k} \frac{\mu_i}{\sqrt{\sum_{j:j \neq i} \mu_j}}$, where $\mu_i = \eta(x_i)(1 - \eta(x_i))$.

We now provide the corresponding expression for the additive attack advantage for RR (at a given level $\epsilon \geq 0$). We recall that the results of [31] imply that every $\epsilon$-label-DP PET $\mathcal{M}$ has advantage bounded as $\text{EAdv}(\mathcal{M}, \mathcal{D}) \leq 1 - \frac{2}{1+e^\epsilon}$. However, one drawback of this bound is that it is distribution independent (or, rather, worst case over distribution $\mathcal{D}$). Yet, the attack advantage depends heavily on $\mathcal{D}$. For an extreme example, if we have $\mathbb{P}_{(x,y)\sim\mathcal{D}}(y = 1) = 1$, the attack advantage is zero for every PET, which is not directly captured by only relying on the properties of $\epsilon$-DP.

Recall that, since RR operates on each example independently, the advantage is independent of the number of examples $m$ (a formal proof is given in Appendix A.6). Hence we derive a bound for $\text{IEAdv}(\mathcal{M}_{\text{RR}}, \mathcal{D}, x_1)$, and an expression for the optimal adversary under RR (see Appendix A.7).

**Theorem 3.5.** *For any data distribution $\mathcal{D}$, the individual expected attack advantage $\text{IEAdv}(\mathcal{M}_{RR}, \mathcal{D}, x_1)$ for randomized response with privacy parameter $\pi = \frac{1}{1+e^\epsilon}$ is equal to* $\left( \min\{\eta(x_1), 1 - \eta(x_1)\} - \pi \right) \cdot \mathbb{I}\{\eta(x_1) \in [\pi, 1 - \pi]\}$ .

We use the above expression in our experiments (Section 4) to estimate the attack advantage of RR for various values of $\epsilon$. Being distribution dependent, this expression leads to much tighter bounds on the attack advantage. For example at $\epsilon = 1$, the bound from [31] is $1 - \frac{2}{1+\epsilon} \approx 0.46$, while for one dataset used in our experiments we see that the attack advantage for RR at $\epsilon = 1$ is only 0.00095.

## 3.2 Multiplicative Attack Advantage

The definitions of attack advantage given so far measure the *absolute* change in successful reconstruction, not the *relative* change. Moreover, they do not fully capture the different levels of confidence in the reconstruction, since they involve an expectation over either $(\mathbf{x}, \mathbf{y})$ or $\mathbf{y}$ given $\mathbf{x}$. Next, we give an additional set of definitions that capture these important nuances. For the sake of brevity, in this section we directly instantiate the adversaries $\mathcal{A}_{\text{informed}}$ and $\mathcal{A}_{\text{uninformed}}$ to Bayes optimal predictors.

**Definition 3.6.** *Given data distribution $\mathcal{D}$, mechanism $\mathcal{M}$, index $i$, and pair of labels $a, b \in \mathcal{Y}$, the multiplicative advantage is the difference of log odds ratios:*

$$I_{a,b}(\mathcal{M}, \mathcal{D}, \mathbf{x}, z, i) = \log \frac{\pi_i(\mathbf{x}, z, a)}{\pi_i(\mathbf{x}, z, b)} - \log \frac{\eta(x_i, a)}{\eta(x_i, b)}$$

*where $\eta(x_i, a) = \mathbb{P}(y_i = a \mid x_i)$ and $\pi_i(\mathbf{x}, z, a) = \mathbb{P}_{\mathbf{y} \sim \mathcal{D}_{\mathcal{Y}|\mathbf{x}}^m}(y_i = a \mid \mathbf{x}, \mathcal{M}(\mathbf{x}, \mathbf{y}) = z)$ is the probability of $y_i = a$ after observing the output $z$ of $\mathcal{M}$. We denote the binary case by $I_{1,0}(\cdot) = I(\cdot)$ and shorten $\eta(x_i) := \eta(x_i, 1)$ and $\pi_i(\mathbf{x}, z) := \pi_i(\mathbf{x}, z, 1)$.*

This particular formulation has the advantage that $I_{a,b}(\mathcal{M}, \mathcal{D}, \mathbf{x}, z, i)$ is large in absolute value whenever there is a large *relative* change in either label probability. This also entails that a multiplicative-style advantage metric is generally stronger than an additive one.

When $\eta(x_i)$ and $\pi_i(\mathbf{x}, z)$ are both less than 1/2 (in the binary setting), we have $I(\mathcal{M}, \mathcal{D}, \mathbf{x}, z, i) = \Theta(\log \frac{\pi_i(\mathbf{x}, z)}{\eta(x_i)})$, with a symmetric expression for the case that both are close to 1. Because we assume i.i.d. sampling, one can easily see that $I_{a,b}(\mathcal{M}, \mathcal{D}, \mathbf{x}, z, i) = \log \frac{\mathbb{P}(\mathcal{M}(\mathbf{x}, \mathbf{y}) = z \mid y_i = a, \mathbf{x})}{\mathbb{P}(\mathcal{M}(\mathbf{x}, \mathbf{y}) = z \mid y_i = b, \mathbf{x})}$ , which makes it clear that for $\epsilon$-label DP mechanisms like $\mathcal{M}_{\text{RR}}$ (at level $\epsilon$) the above log ratio is at most $\epsilon$

in absolute value for all $\mathbf{x}$ and $\mathbf{y}$. On the other hand, for $\mathcal{M}_{\text{LLP}}$, there is always a small chance that all the examples in the bag will have the same label making the above log ratio infinite. However, for large $k$ and distributions $\mathcal{D}$ in which the $\eta(x, a)$ values are not too close to 0 or 1, the leakage $I_{a,b}(\mathcal{M}, \mathcal{D}, \mathbf{x}, z, i)$ might be small in most cases.

What values of multiplicative advantage should be considered acceptable? We argue that this probability should be viewed as a probability of system failure and set appropriately small. (For example, when running with $\mathcal{M}_{\text{LLP}}$ on modern-scale data sets, we might create billions of bags; even a tiny probability of failure can lead to many bags whose individuals have their labels revealed exactly). The next theorem gives high probability bounds for $\mathcal{M}_{\text{LLP}}$ in the simple case when the distribution $\mathcal{D}$ is a product distribution.

**Theorem 3.7.** *Fix a data distribution $\mathcal{D}$ over features and binary labels, let $p = \mathbb{P}_{(x,y) \sim \mathcal{D}}(y = 1)$, and assume the labels are independent of features (i.e., $\mathcal{D}$ is a product of distributions over $\mathcal{X}$ and $\mathcal{Y}$). Then there are universal constants $c_1, c_2 > 0$ such that for $p \in (0, 1)$, all bag sizes $k \geq \frac{c_1 \ln(1/\delta)}{p(1-p)}$, and all $i \in [k]$ we have*

$$\mathbb{P}_{(\mathbf{x},\mathbf{y}) \sim \mathcal{D}^m_{\mathcal{Y}|x_i}} \left( \left| I_{1,0}(\mathcal{M}_{LLP}, \mathcal{D}, \mathbf{x}, \mathcal{M}_{LLP}(\mathbf{x}, \mathbf{y}), i) \right| > c_2 \sqrt{\frac{\ln(1/\delta)}{p(1-p)k}} \right) \leq \delta.$$

Then, when $k$ is sufficiently large and the conditional label probabilities $\eta(x_i)$ are all equal to some constant $p \in (0, 1)$, the probability of failure drops off exponentially as $k$ increases. When working with a data set of size $m$, we can substitute $\delta = \delta' \cdot \frac{k}{m}$ to get a bound on the probability that any bag in the dataset exhibits extreme values of leakage $I_{a,b}$. For the setting covered by Theorem 3.7, one gets that the bags must be of size $k \approx \log(n)/(p(1-p))$ for the probability of extreme bags to converge towards 0. In the experiments in Section 4, we report results on both the additive and the multiplicative reconstruction advantage criteria.

### 3.3 Connection to prior work on auditing and membership inference, and distributional DP

Our approach on quantifying attack advantage can be viewed as fitting into a recent line of work on auditing learning algorithms via membership inference attacks. This class of attacks was introduced by [21] and subsequently studied in theory [11, 18] and practice [29]. Whether used as attacks or empirical lower bounds on differential privacy parameters [22], these approaches set up a hypothesis test for the presence or absence of a particular target record in the training data [15]. They diverge in whether they consider an adversary with access to all the rest of the training data ("fully informed"), or with only distributional knowledge of the rest of the training data (usually made available to the attacker as an independent sample drawn from the same distribution).

In our setting, testing for the presence of an individual makes no sense, since individual records' features are known. In the binary case, our measures are success metrics for testing the hypothesis that a particular individual's label is 1 as opposed to 0: Multiplicative advantage bounds the ratio of true-to false-positive rates of the Bayes' optimal test, while additive advantage measures the difference of true- and false-positive rates. This perspective was also taken in previous work: for example, [26] use such a test to lower-bound the differential privacy parameters of label-DP algorithms. However, their modeling assumes a fully informed adversary (that knows all labels). In contrast, we posit a reasonable model of adversarial uncertainty in order to to look at the risks of mechanisms such as label aggregation. In this respect, our approach follows more in the line of membership inference attacks that use only distributional knowledge, or to variants of differential privacy that assume adversarial uncertainty such as [9, 7, 16].

Further relevant references include [25, 32], which have been developed more in the Cryptography literature. Like here, those papers propose to measure prediction advantage by comparing the information status of an agent before and after seen the obfuscated data. Yet, they do not consider the public features/private label mixed setting we study here, which is clearly necessary when investigating, for instance, the LLP mechanism. As a result, the measures proposed in [25, 32] generally fail to distinguish between the information gained as a result of: (1) correlation to the public features, and (2) the mechanism output. Hence, they overestimate the revelation of the mechanism, at times to the extreme. Consider, for clarity, the case where $\eta(x)$ is either 0 or 1 for all $x$. In this case the adversary's inference advantage is zero for any reasonable measure of advantage, since the adversary has perfect knowledge of the labels without the mechanism output.

# 4 Experiments

In this section, we demonstrate how our advantage measures can be used to quantify and compare the potential privacy leakage of RR, LLP, and two additional PETs with a range of privacy parameters. See Appendix C for full details on our PETs, experimental setup, and additional results.

**Mechanisms.** We empirically evaluate a number of proposed mechanisms for label privacy: randomized response (RR), Label Proportions (LLP), and two further PETs: LLP+Lap, where (zero-mean) Laplace noise is added to the label aggregate in each bag, and LLP+Geom, where geometric noise is added and then clipped so the estimated proportion lies in $[0, 1]$. The noise scale for both LLP+Lap and LLP+Geom is chosen so that the PETs satisfy $\epsilon$ label differential privacy. Postprocessed clipped geometric noise is the mechanism for releasing a binary sum that is optimal among all differentially private mechanisms for several loss measures [20]. To minimize squared error, the optimal post-processing is to correct for the bias introduced by clipping. We empirically study these advantage measures on a variety of synthetic and benchmark datasets, reporting either the AUC vs. advantage trade-off or the prior-vs-posterior scatter plots, that help shed light on the distribution of our advantage measures on different points in the dataset for the various PETs.

**Estimating class conditionals for advantage and PET Utility.** The advantage measures for all PETs we consider can all be computed as a function of the class conditional probabilities $\eta(x) = \mathbb{P}(y = 1|x)$. In our synthetic datasets we compute the advantage measures for different distributions on $\eta(x)$; however, for non-synthetic data sets, the value of $\eta(x)$ is not explicitly known. Instead, we estimate $\eta(x)$ for each $x$ by training a classifier $h$ (without any PETs) and use the prediction probability $h(x)$ as a proxy for $\eta(x)$. We compute the figure of merit (additive or multiplicative advantage) for the optimal informed and uninformed attackers using these estimates.

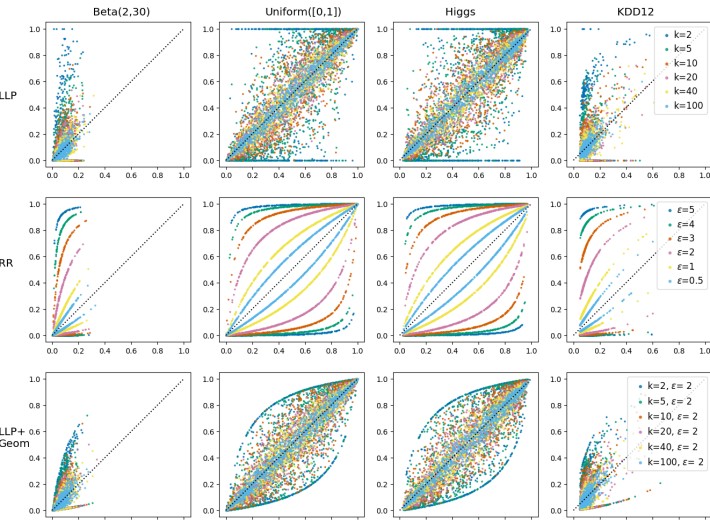

Figure 1: Prior-posterior scatter plots for LLP, RR, and LLP+Geom from two synthetic datasets (where the prior $\eta(x)$ is drawn) and the two real-world datasets (where $\eta(x)$ is approximated). The colors of the dots correspond to different parameter values for the PETs. For each bag size $k$ and distribution, we did 1000 independent runs. The further a point is from the $y = x$ dotted line, the more is revealed about its label as a result of the PET.

We also measure the utility of each PET for trained models. For each dataset, PET, and privacy parameters, we apply the PET to the training labels to produce a privatized version. We then train a model on the privatized data using minibatch gradient descent with the Adam optimizer [24] and a loss function designed for the PET. For RR, the loss debiases the binary crossentropy loss when evaluated on the RR labels, and for LLP, LLP+Laplace, and LLP+Geom, we minimize the Empirical Proportion Risk defined in Section 2. For each dataset, PET, and privacy parameters, we perform a grid search over the learning rate parameter and report the test AUC of the best performing learning rate. All utility results are averaged over multiple runs. The maximum standard error in the mean for the reported AUCs is 0.0076 and the vast majority are less than 0.002.

**Computing posterior distributions for each PET.** Given a list of priors (or estimated priors) for each data point $\{\eta(x_i)\}_{i=1}^n$, we analytically compute the posterior probabilities for each of the PETs we consider.

First, we apply Bayes' theorem and use the fact that the $x_i$'s are independent:

$$\mathbb{P}(y_i = 1|x, \mathcal{M}(x, y) = z) = \frac{\mathbb{P}(\mathcal{M}(x, y) = z|x, y_i = 1) \cdot \mathbb{P}(y_i = 1|x_i)}{\mathbb{P}(\mathcal{M}(x, y) = z|x)}. \tag{1}$$

For RR with flipping probability $p$, Equation 1 evaluates to $\frac{p\eta(x_i)}{p\eta(x_i)-(1-p)(1-\eta(x_i))}$ if the outcome of $\mathcal{M}_{\text{RR}}(y_i) = 0$, and to $\frac{(1-p)\eta(x_i)}{(1-p)\eta(x_i)-p(1-\eta(x_i))}$ if the outcome is 1.

For label aggregation (LLP), the bags are independent, so we consider only a single bag at a time. $\mathcal{M}_{\text{LLP}}$ is deterministic, so in the numerator $\mathbb{P}(\mathcal{M}_{\text{LLP}}(x, y) = z \,|\, x, y_i = 1)$ is the probability density at $z$ of a Poisson binomial distribution with flipping probabilities $(\eta(x_1), \ldots, \eta(x_{i-1}), 1, \eta(x_{i+1}), \ldots, \eta(x_k))$. Similarly, the denominator is the probability density at $z$ of a Poisson binomial distribution with flipping probabilities $(\eta(x_1), \ldots, \eta(x_k))$.

For $\mathcal{M}_{\text{Lap-LLP}}$ and $\mathcal{M}_{\text{Geom-LLP}}$, the terms in Equation 1 are convolutions of the posterior for LLP and the Laplace or Geometric distributions, respectively.

**Results.** Our results are reported in Figures 1, 2, and 3, as well as in Appendix C. We used two synthetic datasets and two real-world datasets (Higgs and KDD12). Notice that Higgs is a relatively balanced dataset, while KDD12 is a quite imbalanced one.

Figure 1 give scatter plots of prior $\mathbb{P}(y = 1 \,|\, \mathbf{x})$ vs. posterior $\mathbb{P}(y = 1 \,|\, \mathbf{x}, \mathcal{M}(\mathbf{x}, \mathbf{y}) = z)$ distribution on the four datasets for LLP, RR and LLP+Geom with different colors for each value of parameters $k$ (for LLP and LLP+Geom) and $\epsilon$ (for RR and LLP+Geom). The behavior of LLP+Lap is reported in Appendix C for completeness, but it turned out to be similar to that of LLP+Geom.

It is instructive to observe how the points spread w.r.t. the main diagonal $y = x$. Points that are on the diagonal have *no label privacy loss* as a result of the PET, because the posterior is identical to the prior. On the other hand, points with a posterior of 0 or 1 have *complete privacy loss*, since the posterior on the private label is deterministic. In general, a wider spread away from $y = x$ indicates more privacy loss. The scatter plots tend to form spindle shapes whose width is determined by the privacy parameters of the PETs. Yet, there is a substantial difference between LLP and RR. While RR generates points on the boundaries of the spindles (middle row), LLP tends to spread such points more uniformly. Moreover, some of the points for LLP lie on the edges of the square $[0, 1]^2$, which correspond to an *infinite* multiplicative advantage. LLP+Lap (bottom row) is somewhere in between, in that the points are also located inside the spindles, but never on the edges of the square. Note that a spindle boundary is the set of points having the same multiplicative advantage measure. Moreover, these differences become more pronounced on skewed datasets like KDD12 as the probability of homogenous bags (bags of all 0 labels or 1 labels) become much more likely.

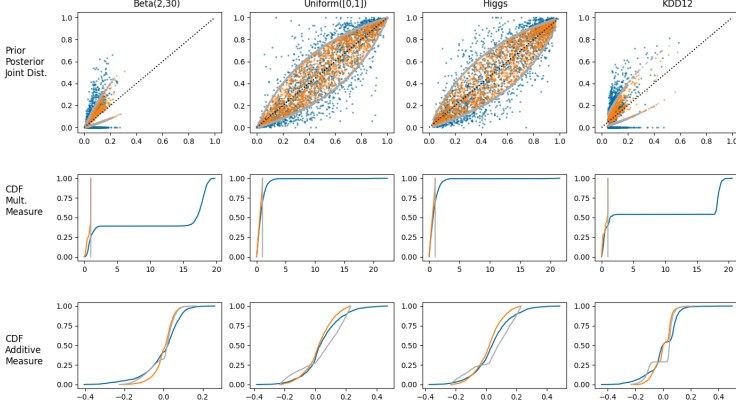

Figure 2: **Top:** Prior-posterior scatter plots for RR (grey), LLP (blue), and LLP+Geom (orange) with $\epsilon = 1$ and $k = 8$ on the same datasets as in Figure 1. With these choice of parameters, the three mechanisms roughly achieve the same AUC on Higgs. **Middle:** Empirical CDFs of (the absolute value of) the multiplicative advantage for the three PETs on the four datasets. **Bottom:** CDFs of the additive advantage.

These scatter plots are already suggestive of the expected behavior of the utility-privacy tradeoff curves that will come next in Figure 3. Given an allowed level of privacy (as measured by either additive or multiplicative advantage) each point in the interior of the spindle associated with that privacy level will also lie in the boundary of a smaller (thus higher privacy) spindle. This will make inference harder, thereby reducing (average) utility at that level of privacy. Following this intuition, we expect RR to achieve a higher utility-privacy curve than LLP, with LLP+Geom somewhere in between. A more detailed comparison between RR and LLP+Geom (or LLP+Lap) is in Appendix D.

Figure 2 helps further illustrate the different behavior of the considered PETs vis-à-vis the advantage measure. On the four datasets, we pick here a set of parameters that make the PETs AUC-comparable on Higgs. We then plot the empirical Cumulative Distribution Functions (CDFs) on the four datasets for both multiplicative (middle row) and additive (last row) measures. While the CDFs of additive

measures are roughly similar across the three PETs, this is not the case for the highly skewed Beta(2,30) and KDD12 datasets, where one can easily spot for LLP (blue line) the presence of a significant mass of points with large multiplicative advantage. Note that in the middle row, the CDF of RR (grey line) is just a vertical line ($x = 1$) while the CDF of LLP+Geom (orange line) first follows the blue line and then the grey one. These high multiplicative advantage points cannot be detected when only relying on the CDFs of the additive measure.

**Utility vs. advantage tradeoff on benchmark datasets.** The experiments so far have compared the distribution of individual additive and multiplicative advantage induced by RR, LLP, and LLP+Geom for various parameter settings and datasets. However, the use of PETs is always a tradeoff between utility and privacy, since if we cared about privacy alone, the best strategy would be to release no information at all. Figure 3 plots the AUC vs advantage for each PET as we vary the PET's privacy parameters. Note that, since the $x$-axis is advantage, we are able to put the normally incomparable privacy parameters of the PETs on equal footing. For RR and LLP, the single privacy parameter ($\epsilon$ and $k$, respectively) traces out an AUC vs Advantage curve. Since LLP+Lap and LLP+Geom have two parameters, there is an area of achievable AUC vs Advantage pairs. For these mechanisms, we plot a separate curve for each bag size $k$ showing the tradeoff when varying $\epsilon$ for that $k$.

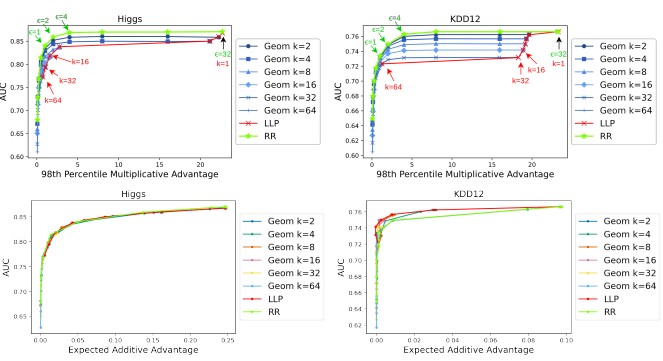

Figure 3: Privacy vs utility tradeoff curves for the various PETs on the Higgs and KDD12 datasets. Utility is measured by AUC on test set, while privacy is either the additive measure (bottom row) or the 98th-percentile of the multiplicative measure (so as to rule out the infinite multiplicative advantage cases that can occur for LLP). Each point corresponds to a setting of the privacy parameter for the PET ($\epsilon$ for RR, $k$ for LLP, and both for LLP+Geom). The $x$-coordinate is the advantage (either additive or multiplicative) value for that PET, while the $y$-coordinate is the test AUC of a model trained from the output of that PET. The AUC of the model trained without a PET roughly corresponds to the top value achieved by these curves.

When measuring privacy loss via multiplicative advantage, RR has the best privacy vs. accuracy trade-off compared to all other PETs. This is consistent with the observations in Fig. 2. In particular, the three mechanisms (with the given parameters) have almost the same AUC on Higgs. Yet when looking at the CDFs of the multiplicative measure one sees that the two DP mechanisms have bounded MAs whereas LLP has a significant number of extreme values. This trend is more pronounced for KDD12 than Higgs. Thus, for the same AUC, LLP has a higher multiplicative advantage, with a more pronounced gap for KDD12 than for Higgs. On the other hand, Fig. 2 also shows that the CDFs of the additive advantage are roughly the same for all three mechanisms (again, at the same AUC). Thus, in Fig. 3 the mechanisms have similar AUC vs advantage tradeoffs.

## 5   Discussion and Conclusions

We study ways to audit label privatization mechanisms for commonly used PETs: randomized response, random label aggregation, and combinations thereof. Together, the additive and multiplicative advantage measures we introduced paint a richer picture of the reconstruction risks posed by different parameter settings, and for the first time allow us to compare their privacy-accuracy trade-off curves.

The measures we propose are tailored to settings where the data are sampled i.i.d. from a distribution, and each record consists of public features together with one sensitive binary feature. Computing these measures empirically requires estimates of the adversary's label uncertainty, which won't be correct if the adversary has significant side information or, crucially, if the same data are re-used in multiple mechanisms. Handling such complex settings requires more general concepts like differential privacy. Another complexity emerges when we consider settings like click prediction, where a minority label (a click) is viewed as qualitatively more revelatory than the majority one (no click). While the measures we consider are agnostic to the label semantics, principled ways to incorporate complex semantics might be valuable and shed light on heuristics used in practice.

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

# A Proofs for Section 3

We start off by providing a preliminary result that will be useful throughout all proofs.

## A.1 Preliminary results

Consider, for a specific $i \in [m]$, the random variables

$$f_{1i}(\mathbf{x}) = \sup_{\mathcal{A}_{\text{informed}}} \mathbb{P}_{\mathbf{y} \sim \mathcal{D}^k_{\mathcal{Y}|\mathbf{x}}} (\mathcal{A}_{\text{informed}}(\mathbf{x}, \mathcal{M}(\mathbf{x}, \mathbf{y}))_i = y_i \mid \mathbf{x})$$

$$\tag{2}$$

$$f_{2i}(\mathbf{x}) = \sup_{\mathcal{A}_{\text{unininformed}}} \mathbb{P}_{\mathbf{y} \sim \mathcal{D}^k_{\mathcal{Y}|\mathbf{x}}} (\mathcal{A}_{\text{uninformed}}(\mathbf{x}, \mathcal{M}_\perp(\mathbf{x}, \mathbf{y}))_i = y_i \mid \mathbf{x}) .$$

The relevance of $f_{1i}$ and $f_{2i}$ stems from the fact that for the mechanisms $\mathcal{M}$ we consider in this paper, the advantages EAdv, IEAdv, and HPAdv will be defined in terms of $f_{1i}$ and $f_{2i}$.

**Lemma A.1.** *For any PET $\mathcal{M}$, any $\mathbf{x} \in \mathcal{X}$, any conditional data distribution $\mathcal{D}_{\mathcal{Y}|\mathbf{x}}$, number of examples $k$, and $i \in [m]$, we have*

$$f_{1i}(\mathbf{x}) - f_{2i}(\mathbf{x}) = \min\{\eta(x_i), 1 - \eta(x_i)\} - \mathbb{E}_{\mathbf{y} \sim \mathcal{D}_{\mathcal{Y}|\mathbf{x}}}\left[\min_{b \in \{0,1\}} \{\mathbb{P}(y_i = b \mid \mathbf{x}, \mathcal{M}(\mathbf{x}, \mathbf{y}))\} \mid \mathbf{x}\right],$$

*where $\eta : x \mapsto \mathbb{P}(y = 1 \mid x)$. Moreover, the optimal (informed) adversary $\mathcal{A}^*$ is:*

$$\mathcal{A}^*(\mathbf{x}, \mathcal{M}(\mathbf{x}, \mathbf{y}))_i := \begin{cases} 1 & \text{if } \mathbb{P}(y_i = 1 \mid \mathbf{x}, \mathcal{M}(\mathbf{x}, \mathbf{y})) \geq 1/2 \\ 0 & \text{otherwise} . \end{cases}$$

*Proof.* Set for brevity $z = \mathcal{M}_\perp(\mathbf{x}, \mathbf{y})$. Since $\mathcal{M}_\perp$ outputs null, and the $(x_j, y_j)$ pairs for $j \in [m]$ are independent, we have

$$\mathbb{P}(y_i = 1 \mid \mathbf{x}, z) = \mathbb{P}(y_i = 1 \mid x_i) = \eta(x_i),$$

It immediately follows that the best attacker $\mathcal{A}_{\text{unininformed}}$ involved in the computation of $f_{2i}(\mathbf{x})$ is the Bayes optimal predictor

$$\hat{y}_i = \arg\max_{b \in \{0,1\}} \mathbb{P}(y_i = b \mid x_i)$$

so that

$$f_{2i}(\mathbf{x}) = 1 - \min\{\eta(x_i), 1 - \eta(x_i)\} .$$

Let us now turn to $f_{1i}$ and fix any adversary $\mathcal{A}_{\text{informed}}$. We lower bound the probability that the adversary makes a mistake on $y_i$. Let $z = \mathcal{M}(\mathbf{x}, \mathbf{y})$ be the output of the PET, and $(\hat{y}_1, \ldots, \hat{y}_m) = \mathcal{A}_{\text{informed}}(\mathbf{x}, z)$ be the output of the adversary. Since $\hat{y}_i$ is $(\mathbf{x}, z)$-measurable, we have

$$\begin{aligned}
\mathbb{P}(\hat{y}_i = y_i \mid \mathbf{x}, z) &= \mathbb{P}(\hat{y}_i = 1 \mid y_i = 1, \mathbf{x}, z)\,\mathbb{P}(y_i = 1 \mid \mathbf{x}, z) \\
&\quad + \mathbb{P}(\hat{y}_i = 0 \mid y_i = 0, \mathbf{x}, z)\,\mathbb{P}(y_i = 0 \mid \mathbf{x}, z) \\
&= \mathbb{1}\{\hat{y}_i = 1\}\,\mathbb{P}(y_i = 1 \mid \mathbf{x}, z) + \mathbb{1}\{\hat{y}_i = 0\}\,\mathbb{P}(y_i = 0 \mid \mathbf{x}, z) .
\end{aligned}$$

Hence

$$\begin{aligned}
\mathbb{P}(\hat{y}_i = y_i \mid \mathbf{x}) &= \mathbb{E}_{z|\mathbf{x}}\Big[\mathbb{1}\{\hat{y}_i = 1\}\,\mathbb{P}(y_i = 1 \mid \mathbf{x}, z) + \mathbb{1}\{\hat{y}_i = 0\}\,\mathbb{P}(y_i = 0 \mid \mathbf{x}, z) \mid \mathbf{x}\Big] \\
&\leq \mathbb{E}_{z|\mathbf{x}}\big[\max\{\mathbb{P}(y_i = 1 \mid \mathbf{x}, z), \mathbb{P}(y_i = 0 \mid \mathbf{x}, z)\} \mid \mathbf{x}\big] \\
&= \mathbb{E}_{\mathbf{y} \sim \mathcal{D}_{\mathcal{Y}|\mathbf{x}}}\big[\max\{\mathbb{P}(y_i = 1 \mid \mathbf{x}, \mathcal{M}(\mathbf{x}, \mathbf{y})), \mathbb{P}(y_i = 0 \mid \mathbf{x}, \mathcal{M}(\mathbf{x}, \mathbf{y}))\} \mid \mathbf{x}\big] \\
&= 1 - \mathbb{E}_{\mathbf{y} \sim \mathcal{D}_{\mathcal{Y}|\mathbf{x}}}\big[\min\{\mathbb{P}(y_i = 1 \mid \mathbf{x}, \mathcal{M}(\mathbf{x}, \mathbf{y})), \mathbb{P}(y_i = 0 \mid \mathbf{x}, \mathcal{M}(\mathbf{x}, \mathbf{y}))\} \mid \mathbf{x}\big] \\
&= f_{1i}(\mathbf{x}) ,
\end{aligned}$$

where the inequality holds because the maximum probability term is never smaller than the probability term selected by the indicator variables. Incidentally, the above also shows that the optimal informed adversary is the one that selects the larger of the two probability terms with probability one, which makes the inequality above hold with equality. $\qquad\square$

At this point, we use the fact that

$$\text{EAdv}(\mathcal{M}_{\text{LLP}}, \mathcal{D}) = \mathbb{E}_{\mathbf{x}}[f_{1i}(\mathbf{x}) - f_{2i}(\mathbf{x})]$$

$$\tag{3}$$

holds for all $i \in [m]$ since the data sequence $(x_1, y_1), \ldots, (x_k, y_k)$ is i.i.d.

## A.2 Proof of Theorem 3.2

**Theorem 3.2.** *Fix a data distribution $\mathcal{D}$, let $p = \mathbb{P}_{(x,y) \sim \mathcal{D}}(y = 1)$, and fix an arbitrary threshold $\beta \in [0, 1/2]$. If labels are independent of features (i.e., $\mathcal{D}$ is a product of distributions over $\mathcal{X}$ and $\mathcal{Y}$), then for all bag sizes $k \geq 1$ we have:*

$$\mathrm{EAdv}(\mathcal{M}_{LLP}, \mathcal{D}) = \min\{p, 1-p\} - \mathbb{E}_\alpha[\min\{\alpha, 1-\alpha\}] \leq \begin{cases} \sqrt{\frac{p(1-p)}{k}} & \text{if } p \in [0,1] \\ e^{-\Omega(\beta^2 k)} & \text{if } |p - 1/2| \geq \beta, \end{cases}$$

*where $\Omega(\cdot)$ hides constants independent of $\beta$ and $k$.*

*Proof.* For a given $(x_j, y_j)$ in the dataset, we can consider without loss of generality only the size-$k$ bag that $x_j$ falls into, and then refer the indexing to this bag only.

Then set for brevity $\Sigma = k\alpha = \sum_{i=1}^k y_i$. From (3), we can write

$$\mathrm{EAdv}(\mathcal{M}_{\mathrm{LLP}}, \mathcal{D}) = \mathbb{E}_{\mathbf{x}} \left[ \min\{\eta(x_i), 1 - \eta(x_i)\} \right] - \mathbb{E}_\Sigma \left[ \min_{b \in \{0,1\}} \{ \mathbb{P}(y_i = b \mid \Sigma) \} \right]$$

$$= \min\{p, 1-p\} - \mathbb{E}_\Sigma \left[ \min_{b \in \{0,1\}} \{ \mathbb{P}(y_i = b \mid \Sigma) \} \right].$$

Since $\mathbb{P}(y_i = 1 \mid \Sigma) = \alpha$ independent of $i$ and $p$, the minimum value in the second expectation is

$$\mathbb{E}_\alpha[\min\{\alpha, 1-\alpha\}],$$

and the claimed equality for $\mathrm{EAdv}(\mathcal{M}_{\mathrm{LLP}}, \mathcal{D})$ follows.

As for the inequality with general $p \in [0, 1]$, note that, when $a, b \in [0, 1]$,

$$\min\{a, 1-a\} - \min\{b, 1-b\} \leq |a - b|. \tag{4}$$

If applied to the expression

$$\min\{p, 1-p\} - \mathbb{E}_\alpha[\min\{\alpha, 1-\alpha\}]$$

this gives

$$\mathrm{EAdv}(\mathcal{M}_{\mathrm{LLP}}, \mathcal{D}) \leq \mathbb{E}_\alpha[|\alpha - p|] \leq \sqrt{\mathbb{E}_\alpha[(\alpha - p)^2]} = \sqrt{\frac{p(1-p)}{k}},$$

where the second inequality is Jensen's.

Finally, in the case where $|p - 1/2| \geq \beta$, for some gap $\beta > 0$, we can proceed through a more direct analysis. Assume $p \leq 1/2 - \beta$. We can write

$$\begin{aligned} \min\{\alpha, 1-\alpha\} &= \min\{\alpha, 1-\alpha\}\mathbb{I}\{\alpha \leq 1/2\} + \min\{\alpha, 1-\alpha\}\mathbb{I}\{\alpha > 1/2\} \\ &\quad + \alpha\mathbb{I}\{\alpha > 1/2\} - \alpha\mathbb{I}\{\alpha > 1/2\} \\ &= \alpha\mathbb{I}\{\alpha \leq 1/2\} + (1-\alpha)\mathbb{I}\{\alpha > 1/2\} + \alpha\mathbb{I}\{\alpha > 1/2\} - \alpha\mathbb{I}\{\alpha > 1/2\} \\ &= \alpha - (2\alpha - 1)\mathbb{I}\{\alpha > 1/2\} \\ &\geq \alpha - \mathbb{I}\{\alpha > 1/2\}. \end{aligned}$$

Hence

$$\mathbb{E}_\alpha[\min\{\alpha, 1-\alpha\}] \geq p - \mathbb{P}(\alpha > 1/2).$$

Now, $p < 1/2$ implies $\min\{p, 1-p\} = p$, which leads us to

$$\min\{p, 1-p\} - \mathbb{E}_\alpha[\min\{\alpha, 1-\alpha\}] \leq \mathbb{P}(\alpha > 1/2).$$

Finally, by the standard Bernstein inequality we have

$$\mathbb{P}(\alpha > 1/2) \leq \exp\left(-\frac{k(1/2 - p)^2}{2p(1-p) + (1-2p)/3}\right) = e^{-\Omega(k\beta^2)},$$

which gives the second inequality.

A similar argument holds if we reverse the assumption on $p$ to $p \geq 1/2 + \beta$. $\qquad\square$

## A.3 Proof of Theorem 3.3

Again, as in the proof of Theorem 3.2, for a given $(x_j, y_j)$ in the dataset, we can consider with no loss of generality only the size-$k$ bag that $x_j$ falls into, and then refer the indexing to this bag only.

Recall that, for a general distribution $\mathcal{D}$ over $\mathcal{X} \times \mathcal{Y}$ the distribution of random variable $k\alpha = \sum_{j=1}^{k} y_j$ conditioned on $\mathbf{x} = (x_1, \ldots, x_k)$ is Poisson Binomial (PBin) with parameters $\{\eta(x_j)\}_{j=1}^{k}$, that is, the distribution of the sum of $k$ independent Bernoulli random variables $y_j$, each with its own bias $\eta(x_j)$, where $\eta(x) = \mathbb{P}(y = 1 \mid x)$.

**Notation.** For simplicity of notation we let $\eta_i = \eta(x_i)$ be the conditional positive probability of a label given feature vector $x$ and let $p = \mathbb{E}[\eta_i]$. For fixed feature vectors $\mathbf{x} = (x_1, \ldots, x_k, x_{k+1})$, let $A_i$ denote a Bernoulli random variable with mean $\eta_i$, and let $Z_k = \sum_{j=1}^{k} A_j$

We recall the statement of the theorem we want to prove.

**Theorem 3.3.** *Let $\mathcal{D}$ be an arbitrary distribution on $\mathcal{X} \times \mathcal{Y}$, $p = \mathbb{E}[\eta(x)]$, and $\mu = \mathbb{E}[\eta(x)(1-\eta(x))]$. Then, for all bag sizes $k \geq 2$ we have:*

$$\mathrm{EAdv}(\mathcal{M}_{LLP}, \mathcal{D}) = \widetilde{O}\left( \frac{\mu^{1/4}(p(1-p))^{1/4}}{\sqrt{k}} + \frac{\mu^{1/4}}{k} \right) ,$$

*where $\widetilde{O}$ hides logarithmic factors in $k$.*

The above theorem is a direct consequence of the following Lemma.

**Lemma A.2.** *Let $\mathcal{B}$ denote a bag with $k + 1$ elements. Let*

$$c_k = \frac{1}{\sqrt{k}} \left( \frac{1}{3}\log 8k + \frac{1}{6}\sqrt{2\log 8k + 12kp(1-p)\log 8k} \right) = \widetilde{O}\left( \sqrt{p(1-p)} + \frac{1}{\sqrt{k}} \right) .$$

*Then*

$$\mathrm{EAdv}(\mathcal{M}_{LLP}, \mathcal{D}) \leq k^{1/4}\sqrt{2c_k}\sqrt{\left( \frac{1}{e^{3/2}} + \frac{\pi}{4} + \frac{\pi}{e} \right)\frac{\mathbb{E}[\eta_1(1-\eta_1)]^{1/2}}{k^{3/2}} + \frac{\mathbb{E}[\eta_1(1-\eta_1)]}{k}} .$$

*Proof.* Set for brevity $\mathcal{M} = \mathcal{M}_{LLP}(\mathbf{x}, \mathbf{y})$. From (3) we have

$\mathrm{EAdv}(\mathcal{M}_{LLP}, \mathcal{D})$

$$= \mathbb{E}_{x_{k+1}} \left[ \min\{\eta(x_{k+1}), 1 - \eta(x_{k+1})\} \right] - \mathbb{E}_{(\mathbf{x},\mathbf{y})} \left[ \min\left\{ \mathbb{P}(y_{k+1} = 1|\mathbf{x}, \mathcal{M}), \mathbb{P}(y_{k+1} = 0|\mathbf{x}, \mathcal{M}) \right\} \right]$$

$$\leq \mathbb{E}_{(\mathbf{x},\mathbf{y})} \left[ |\eta(x_{k+1}) - \mathbb{P}(y_{k+1} = 1|\mathbf{x}, \mathcal{M})| \right] ,$$

where we have again used (4). We now focus on calculating $\mathbb{P}(y_{k+1} = 1|\mathbf{x}, \mathcal{M})$. Let $\Sigma = \sum_{j=1}^{k+1} y_j$. Note that for a given realization of feature vector $\mathbf{x}$, $y_{k+1}$ is distributed like $A_{k+1}$ and the output $\mathcal{M}$ is distributed like $Z_{k+1}$. Therefore:

$$\mathbb{P}(y_{k+1} = 1 \mid \mathbf{x}, \mathcal{M}) = \mathbb{P}(A_{k+1} = 1 \mid \mathbf{x}, Z_{k+1} = \Sigma)$$

$$= \frac{\mathbb{P}(A_{k+1} = 1, Z_{k+1} = \Sigma \mid \mathbf{x})}{\mathbb{P}(Z_{k+1} = \Sigma \mid \mathbf{x})}$$

$$= \eta_{k+1} \frac{\mathbb{P}(Z_k = \Sigma - 1 \mid \mathbf{x})}{\mathbb{P}(Z_{k+1} = \Sigma \mid \mathbf{x})} .$$

Using this expression in the original expectation we see that we can bound the advantage as

$$\mathbb{E}_{(\mathbf{x},\mathbf{y})} \left[ \eta_{k+1} \left| 1 - \frac{\mathbb{P}(Z_k = \Sigma - 1 \mid \mathbf{x})}{\mathbb{P}(Z_{k+1} = \Sigma \mid \mathbf{x})} \right| \right]$$

Again, note that for a fixed $\mathbf{x}$, the variable $\Sigma$ is distributed like $Z_{k+1}$. Therefore. taking expectation over $\mathbf{y}$ the above expression can be rewritten as

$$\mathbb{E}_{\mathbf{x}} \left[ \eta_{k+1} \sum_{s=0}^{k+1} \mathbb{P}(Z_{k+1} = s \mid \mathbf{x}) \times \left| 1 - \frac{\mathbb{P}(Z_k = s - 1 \mid \mathbf{x})}{\mathbb{P}(Z_{k+1} = s|\mathbf{x})} \right| \right] =$$

$$\mathbb{E}_{\mathbf{x}} \left[ \eta_{k+1} \sum_{s=0}^{k+1} \left| \mathbb{P}(Z_{k+1} = s \mid \mathbf{x}) - \mathbb{P}(Z_k = s - 1 \mid \mathbf{x}) \right| \right]$$

Finally, note that since $Z_{k+1} = Z_k + A_{k+1}$ we also have
$$\mathbb{P}(Z_{k+1} = s \mid \mathbf{x}) = \mathbb{P}(A_{k+1} = 1 \mid \mathbf{x})\,\mathbb{P}(Z_k = s - 1 \mid \mathbf{x}) + \mathbb{P}(A_{k+1} = 0 \mid \mathbf{x})\,\mathbb{P}(Z_k = s \mid \mathbf{x})$$
$$= \eta_{k+1}\,\mathbb{P}(Z_k = s - 1 \mid \mathbf{x}) + (1 - \eta_{k+1})\,\mathbb{P}(Z_k = s \mid \mathbf{x}) .$$

Therefore, we conclude that the advantage can be bounded by
$$\mathbb{E}_{\mathbf{x}}\left[\eta_{k+1}(1 - \eta_{k+1})\sum_{s=0}^{k+1}\Big|\mathbb{P}(Z_k = s \mid \mathbf{x}) - \mathbb{P}(Z_k = s - 1 \mid \mathbf{x})\Big|\right] =$$
$$\mathbb{E}_{\mathbf{x}}\left[\eta_{k+1}(1 - \eta_{k+1})\right]\,\mathbb{E}\left[\sum_{s=0}^{k+1}\Big|\mathbb{P}(Z_k = s \mid \mathbf{x}) - \mathbb{P}(Z_k = s - 1 \mid \mathbf{x})\Big|\right] , \tag{5}$$

where we have used the fact that the random variables $\eta_i$ are independent from each other. Using also the fact that $\eta_{k+1}$ has the same distribution as $\eta_1$ combined with Lemma A.3 below, we have that the above quantity is bounded by:
$$k^{1/4}\sqrt{2c_k}\sqrt{\mathbb{E}[\eta_1(1 - \eta_1)]^2\,\mathbb{E}[\eta_1^2 + (1 - \eta_1)^2]^k + \frac{\pi\,\mathbb{E}[\eta_1(1 - \eta_1)]^{1/2}}{4k^{3/2}} + \frac{\pi\,\mathbb{E}[\eta_1(1 - \eta_1)]}{ek^2}}$$
$$+ \frac{\mathbb{E}[\eta_1(1 - \eta_1)]}{k} . \tag{6}$$

Moreover, notice that $\mathbb{E}[\eta_1^2 + (1 - \eta_1)^2] + 2\,\mathbb{E}[\eta_1(1 - \eta_1)] = 1$. Therefore $\mathbb{E}[\eta_1^2 + (1 - \eta_1)^2] = 1 - 2\,\mathbb{E}[\eta_1(1 - \eta_1)]$, and using the fact that $\eta_1(1 - \eta_1) \leq \frac{1}{4}$ we have
$$\mathbb{E}[\eta_1(1 - \eta_1)]^2\,\mathbb{E}[\eta_1^2 + (1 - \eta_1)^2]^k = \mathbb{E}[\eta_1(1 - \eta_1)]^2(1 - 2\,\mathbb{E}[\eta_1(1 - \eta_1)])^k$$
$$\leq \max_{\frac{1}{4} \geq x \geq 0} x^2(1 - 2x)^k .$$

But a simple calculation shows that the above function is maximized at $x_k^\star = \min\{\frac{1}{k+2}, 1/4\}$, thus we must have
$$\mathbb{E}[\eta_1(1 - \eta_1)]^2\,\mathbb{E}[\eta_1^2 + (1 - \eta_1)^2]^k \leq (x_k^\star)^2(1 - 2x_k^\star)^k \leq \frac{1}{(ek)^2}$$

the last inequality holding for all $k \geq 1$. In addition, we have the trivial bound $\mathbb{E}[\eta_1(1 - \eta_1)]^2\,\mathbb{E}[\eta_1^2 + (1 - \eta_1)^2] \leq \mathbb{E}[\eta_1(1 - \eta_1)]^2$, so that
$$\mathbb{E}[\eta_1(1 - \eta_1)]^2\,\mathbb{E}[\eta_1^2 + (1 - \eta_1)^2]^k \leq \min\left\{\mathbb{E}[\eta_1(1 - \eta_1)]^2, \frac{1}{e^2k^2}\right\}$$
$$\leq \frac{\mathbb{E}[\eta_1(1 - \eta_1)]^2}{\mathbb{E}[\eta_1(1 - \eta_1)]^2\,e^2\,k^2 + 1}$$
$$\text{(using } \min\{a, b\} \leq \frac{ab}{a+b}, \text{ with } a = \mathbb{E}[\eta_1(1 - \eta_1)]^2 \text{ and } b = \frac{1}{e^2k^2})$$
$$\leq \frac{\sqrt{\mathbb{E}[\eta_1(1 - \eta_1)]}}{e^{3/2}k^{3/2}}$$
$$\text{(using } x^2 - x^{3/2} + 1 \geq 0, \text{ with } x = ek\,\mathbb{E}[\eta_1(1 - \eta_1)]) .$$

Replacing this bound in (6) we obtain the following upper bound on the advantage:
$$k^{1/4}\sqrt{2c_k}\sqrt{\left(\frac{1}{e^{3/2}} + \frac{\pi}{4}\right)\frac{\mathbb{E}[\eta_1(1 - \eta_1)]^{1/2}}{k^{3/2}} + \frac{\pi\,\mathbb{E}[\eta_1(1 - \eta_1)]}{ek^2}} + \frac{\mathbb{E}[\eta_1(1 - \eta_1)]}{k}$$
$$\leq k^{1/4}\sqrt{2c_k}\sqrt{\left(\frac{1}{e^{3/2}} + \frac{\pi}{4} + \frac{\pi}{e}\right)\frac{\mathbb{E}[\eta_1(1 - \eta_1)]^{1/2}}{k^{3/2}}} + \frac{\mathbb{E}[\eta_1(1 - \eta_1)]}{k} ,$$

as claimed. $\qquad\square$

**Lemma A.3.** *Let $c_k$ be as in Lemma A.2. Then the following bound holds:*
$$\mathbb{E}_{\mathbf{x}}\left[\sum_{s=0}^{k+1}|\mathbb{P}(Z_k = s \mid \mathbf{x}) - \mathbb{P}(Z_k = s - 1 \mid \mathbf{x})|\right] \leq$$
$$\frac{1}{k} + k^{1/4}\sqrt{2c_k}\sqrt{\mathbb{E}[\eta_1^2 + (1 - \eta_1)^2]^k + \frac{\pi}{(4\,\mathbb{E}[\eta_1(1 - \eta)_1)]k)^{3/2}} + \frac{\pi}{e\,\mathbb{E}[\eta_1(1 - \eta_1)]k^2}}$$

*Proof.* Let $a > 0$ and $b < k$, and let $[a, b] = \{j \in \mathbb{N} | a \le j \le b\}$. For any $\mathbf{x}$ we then have

$$\sum_{s=0}^{k+1} |\mathbb{P}(Z_k = s \,|\, \mathbf{x}) - \mathbb{P}(Z_k = s - 1 \,|\, \mathbf{x})|$$

$$= \sum_{s \in [a,b]} |\mathbb{P}(Z_k = s \,|\, \mathbf{x}) - \mathbb{P}(Z_k = s - 1 \,|\, \mathbf{x})| + \sum_{s \notin [a,b]} |\mathbb{P}(Z_k = s \,|\, \mathbf{x}) - \mathbb{P}(Z_k = s - 1)|$$

$$\le \sum_{s \in [a,b]} |\mathbb{P}(Z_k = s \,|\, \mathbf{x}) - \mathbb{P}(Z_k = s - 1 \,|\, \mathbf{x})| + \sum_{s \notin [a,b]} \mathbb{P}(Z_k = s \,|\, \mathbf{x}) + \mathbb{P}(Z_k = s - 1 \,|\, \mathbf{x})$$

$$\le \sum_{s \in [a,b]} |\mathbb{P}(Z_k = s \,|\, \mathbf{x}) - \mathbb{P}(Z_k = s - 1 \,|\, \mathbf{x})| + 2\,\mathbb{P}(Z_k \notin [a, b] \,|\, \mathbf{x})$$

Taking expectation over both sides with respect to $\mathbf{x}$ we have

$$\mathbb{E}_{\mathbf{x}}\left[\sum_{s=0}^{k+1} |\mathbb{P}(Z_k = s \,|\, \mathbf{x}) - \mathbb{P}(Z_k = s - 1 \,|\, \mathbf{x})|\right]$$

$$\le \mathbb{E}_{\mathbf{x}}\left[\sum_{s \in [a,b]} |\mathbb{P}(Z_k = s \,|\, \mathbf{x}) - \mathbb{P}(Z_k = s - 1 \,|\, \mathbf{x})|\right] + 2\mathbb{E}_{\mathbf{x}}[\mathbb{P}(Z_k \notin [a, b] \,|\, \mathbf{x})] \qquad (7)$$

Let now $Q_k$ denote the probability measure associated with a binomial random variable with parameters $(k, p)$. Since $Z_k$ is a Poisson-Binomial random variable with parameters $\eta_1, \dots, \eta_k$, the probability $\mathbb{P}(Z_k \notin [a, b] \,|\, \mathbf{x})$ is a *linear* function in each individual $\eta_i$, so that $\mathbb{E}_{\mathbf{x}}[\mathbb{P}(Z_k \notin [a, b] \,|\, \mathbf{x})] = Q_k([a, b]^c)$. We now proceed to bound the first expectation in (7). By Cauchy-Schwartz inequality we have

$$\mathbb{E}_{\mathbf{x}}\left[\sum_{s \in [a,b]} |\mathbb{P}(Z_k = s \,|\, \mathbf{x}) - \mathbb{P}(Z_k = s - 1 \,|\, \mathbf{x})|\right]$$

$$\le \mathbb{E}_{\mathbf{x}}\left[\sqrt{(b - a) \sum_{s \in [a,b]} (\mathbb{P}(Z_k = s \,|\, \mathbf{x}) - \mathbb{P}(Z_k = s - 1 \,|\, \mathbf{x}))^2}\right]$$

$$\le \mathbb{E}_{\mathbf{x}}\left[\sqrt{(b - a) \sum_{s=0}^{k+1} (\mathbb{P}(Z_k = s \,|\, \mathbf{x}) - \mathbb{P}(Z_k = s - 1 \,|\, \mathbf{x}))^2}\right]$$

$$\le \sqrt{\mathbb{E}_{\mathbf{x}}\left[(b - a) \sum_{s=0}^{k+1} (\mathbb{P}(Z_k = s \,|\, \mathbf{x}) - \mathbb{P}(Z_k = s - 1 \,|\, \mathbf{x}))^2\right]},$$

where the last inequality holds by Jensen's inequality. Let

$$A = \mathbb{E}[\eta_1^2 + (1 - \eta_1)^2] \qquad \text{and} \qquad B = 2\,\mathbb{E}[\eta_1(1 - \eta_1)].$$

By Lemma A.4 below we have that the above term is bounded by

$$\sqrt{(b - a)\left(A^k + \frac{\pi}{(4Bk)^{3/2}} + \frac{\pi}{eBk^2}\right)},$$

so that (7) gives

$$\mathbb{E}_{\mathbf{x}}\left[\sum_{s=0}^{k+1} |\mathbb{P}(Z_k = s \,|\, \mathbf{x}) - \mathbb{P}(Z_k = s - 1 \,|\, \mathbf{x})|\right] \le 2Q_k([a, b]^c) + \sqrt{(b - a)\left(A^k + \frac{\pi}{(4Bk)^{3/2}} + \frac{\pi}{eBk^2}\right)}.$$

Let $a = \max\{kp - \sqrt{k}c_k, 0\}$ and $b = \min\{kp + \sqrt{k}c_k, 1\}$. By Bernstein's inequality applied to binomial random variables we have that $Q_k([a, b]^c) = \frac{1}{2k}$. Hence, with this choice of $a$ and $b$ we

obtain

$$\mathbb{E}_{\mathbf{x}}\left[\sum_{s=0}^{k+1}|\mathbb{P}(Z_k = s \mid \mathbf{x}) - \mathbb{P}(Z_k = s - 1 \mid \mathbf{x})|\right] \leq \frac{1}{k} + \sqrt{2\sqrt{k}c_k}\sqrt{A^k + \frac{\pi}{(4Bk)^{3/2}} + \frac{\pi}{eBk^2}} \ .$$

The lemma follows by replacing the values of $A$ and $B$. $\qquad\square$

**Lemma A.4.** *The following inequality holds*

$$\mathbb{E}_{\mathbf{x}}\left[\sum_{s=0}^{k+1}(\mathbb{P}(Z_k = s \mid \mathbf{x}) - \mathbb{P}(Z_k = s - 1 \mid \mathbf{x}))^2\right]$$

$$\leq \mathbb{E}[\eta_1^2 + (1-\eta_1)^2]^k + \frac{\pi}{(8\,\mathbb{E}[\eta_1(1-\eta_1)]k)^{3/2}} + \frac{\pi}{2e\,\mathbb{E}[\eta_1(1-\eta_1)]k^2} \ .$$

*Proof.* Let $p_s(\mathbf{x}) = \mathbb{P}(Z_k = s \mid \mathbf{x}) - \mathbb{P}(Z_k = s - 1 \mid \mathbf{x})$, for $s = 0, \ldots, k+1$. Further, for $k + 1 \geq u \geq 0$ let $g_u(\mathbf{x}) = \frac{1}{\sqrt{k+2}}\sum_{s=0}^{k+1} p_s(\mathbf{x})e^{2\pi i \frac{us}{k+2}}$ denote the discrete Fourier transform. Since, for any $\mathbf{x}$, the mapping

$$\mathbf{p}(\mathbf{x}) := (p_0(\mathbf{x}), \ldots, p_{k+1}(\mathbf{x})) \mapsto (g_1(\mathbf{x}), \ldots, g_{k+1}(\mathbf{x}) := \mathbf{g}(\mathbf{x})$$

is a unitary linear transformation [8] we can write:

$$\sum_{s=0}^{k+1}(\mathbb{P}(Z_k = s \mid \mathbf{x}) - \mathbb{P}(Z_k = s - 1 \mid \mathbf{x}))^2 = \|\mathbf{p}(\mathbf{x})\|^2 = \|\mathbf{g}(\mathbf{x})\|^2 = \sum_{u=0}^{k+1}|g_u(\mathbf{x})|^2.$$

Moreover, by Lemma A.5 below we have that

$$g_u(\mathbf{x}) = (1 - e^{\frac{2\pi i u}{k+2}})\frac{1}{\sqrt{k+2}}\prod_{j=1}^{k}(1 - \eta_j + \eta_j e^{\frac{2\pi i u}{k+2}})$$

and therefore

$$|g_u(\mathbf{x})|^2 = g_u(\mathbf{x})\overline{g_u(\mathbf{x})} = \frac{1}{k+2}\left(1 - \cos\frac{2\pi u}{k+2}\right)\prod_{j=1}^{k}\left((1-\eta_j)^2 + \eta_j^2 + 2\eta_j(1-\eta_j)\cos\frac{2\pi u}{k+2}\right) \ .$$

Therefore we can write

$$\mathbb{E}_{\mathbf{x}}\left[\sum_{s=0}^{k+1}(\mathbb{P}(Z_k = s \mid \mathbf{x}) - \mathbb{P}(Z_k = s - 1 \mid \mathbf{x}))^2\right]$$

$$= \frac{1}{k+2}\mathbb{E}_{\mathbf{x}}\left[\sum_{u=0}^{k+1}\left(1 - \cos\frac{2\pi u}{k+2}\right)\prod_{j=1}^{k}\left((1-\eta_j)^2 + \eta_j^2 + 2\eta_j(1-\eta_j)\cos\frac{2\pi u}{k+2}\right)\right]$$

$$= \frac{1}{k+2}\sum_{u=0}^{k+1}\left(1 - \cos\frac{2\pi u}{k+2}\right)\prod_{j=1}^{k}\mathbb{E}_{\mathbf{x}}\left[\left((1-\eta_j)^2 + \eta_j^2 + 2\eta_j(1-\eta_j)\cos\frac{2\pi u}{k+2}\right)\right], \qquad (8)$$

Where we have used the fact that the random variables $\eta_j$ are independent. Finally by linearity of expectation and the fact that $\eta_j$ is distributed as $\eta_1$ for all $j$, we have

$$\mathbb{E}_{\mathbf{x}}\left[\sum_{s=0}^{k+1}(\mathbb{P}(Z_k = s \mid \mathbf{x}) - \mathbb{P}(Z_k = s - 1 \mid \mathbf{x}))^2\right]$$

$$\frac{1}{k+2}\sum_{u=0}^{k+1}\left(1 - \cos\frac{2\pi u}{k+2}\right)\left(\mathbb{E}[\eta_1^2 + (1-\eta_1)^2] + 2\,\mathbb{E}[\eta_1(1-\eta_1)]\cos\frac{2\pi u}{k+2}\right)^k \ . \qquad (9)$$

Applying Proposition A.6 below with $a = \mathbb{E}[\eta_1^2 + (1-\eta_1)^2]$ and $b = 2\,\mathbb{E}[\eta_1(1-\eta_1)]$ we have that the above expression is bounded by

$$\mathbb{E}[\eta_1^2 + (1-\eta_1)^2]^k + \frac{\pi}{(8\,\mathbb{E}[\eta_1(1-\eta_1)]k)^{3/2}} + \frac{\pi}{2e\,\mathbb{E}[\eta_1(1-\eta_1)]k^2}$$

which gives the claimed result. $\qquad\square$

**Lemma A.5.** *Let $p_s(\mathbf{x}) = \mathbb{P}(Z_k = s \mid \mathbf{x}) - \mathbb{P}(Z_k = s - 1 \mid \mathbf{x})$ and let $g_u(\mathbf{x}) = \frac{1}{\sqrt{k+2}} \sum_{s=0}^{k+1} p_s(\mathbf{x}) e^{\frac{2\pi i u s}{k+2}}$ denote the discrete Fourier transform of $\mathbf{p}(\mathbf{x}) = (p_1(\mathbf{x}), \ldots, p_s(\mathbf{x}))$. Then*

$$g_u(\mathbf{x}) = \frac{1}{\sqrt{k+2}}(1 - e^{\frac{2\pi i u}{k+2}}) \prod_{j=1}^{k}(1 - \eta_j + \eta_j e^{\frac{2\pi i u}{k+2}})$$

*Proof.* By definition of $g_u(\mathbf{x})$ we have:

$$\frac{1}{\sqrt{k+2}}\left(\sum_{s=0}^{k+1} e^{\frac{2\pi i u s}{k+2}} \mathbb{P}(Z_k = s \mid \mathbf{x}) - \sum_{s=0}^{k+1} e^{\frac{2\pi i u s}{k+2}} \mathbb{P}(Z_k = s - 1 \mid \mathbf{x})\right)$$

$$= \frac{1}{\sqrt{k+2}}\left(\sum_{s=0}^{k+1} e^{\frac{2\pi i u s}{k+2}} \mathbb{P}(Z_k = s \mid \mathbf{x}) - e^{\frac{2\pi i u}{k+2}} \sum_{s=0}^{k+1} e^{\frac{2\pi i u(s-1)}{k+2}} \mathbb{P}(Z_k = s - 1 \mid \mathbf{x})\right)$$

$$= \frac{1}{\sqrt{k+2}}(1 - e^{\frac{2\pi i u}{k+2}})\mathbb{E}_{Z_k}[e^{\frac{2\pi i u}{k+2} Z_k} \mid \mathbf{x}]$$

$$= \frac{1}{\sqrt{k+2}}(1 - e^{\frac{2\pi i u}{k+2}})\phi_{Z_k \mid \mathbf{x}}\left(\frac{2\pi u}{k+2}\right)$$

where $\phi_{Z_k \mid \mathbf{x}}$ denotes the characteristic function of $Z_k$ conditioned on $\mathbf{x}$. Using the fact that $Z_k = \sum_{j=1}^{k} A_j$ and that $A_1, \ldots, A_k$ are independent given $\mathbf{x}$, we have $\phi_{Z_k \mid \mathbf{x}} = \prod_{j=1}^{k} \phi_{A_j \mid x_j}$. The result follows from the fact that $A_j$ is a Bernoulli random variable and therefore $\phi_{A_j \mid x_j}(z) = (1 - \eta_j + \eta_j e^{iz})$. $\qquad\square$

**Proposition A.6.** *For any $a, b, k > 0$ such that $a + b = 1$ we have*

$$\frac{1}{k+2} \sum_{u=0}^{k+1} \left(1 - \cos\frac{2\pi u}{k+2}\right)\left(a + b\cos\frac{2\pi u}{k+2}\right)^k \le a^k + \frac{\pi}{(4kb)^{3/2}} + \frac{\pi}{ebk^2}$$

*Proof.* Using the fact that $\cos\frac{2\pi u}{k+2} \le 0$ for $u \in [(k+2)/4, 3(k+2)/4]$ we have that

$$\sum_{u=0}^{k+1} \left(1 - \cos\frac{2\pi u}{k+2}\right)\left(a + b\cos\frac{2\pi u}{k+2}\right)^k = \sum_{u=0}^{k+2/4} \left(1 - \cos\frac{2\pi u}{k+2}\right)\left(a + b\cos\frac{2\pi u}{k+2}\right)^k$$

$$+ \sum_{u=(k+2)/4+1}^{3(k+2)/4} \left(1 - \cos\frac{2\pi u}{k+2}\right)\left(a + b\cos\frac{2\pi u}{k+2}\right)^k$$

$$+ \sum_{u=3(k+2)/4+1}^{k+1} \left(1 - \cos\frac{2\pi u}{k+2}\right)\left(a + b\cos\frac{2\pi u}{k+2}\right)^k$$

$$\le (k+2)a^k + \sum_{u=0}^{(k+2)/4} \left(1 - \cos\frac{2\pi u}{k+2}\right)\left(a + b\cos\frac{2\pi u}{k+2}\right)^k$$

$$+ \sum_{u=3(k+2)/4+1}^{k+1} \left(1 - \cos\frac{2\pi u}{k+2}\right)\left(a + b\cos\frac{2\pi u}{k+2}\right)^k$$

$$= (k+2)a^k + 2\sum_{u=0}^{(k+2)/4} \left(1 - \cos\frac{2\pi u}{k+2}\right)\left(a + b\cos\frac{2\pi u}{k+2}\right)^k,$$

where we used the fact that $(1 - \cos t) \le 2$ for the first inequality and the symmetry of the cosine function for the last equality. We now apply the result of Proposition A.8 to the above expression to

see that

$$\sum_{u=0}^{(k+2)/4} \left(1 - \cos\frac{2\pi u}{k+2}\right)\left(a + b\cos\frac{2\pi u}{k+2}\right)^k \leq 2\pi^2 \sum_{u=0}^{(k+2)/4} \frac{u^2}{(k+2)^2}\left(1 - 4\pi b\frac{u^2}{(k+2)^2}\right)^k$$

$$\leq 2\pi^2 \sum_{u=0}^{(k+2)/4} \frac{u^2}{(k+2)^2}e^{-4\pi kb\frac{u^2}{(k+2)^2}}$$

Therefore we conclude that

$$\frac{1}{k+2}\sum_{u=0}^{k+1}\left(1 - \cos\frac{2\pi u}{k+2}\right)\left(a + b\cos\frac{2\pi u}{k+2}\right)^k \leq a^k + 4\pi^2 \sum_{u=0}^{(k+2)/4} \frac{u^2}{(k+2)^2}e^{-4\pi kb\frac{u^2}{(k+2)^2}}\frac{1}{k+2}.$$

Finally, applying Proposition A.7 with $m = k + 2$ and $\alpha = 4\pi kb$ we can upper bound the above quantity by:

$$a^k + 4\pi^2\left(\frac{\sqrt{\pi}}{4(4\pi kb)^{3/2}} + \frac{1}{4e\pi bk(k+2)}\right) \leq a^k + \frac{\pi}{(4kb)^{3/2}} + \frac{\pi}{ebk^2},$$

which concludes the proof. $\qquad\square$

**Proposition A.7.** *Let $m > 0$ and $\alpha > 0$. Then*

$$\sum_{u=0}^{m/4} \frac{u^2}{m^2}e^{-\alpha\frac{u^2}{m^2}}\frac{1}{m} \leq \frac{\sqrt{\pi}}{4\alpha^{3/2}} + \frac{1}{e\alpha m}$$

*Proof.* Let $f\colon \mathbb{R} \to \mathbb{R}$ be given by $x \mapsto x^2 e^{-\alpha x^2}$. Note that the sum we are attempting to bound is then given by:

$$\sum_{u=0}^{m/4} f\left(\frac{u}{m}\right)\frac{1}{m}$$

Note also that $f$ has a maximum at $x_0 = \frac{1}{\sqrt{\alpha}}$. Thus $f$ is increasing for $x < x_0$ and decreasing otherwise. In particular if

$$\sum_{u=0}^{m/4} f\left(\frac{u}{m}\right)\frac{1}{m} = \sum_{u=0}^{\lfloor x_0\rfloor} f\left(\frac{u}{m}\right)\frac{1}{m} + \sum_{u=\lceil x_0\rceil}^{m/4} f\left(\frac{u}{m}\right)\frac{1}{m} := L + U,$$

then $L$ corresponds to a lower Riemman sum for $f$ and $L \leq \int_0^{\frac{\lfloor x_0\rfloor+1}{m}} f(x)dx$. Similarly $U$ is an upper Riemman sum for $f$ and $U \leq \int_{\frac{\lceil x_0\rceil-1}{m}}^{1/4-\frac{1}{m}} f(x)dx$. Therefore we have

$$\sum_{u=0}^{m/4} f\left(\frac{u}{m}\right)\frac{1}{m} \leq \int_0^{1/4-1/m} x^2 e^{-\alpha x^2}dx + \int_{\frac{\lceil x_0\rceil-1}{m}}^{\frac{\lfloor x_0\rfloor+1}{m}} f(x)dx$$

$$\leq \int_0^\infty x^2 e^{-\alpha x^2} + \frac{1}{m}\max_x f(x)$$

$$= \frac{\sqrt{\pi}}{4\alpha^{3/2}} + \frac{1}{e\alpha m},$$

as claimed. $\qquad\square$

**Proposition A.8.** *The following inequality holds for any $t \in [0, 1/4]$:*

$$1 - 2(\pi t)^2 \leq \cos 2\pi t \leq 1 - 4\pi t^2$$

*Proof.* For the lower bound we start from the fact that for any $x \geq 0$ it is well known that

$$\sin 2\pi x \leq 2\pi x.$$

Integrating this inequality from $[0, t]$ we have that $\int_0^t \sin 2\pi x \leq \pi t^2$. Since $\int_0^t \sin 2\pi x = \frac{1}{2\pi}(1 - \cos 2\pi t)$ the lower bound follows.

For the upper bound we proceed in a similar fashion. By the fact that $\sin 2\pi x$ is concave for $x \in [0, 1/4]$ we have that

$$\sin 2\pi x = \sin 2\pi((1 - 4x) \cdot 0 + 4x \cdot \frac{1}{4}) \geq (1 - 4x)\sin 0 + 4x \sin \frac{\pi}{2} = 4x.$$

Again integrating the above inequality from $0$ to $t$ we have $\frac{1 - \cos 2\pi t}{2\pi} \geq 2t^2$. $\qquad\square$

### A.4 Proof of Theorem 3.4

*Proof.* The proof follows the same argument of the proof used in 3.3, where we see that, in view of the factorization in Equation 5 we may bound the individual expected advantage as:

$$\text{IEAdv}(\mathcal{M}, \mathcal{D}, x_{k+1}) \leq \eta(x_{k+1})(1 - \eta(x_{k+1}) \underset{x_1,\ldots,x_k}{\mathbb{E}} \left[ \sum_{s=0}^{k} |P(Z_k = s|\mathbf{x}) - P(Z_k = s - 1|\mathbf{x})| \right].$$

But from Lemma A.3 it follows that this can be bounded by

$$\mu_i \, O \left( (p(1-p))^{1/4} \sqrt{(1 - 2\mu)^k k^{1/2} + \frac{1}{\mu^{3/2}k}} + \sqrt{\frac{(1 - 2\mu)^k}{k^{1/2}} + \frac{1}{\mu^{3/2}k^2}} \right).$$

Now, since $(1 - 2\mu)^k k^{1/2} < \frac{\mu}{k}$ for $k \geq \frac{2}{\mu} \log\left(\frac{1}{\mu}\right)$, we upper bound accordingly the two terms involving $(1 - 2\mu)^k$, and the result follows. $\qquad\square$

### A.5 Tail distribution of the attack advatange

We are also interested in the *tail* of the distribution of the attack advantage, that is, the tail of the difference between

$$f_1(\mathbf{x}) = \sup_{\mathcal{A}_{\text{informed}}} \underset{\substack{\mathbf{y} \sim \mathcal{D}_{\mathcal{Y}|\mathbf{x}}^m \\ i \sim \text{Uniform}([m]) \\ \text{coins of } \mathcal{M}}}{\mathbb{P}} (\mathcal{A}_{\text{informed}}(\mathbf{x}, \mathcal{M}(\mathbf{x}, \mathbf{y}))_i = y_i \mid \mathbf{x})$$

and

$$f_2(\mathbf{x}) = \sup_{\mathcal{A}_{\text{uninformed}}} \underset{\substack{\mathbf{y} \sim \mathcal{D}_{\mathcal{Y}|\mathbf{x}}^m \\ i \sim \text{Uniform}([m]) \\ \text{coins of } \mathcal{M}}}{\mathbb{P}} (\mathcal{A}_{\text{uninformed}}(\mathbf{x}, \mathcal{M}_{\perp}(\mathbf{x}, \mathbf{y}))_i = y_i \mid \mathbf{x})$$

Specifically, we define

$$\text{HPAdv}(\mathcal{M}, \mathcal{D}, \theta) = \underset{\mathbf{x} \sim \mathcal{D}_{\mathcal{X}}^m}{\mathbb{P}} (f_1(\mathbf{x}) - f_2(\mathbf{x})) > \theta) \,,$$

viewed as a function of $\theta \in [0, 1]$, which we call the *High Probability attack Advantage* (at level $\theta$). This is simply a high-probability version of $\text{EAdv}(\mathcal{A}, \mathcal{M}, \mathcal{D})$ over the generation of $\mathbf{x}$. A similar high-probability version can be defined for the individual version of the attack advantage.

We begin by observing that in the special case of $\mathcal{M}_{\text{RR}}$, we have $\text{IEAdv}_1(\mathcal{M}_{\text{RR}}, \mathcal{D}, x_1) = f_1(\mathbf{x}) - f_2(\mathbf{x})$, so that the high probability bound involved in the computation of $\text{HPAdv}(\mathcal{M}_{\text{RR}}, \mathcal{D}, \theta)$ is a direct consequence of the tail properties of the function $\eta(x_1) = \mathbb{P}(y_1 = 1 \mid x_1)$ as applied to the expression for $\text{IEAdv}_{1,1}(\mathcal{M}_{\text{RR}}, \mathcal{D}, x_1)$ in Theorem A.17. Thus, for $\mathcal{M}_{\text{RR}}$, the connection from IEAdv to HPAdv is immediate.

We continue with studying HPAdv for $\mathcal{M}_{\text{LLP}}$. We have the following bound.

**Theorem A.9.** *Under the same assumptions and notation as in Theorem 3.3 (main body) there are universal constants $c_1, c_2 > 0$ such that, for $k = \Omega(1/\mu)$,*

$$\text{HPAdv}\left(\mathcal{M}_{LLP}, \mathcal{D}, c_1\sqrt{\frac{\mu}{k}}\right) \leq (k+1)\, e^{-c_2\, k\mu}\,.$$

In order to prove this theorem, we first need ancillary results related to Poisson Binomial distributions.

We denote a Poisson Binomial distribution with parameters $\{\eta_1, \ldots, \eta_k\}$ by $\text{PBin}(\eta_1, \ldots, \eta_k)$.

**Lemma A.10.** *Let $s, \Sigma \sim PBin(\eta_1, \ldots, \eta_k)$, $\Sigma^{(1)} \sim PBin(\eta_2, \ldots, \eta_k)$, where $\eta_1, \ldots, \eta_k \in [0,1]$ are such that*

$$\sum_{i=1}^{k} \eta_i(1-\eta_i) \geq 1\,.$$

*Then*

$$\mathbb{E}_s\left[\left|\frac{\mathbb{P}(\Sigma^{(1)} = s) - \mathbb{P}(\Sigma^{(1)} = s-1)}{\mathbb{P}(\Sigma = s)}\right|\right] \leq \frac{9}{\sqrt{\sum_{i=2}^{k} \eta_i(1-\eta_i)}}\,.$$

*Proof.* We will use the following distribution to approximate Poisson Binomial distributions.

**Definition A.11.** *[28] We say that an integer random variable $Y$ is distributed according to the translated Poisson distribution with parameters $\mu$ and $\sigma^2$, denoted $TP(\mu, \sigma^2)$, if and only if $Y$ can be written as*

$$Y = \lfloor \mu + \sigma^2 \rfloor + Z,$$

*where $Z$ is a random variable distributed according to Poisson$(\sigma^2 + \{\mu - \sigma^2\})$, being $\{\mu - \sigma^2\}$ the fractional part of $\mu - \sigma^2$, and $\lfloor \mu + \sigma^2 \rfloor$ the integer part of $\mu + \sigma^2$.*

Let us then define two random variables $\Gamma$ and $\Gamma^{(1)}$. Variable $\Gamma$ is $\text{TP}(\mu, \sigma^2)$ with $\mu = \sum_{i=1}^{k} \eta_i$ and $\sigma^2 = \sum_{i=1}^{k} \eta_i(1-\eta_i)$, while $\Gamma^{(1)}$ is $\text{TP}(\mu_1, \sigma_1^2)$, where $\mu_1 = \sum_{i=2}^{k} \eta_i$ and $\sigma_1^2 = \sum_{i=2}^{k} \eta_i(1-\eta_i)$.

Let $d_{\text{TV}}$ denote the total variation distance. We can write

$$\mathbb{E}_s\left[\left|\frac{\mathbb{P}(\Sigma^{(1)} = s) - \mathbb{P}(\Sigma^{(1)} = s-1)}{\mathbb{P}(\Sigma = s)}\right|\right] = \sum_{s=0}^{k} \mathbb{P}(\Sigma = s) \times \left[\left|\frac{\mathbb{P}(\Sigma^{(1)} = s) - \mathbb{P}(\Sigma^{(1)} = s-1)}{\mathbb{P}(\Sigma = s)}\right|\right]$$

$$= \sum_{s=0}^{k} \left|\mathbb{P}(\Sigma^{(1)} = s) - \mathbb{P}(\Sigma^{(1)} = s-1)\right|$$

$$\leq \sum_{s=0}^{k} \left|\mathbb{P}(\Sigma^{(1)} = s) - \mathbb{P}(\Gamma = s)\right| + \sum_{s=0}^{k} \left|\mathbb{P}(\Gamma = s) - \mathbb{P}(\Gamma^{(1)} = s-1)\right|$$

$$+ \sum_{s=0}^{k} \left|\mathbb{P}(\Gamma^{(1)} = s-1) - \mathbb{P}(\Sigma^{(1)} = s-1)\right| \qquad (10)$$

$$\leq d_{\text{TV}}(\Sigma, \Gamma) + \sum_{s=0}^{k} \left|\mathbb{P}(\Gamma = s) - \mathbb{P}(\Gamma^{(1)} = s-1)\right| + d_{\text{TV}}(\Sigma^{(1)}, \Gamma^{(1)})$$

$$\leq \frac{\sqrt{\sum_{i=1}^{k} \eta_i^3(1-\eta_i)} + 2}{\sum_{i=1}^{k} \eta_i(1-\eta_i)} + \frac{\sqrt{\sum_{i=2}^{k} \eta_i^3(1-\eta_i)} + 2}{\sum_{i=2}^{k} \eta_i(1-\eta_i)}$$

$$(11)$$

$$+ \sum_{s=0}^{k} \left|\mathbb{P}(\Gamma = s) - \mathbb{P}(\Gamma^{(1)} = s-1)\right|$$

$$\leq \frac{3}{\sqrt{\sigma}} + \frac{3}{\sqrt{\sigma_1}} + \sum_{s=0}^{k} \left|\mathbb{P}(\Gamma = s) - \mathbb{P}(\Gamma^{(1)} = s-1)\right|,$$

$$(12)$$

where (10) follows from the triangle inequality, (11) follows from Lemma A.12 below, and (12) follows from the overapproximation

$$\frac{\sqrt{\sum_{i=1}^{k} \eta_i^3(1-\eta_i)} + 2}{\sum_{i=1}^{k} \eta_i(1-\eta_i)} \leq \frac{\sqrt{\sum_{i=1}^{k} \eta_i(1-\eta_i)} + 2}{\sum_{i=1}^{k} \eta_i(1-\eta_i)} \leq \frac{3}{\sqrt{\sum_{i=1}^{k} \eta_i(1-\eta_i)}} ,$$

the latter inequality exploiting the condition $\sum_{i=1}^{k} \eta_i(1-\eta_i) \geq 1$.

**Lemma A.12.** *[[28], see (3.4) therein] Let $J_1, \ldots, J_k$ be independent random Bernoulli with parameters $p_1, \ldots, p_k$. Then*

$$d_{TV}\left(\sum_{i=1}^{k} J_i, TP(\mu, \sigma^2)\right) \leq \frac{\sqrt{\sum_{i=1}^{k} p_i^3(1-p_i)} + 2}{\sum_{i=1}^{k} p_i(1-p_i)}$$

*where $\mu = \sum_{i=1}^{k} p_i$ and $\sigma^2 = \sum_{i=1}^{k} p_i(1-p_i)$.*

We recall a lemma which will be useful to upper bound the last term in (12).

**Lemma A.13.** *[6] For $\mu_1, \mu_2 \in \mathbb{R}$ and $\sigma_1^2, \sigma_2^2 \in \mathbb{R}_+$ such that $\lfloor \mu_1 - \sigma_1^2 \rfloor \leq \lfloor \mu_2 - \sigma_2^2 \rfloor$, it holds that*

$$d_{TV}\left(TP(\mu_1, \sigma_1^2), TP(\mu_2, \sigma_2^2)\right) \leq \frac{|\mu_1 - \mu_2|}{\sigma_1} + \frac{|\sigma_1^2 - \sigma_2^2| + 1}{\sigma_1^2} .$$

We will need the following technical lemma as well.

**Lemma A.14.** *For $\mu \in \mathbb{R}$ and $\sigma_1^2, \sigma_2^2 \in \mathbb{R}_+$, let us define $X_1 \sim TP(\mu, \sigma_1^2)$ and $X_2 \sim TP(\mu+1, \sigma_2^2)$. Then*

$$\mathbb{P}(X_1 = \ell - 1) = \mathbb{P}(X_2 = \ell)$$

*for all $\ell \in \mathbb{Z}$.*

*Proof of Lemma A.14.* One has to observe that Translated Poisson distribution is equivalent to a Poisson distribution with parameter $\delta$ where $\delta = \mu - \gamma$ and $\gamma = \lfloor \mu - \sigma^2 \rfloor$, but it is shifted by $\gamma$. Moreover, note that $\gamma + 1 = \lfloor \mu + 1 - \sigma^2 \rfloor$, thus $\delta + 1 = \mu - \gamma + 1$. This observation implies the claim with $\mu + 1$. $\qquad \square$

We continue by using both Lemma A.13 and Lemma A.14. Define $\Gamma^{(2)} \sim TP(\mu_1 + 1, \sigma_1^2)$. We can write

$$\sum_{s=0}^{k} \left| \mathbb{P}(\Gamma = s) - \mathbb{P}(\Gamma^{(1)} = s-1) \right| = \sum_{s=0}^{k} \left| \mathbb{P}(\Gamma = s) - \mathbb{P}(\Gamma^{(2)} = s) \right|$$

$$= d_{TV}\left(TP(\mu, \sigma^2), TP(\mu_1 + 1, \sigma_1^2)\right)$$

$$\leq \frac{|\mu - \mu_1 - 1|}{\sigma} + \frac{|\sigma^2 - \sigma_1^2| + 1}{\sigma^2}$$

$$\text{(from Lemma A.13)}$$

$$= \frac{1 - \eta_1}{\sigma} + \frac{\eta_1(1-\eta_1) + 1}{\sigma^2}$$

$$\leq \frac{3}{\sqrt{\sigma}} ,$$

the latter inequality using again the condition $\sum_{i=1}^{k} \eta_i(1-\eta_i) \geq 1$.

Piecing together gives

$$\mathbb{E}_s\left[\left|1 - \frac{\mathbb{P}(\Sigma^{(1)} = s-1)}{\mathbb{P}(\Sigma = s)}\right|\right] \leq \frac{6}{\sqrt{\sigma}} + \frac{3}{\sqrt{\sigma_1}} \leq \frac{9}{\sqrt{\sigma_1}} .$$

This concludes the proof. $\qquad \square$

With these results handy, we are ready to prove Theorem A.9. Recall the functions $f_1(\mathbf{x})$ and $f_2(\mathbf{x})$ defined in the main body.

*Proof of Theorem A.9.* Let $\Sigma = \sum_{j=1}^{k} y_j$. From Lemma A.1, we see that

$$f_1(\mathbf{x}) = 1 - \frac{1}{k} \sum_{i=1}^{k} \mathbb{E}_{\mathbf{y} \sim \mathcal{D}_{\mathcal{Y}|\mathbf{x}}} \left[ \min_{b \in \{0,1\}} \{\mathbb{P}(y_i = b \mid \mathbf{x}, \Sigma)\} \mid \mathbf{x} \right]$$

and

$$f_2(\mathbf{x}) = 1 - \frac{1}{k} \sum_{i=1}^{k} \min\{\eta(x_i), 1 - \eta(x_i)\}$$

so that

$$f_1(\mathbf{x}) - f_2(\mathbf{x}) = \frac{1}{k} \sum_{i=1}^{k} \left( \min\{\eta(x_i), 1 - \eta(x_i)\} - \mathbb{E}_{\mathbf{y} \sim \mathcal{D}_{\mathcal{Y}|\mathbf{x}}} \left[ \min_{b \in \{0,1\}} \{\mathbb{P}(y_i = b \mid \mathbf{x}, \Sigma)\} \mid \mathbf{x} \right] \right)$$

$$\leq \frac{1}{k} \sum_{i=1}^{k} \left| \eta(x_i) - \mathbb{P}_{\Sigma|\mathbf{x}}(y_i = 1 \mid \Sigma, \mathbf{x}) \right| \,,$$

where in the inequality we have again used (4).

But, for any given $s \in \{0, 1, \dots, k\}$,

$$\mathbb{P}(y_i = 1 \mid \Sigma = s, \mathbf{x}) = \mathbb{P}(y_i = 1 \mid \mathbf{x}) \frac{\mathbb{P}(\Sigma = s \mid y_i = 1, \mathbf{x})}{\mathbb{P}(\Sigma = s \mid \mathbf{x})}$$

$$= \eta(x_i) \frac{\mathbb{P}(\Sigma^{(-i)} = s - 1 \mid \mathbf{x})}{\mathbb{P}(\Sigma = s \mid \mathbf{x})}$$

$$= \eta(x_i) \frac{\mathbb{P}(\Sigma^{(-i)} = s - 1 \mid \mathbf{x})}{\eta(x_i) \mathbb{P}(\Sigma^{(-i)} = s - 1 \mid \mathbf{x}) + (1 - \eta(x_i)) \mathbb{P}(\Sigma^{(-i)} = s \mid \mathbf{x})}$$

where $\Sigma^{(-i)} = \sum_{j:j=1,j\neq i}^{k} y_j$ .

Hence

$$f_1(\mathbf{x}) - f_2(\mathbf{x}) \leq \frac{1}{k} \sum_{i=1}^{k} \eta(x_i)(1 - \eta(x_i)) \mathbb{E}_s \left[ \left| \frac{\mathbb{P}(\Sigma^{(-i)} = s \mid \mathbf{x}) - \mathbb{P}(\Sigma^{(-i)} = s - 1 \mid \mathbf{x})}{\mathbb{P}(\Sigma = s \mid \mathbf{x})} \right| \mid \mathbf{x} \right] , \quad (13)$$

where $s$ (conditioned on $\mathbf{x}$) is distributed as $\mathrm{PBin}(\eta(x_1), \dots, \eta(x_k))$. Set for brevity $\mu = \mathbb{E}[\eta(x)(1 - \eta(x)]$, and introduce the short-hand notation

$$C_{\beta,i} = C_\beta(\mathbf{x}^{(-i)}) = \mathbb{I}\left\{ \sum_{j:j=1,j\neq i}^{k} \eta(x_j)(1 - \eta(x_j)) \leq (k-1)\mu - \beta \right\} ,$$

for some $\beta \in [0, (k-1)\mu)$ to be specified. In the above $\mathbf{x}^{(-i)}$ denotes $\mathbf{x}$ with its $i$-th item $x_i$ removed.

We have

$$\eta(x_i)(1 - \eta(x_i)) \mathbb{E}_s \left[ \left| \frac{\mathbb{P}(\Sigma^{(-i)} = s \mid \mathbf{x}) - \mathbb{P}(\Sigma^{(-i)} = s - 1 \mid \mathbf{x})}{\mathbb{P}(\Sigma = s \mid \mathbf{x})} \right| \mid \mathbf{x} \right]$$

$$= C_{\beta,i} \eta(x_i)(1 - \eta(x_i)) \mathbb{E}_s \left[ \left| \frac{\mathbb{P}(\Sigma^{(-i)} = s \mid \mathbf{x}) - \mathbb{P}(\Sigma^{(-i)} = s - 1 \mid \mathbf{x})}{\mathbb{P}(\Sigma = s \mid \mathbf{x})} \right| \mid \mathbf{x} \right]$$

$$+ (1 - C_{\beta,i}) \eta(x_i)(1 - \eta(x_i)) \mathbb{E}_s \left[ \left| \frac{\mathbb{P}(\Sigma^{(-i)} = s \mid \mathbf{x}) - \mathbb{P}(\Sigma^{(-i)} = s - 1 \mid \mathbf{x})}{\mathbb{P}(\Sigma = s \mid \mathbf{x})} \right| \mid \mathbf{x} \right]$$

$$\leq C_{\beta,i} + (1 - C_{\beta,i}) \eta(x_i)(1 - \eta(x_i)) \mathbb{E}_s \left[ \left| \frac{\mathbb{P}(\Sigma^{(-i)} = s \mid \mathbf{x}) - \mathbb{P}(\Sigma^{(-i)} = s - 1 \mid \mathbf{x})}{\mathbb{P}(\Sigma = s \mid \mathbf{x})} \right| \mid \mathbf{x} \right] ,$$

since
$$\eta(x_i)(1 - \eta(x_i))\mathbb{E}_s\left[\left|\frac{\mathbb{P}(\Sigma^{(-i)} = s \mid \mathbf{x}) - \mathbb{P}(\Sigma^{(-i)} = s - 1 \mid \mathbf{x})}{\mathbb{P}(\Sigma = s \mid \mathbf{x})}\right| \,\Big|\, \mathbf{x}\right] \le 1 \,.$$

for every $i$ and every realization $\mathbf{x}$.

On the other hand, a closer look at term
$$(1 - C_{\beta,i})\mathbb{E}_s\left[\left|\frac{\mathbb{P}(\Sigma^{(-i)} = s \mid \mathbf{x}) - \mathbb{P}(\Sigma^{(-i)} = s - 1 \mid \mathbf{x})}{\mathbb{P}(\Sigma = s \mid \mathbf{x})}\right| \,\Big|\, \mathbf{x}\right]$$

reveals that we are in a position to apply Lemma A.10. This allows us to conclude that this expectation is upper bounded by[1]
$$\frac{9(1 - C_{\beta,i})}{\sqrt{\sum_{j:j\neq i,j=1}^{k} \eta(x_j)(1 - \eta(x_j))}}$$

for every realization of $\mathbf{x}$. Using the definition of $C_{\beta,i}$ the above can be further upper bounded by
$$\frac{9}{\sqrt{(k-1)\mu - \beta}} \,.$$

Putting together, we have obtained
$$f_1(\mathbf{x}) - f_2(\mathbf{x}) \le \frac{1}{k}\sum_{i=1}^{k}\left(C_{\beta,i} + \frac{9\eta(x_i)(1 - \eta(x_i))}{\sqrt{(k-1)\mu - \beta}}\right)$$
$$= \frac{1}{k}\sum_{i=1}^{k}C_{\beta,i} + \frac{9\sum_{i=1}^{k}\eta(x_i)(1 - \eta(x_i))}{k\sqrt{(k-1)\mu - \beta}}$$

for every $\mathbf{x}$. Now, from the standard Bernstein inequality,[2]
$$\mathbb{P}_{\mathbf{x}\sim\mathcal{D}_{\mathcal{X}}}\left(C_{\beta,i} = 0 \ \forall i\right) \ge 1 - k\,e^{-\frac{\beta^2}{2k\mu + \beta/6}}$$

and
$$\mathbb{P}_{\mathbf{x}\sim\mathcal{D}_{\mathcal{X}}}\left(\sum_{i=1}^{k}\eta(x_i)(1 - \eta(x_i)) \le k\mu + \beta\right) \ge 1 - e^{-\frac{\beta^2}{2k\mu + \beta/6}}$$

for every $\beta > 0$. This yields
$$\mathbb{P}_{\mathbf{x}\sim\mathcal{D}_{\mathcal{X}}}\left(f_1(\mathbf{x}) - f_2(\mathbf{x}) > \frac{9(k\mu + \beta)}{k\sqrt{(k-1)\mu - \beta}}\right) \le (k+1)\,e^{-\frac{\beta^2}{2k\mu + \beta/6}} \tag{14}$$

holding for $0 \le \beta < (k-1)\mu$.

One can easily see that choosing $\beta = (k-1)\mu/2$ yields, when $k = \Omega(1/\mu)$,
$$\mathbb{P}_{\mathbf{x}\sim\mathcal{D}_{\mathcal{X}}}\left(f_1(\mathbf{x}) - f_2(\mathbf{x}) = \Omega\left(\sqrt{\frac{\mu}{k}}\right)\right) \le k\,e^{-\Theta(k\mu)} \,,$$

which concludes the proof. $\qquad\square$

*Remark* A.15. The analysis contained above also allows us to provide bounds on the additive advantage that are fully conditional on $\mathbf{x} = (x_1, \ldots, x_m)$. For instance, combining Lemma A.10 with Eq. (13), one can easily derive a bound of the form
$$f_1(\mathbf{x}) - f_2(\mathbf{x}) = O\left(\frac{1}{k}\sum_{i=1}^{k}\frac{\eta(x_i)(1 - \eta(x_i))}{\sqrt{\sum_{j:j\neq i}\eta(x_j)(1 - \eta(x_j))}}\right) \,.$$

Then, in order for this bound to be of the form $1/\sqrt{k}$, we need to make further assumptions on the function $x \to \mathbb{P}(y = 1|x)$, the most obvious one being $\mathbb{P}(y = 1|x) \in (0, 1)$ (bounded away from 0 and 1) for all $x$.

---

[1]The condition $\sum_{i=1}^{k}\eta_i(1 - \eta_i) \ge 1$ therein is implied by $C_{\beta,i}(\mathbf{x}^{(-i)}) = 0$, which is equivalent to $(k-1)\mu \ge 1 + \beta$. This, in turn, given our choice of $\beta$, will read as $k = \Omega(1/\mu)$.

[2]Note that we are upper bounding the second moments of the random variables $\eta(x_i)(1 - \eta(x_i))$ with their first moment $\mu$, since the variables $\eta(x_i)(1 - \eta(x_i))$ are in $[0, 1/4]$.

## A.6 The advantage of RR is independent of the number of examples

**Lemma A.16.** *For all data distributions $\mathcal{D}$, all $x_i$, and all datasets $S$ of size $m$, $\mathrm{IEAdv}(\mathcal{M}_{RR}, \mathcal{D}, x_i)$ is independent of $m$, and so is $\mathrm{EAdv}(\mathcal{M}_{RR}, \mathcal{D})$.*

*Proof.* Fix any $m$, any $x_i$, and any data distribution $\mathcal{D}$. Then, by Lemma A.1,

$$\mathrm{IEAdv}(\mathcal{M}_{\mathrm{RR}}, \mathcal{D}, x_i) = \min\{\eta(x_i), 1 - \eta(x_i)\} \tag{15}$$

$$- \mathop{\mathbb{E}}_{y_i \sim \mathcal{D}_{\mathcal{Y}|x_i}} \left[ \min\left\{ \mathop{\mathbb{P}}_{(\mathbf{x}^{(-i)}, \mathbf{y}^{(-i)})}(y_i = 1 \mid \mathbf{x}, \mathcal{M}_{\mathrm{RR}}(\mathbf{x}, \mathbf{y})), \mathop{\mathbb{P}}_{(\mathbf{x}^{(-i)}, \mathbf{y}^{(-i)})}(y_i = 0 \mid \mathbf{x}, \mathcal{M}_{\mathrm{RR}}(\mathbf{x}, \mathbf{y})) \right\} \middle| x_i \right].$$

The first term already has no dependence on $m$. We focus on the second term.

The key to this proof is that a noisy label $RR(y_i) = \tilde{y}_i$ is independent of the other true labels $y_j, j \neq i$. Thus, we have

$$\mathbb{P}(y_i = 1 \mid \mathbf{x}, \mathcal{M}_{\mathrm{RR}}(\mathbf{x}, \mathbf{y})) = \mathbb{P}(y_i = 1 \mid \mathbf{x}, \mathcal{M}_{\mathrm{RR}}(\mathbf{x}, \mathbf{y})_i) = \mathbb{P}(y_i = 1 \mid x_i, \mathcal{M}_{\mathrm{RR}}(x_i, y_i)).$$

Applying this to (15), we have

$$\mathop{\mathbb{E}}_{y_i \sim \mathcal{D}_{\mathcal{Y}|x_i}} \left[ \min\left\{ \mathop{\mathbb{P}}_{(\mathbf{x}^{(-i)}, \mathbf{y}^{(-i)})}(y_i = 1 \mid \mathbf{x}, \mathcal{M}_{\mathrm{RR}}(\mathbf{x}, \mathbf{y})), \mathop{\mathbb{P}}_{(\mathbf{x}^{(-i)}, \mathbf{y}^{(-i)})}(y_i = 0 \mid \mathbf{x}, \mathcal{M}_{\mathrm{RR}}(\mathbf{x}, \mathbf{y})) \right\} \mid x_i \right]$$

$$= \mathop{\mathbb{E}}_{y_i \sim \mathcal{D}_{\mathcal{Y}|x_i}} \left[ \min\left\{ \mathbb{P}(y_i = 1 \mid x_i, \mathcal{M}_{\mathrm{RR}}(x_i, y_i)), \mathbb{P}(y_i = 0 \mid x_i, \mathcal{M}_{\mathrm{RR}}(x_i, y_i)) \right\} \mid x_i \right].$$

Plugging this back into (15) gives us $\mathrm{EAdv}_1(\mathcal{M}_{\mathrm{RR}}, \mathcal{D})$ as desired. Taking the expectation over $x_i \sim \mathcal{D}_{\mathcal{X}}$ completes the proof of the second claim. $\qquad\square$

## A.7 Proof of Theorem 3.5

We prove a more verbose version of Theorem 3.5 which includes the optimal adversary:

**Theorem A.17.** *For any data distribution $\mathcal{D}$, the individual expected attack advantage for randomized response with privacy parameter $\pi = \frac{1}{1+e^\epsilon}$ is*

$$\mathrm{IEAdv}(\mathcal{M}_{RR}, \mathcal{D}, x_1) = \big(\min\{\eta(x_1), 1 - \eta(x_1)\} - \pi\big) \cdot \mathbb{I}\{\eta(x_1) \in [\pi, 1 - \pi]\}.$$

*The optimal adversary maximizing $\mathrm{IEAdv}_1(\mathcal{M}_{RR}, \mathcal{D}, x_1)$ is*

$$\mathcal{A}^*(x_1, \tilde{y}_1) = \begin{cases} 1, & \text{if } \eta(x_1) > 1 - \pi \\ 0, & \text{if } \eta(x_1) < \pi \\ \tilde{y}_1, & \text{otherwise} . \end{cases}$$

In words, upon receiving $x_1$, the optimal attacker $\mathcal{A}^*$ predicts the associated label $y_1$ as the noisy label $\tilde{y}_1$ if the distribution $\mathcal{D}$ contains enough uncertainty about $y_1$ (the condition $\eta(x_1) \in [\pi, 1 - \pi]$ holds) given the extra noise level $\pi$ injected. Otherwise, it predicts deterministically (as $y_1$ is indeed close to being deterministic when $\eta(x_1) \notin [\pi, 1 - \pi]$).

*Proof.* First we characterize the optimal attacker for which the individual expected advantage is maximal. The Bayes optimal decision which minimizes the loss $\mathbb{I}\{A(x_1, \tilde{y}_1) \neq y\}$ conditioned on $x_1$ and $\tilde{y}_1$ is

$$A'(x_1, \tilde{y}_1) = \mathbb{I}\{\mathbb{P}(y_1 = 1 | x_1, \tilde{y}_1) > \mathbb{P}(y_1 = 0 | x_1, \tilde{y}_1)\}$$

which can be written as

$$1 < \frac{\mathbb{P}(y_1 = 1 | x_1, \tilde{y}_1)}{\mathbb{P}(y_1 = 0 | x_1, \tilde{y}_1)} = \frac{\mathbb{P}(\tilde{y}_1 | y_1 = 1, x_1)\, \mathbb{P}(y_1 = 1 | x_1)}{\mathbb{P}(\tilde{y}_1 | y_1 = 0, x_1)\, \mathbb{P}(y_1 = 0 | x_1)} \tag{16}$$

Assume that $\tilde{y}_1 = 1$, in which case (16) becomes

$$\frac{\pi}{1 - \pi} < \frac{\eta(x_1)}{1 - \eta(x_1)} \tag{17}$$

which is true whenever $\eta(x_1) > \pi$, and on the other hand, if $\eta(x_1) < \pi$ then (17) does not hold anymore, thus $A'(x_1, \tilde{y}_1) = 0$. A similar argument holds for $\tilde{y}_1 = 0$: The Bayes optimal decision $A'(x_1, \tilde{y}_1) = 1$ if and only if $\eta(x_1) > 1 - \pi$. Putting together, and noting that $\pi \leq 1/2$, this implies that the optimal attack is

$$A^*(x_1, \tilde{y}_1) = \begin{cases} 1, & \text{if } \eta(x_1) > 1 - \pi \\ 0, & \text{if } \eta(x_1) < \pi \\ \tilde{y}_1, & \text{otherwise .} \end{cases}$$

To compute the reconstruction advantage of optimal attacker $A^*$, we may decompose the feature set into parts as

$$G(\pi) = \{x_1 \, : \, \eta(x_1) \in [\pi, 1 - \pi]\}$$

and its complement set $G^C(\pi)$. It is clear the advantage of the optimal attacker $A^*$ restricted to $G^C(\pi)$ is 0 since $\tilde{y}_1$ gives no extra information on $y_1$ when $x_1 \in G^C(\pi)$, that is, the value of $A^*(x_1, \tilde{y}_1)$ is independent of $\tilde{y}_1$ when $x_1 \in G^C(\pi)$. On the other hand, the advantage of $A^*$ for any $x_1 \in G(\pi)$ is

$$(1 - \pi) - (1 - \min\{\eta(x_1), 1 - \eta(x_1)\}) = \min\{\eta(x_1), 1 - \eta(x_1)\} - \pi \ ,$$

since $\mathcal{M}_{\mathrm{RR}}$ reveals the true label with probability $1 - \pi$, while the attack utility (conditioned on $x_1$) of the uninformed attacker $1 - \min\{\eta(x_1), 1 - \eta(x_1)\}$. This concludes the proof. $\qquad\square$

# B  Proofs for Section 3.2

## B.1  Proof of Theorem 3.7

*Proof.* Consider the difference of log-odds ratio $I_{k,i}(\mathcal{M}, \mathcal{D}, \mathbf{x}, z, i)$, where $z = \mathcal{M}_{\mathrm{LLP}}(\mathbf{x}, \mathbf{y}) = \frac{1}{k} \sum_{i=1}^{k} y_i$ is the output of the aggregation PET. Since $\mathbf{y}$ is independent of $\mathbf{x}$, we can write

$$I_{1,0}(\mathcal{M}, \mathcal{D}, \mathbf{x}, z, i) = I(z) = \ln \frac{\mathbb{P}(y_1 = 1 \,|\, z)}{\mathbb{P}(y_1 = 0 \,|\, z)} - \ln \frac{\mathbb{P}(y_1 = 1)}{\mathbb{P}(y_1 = 0)}$$

$$= \ln \frac{z}{1 - z} - \ln \frac{p}{1 - p}$$

$$= \ln \frac{z(1 - p)}{(1 - z)p} \ ,$$

where we used the fact that, since $kz$ is a binomial random variable, $\mathbb{P}(y_1 = 1 \,|\, z) = z$, independent of $p$. From the standard Bernstein inequality we have, with probability $\geq 1 - \delta$,

$$p - B \leq z \leq p + B \ ,$$

where

$$B = B(k, p, \delta) = \sqrt{\frac{2p(1 - p)\ln(1/\delta)}{k}} + \frac{2}{3k}\ln(1/\delta) \ .$$

Since $z \to \ln \frac{z}{1-z}$ is monotonically increasing in $z \in [0, 1]$, this yields

$$\ln\left(1 - \frac{B}{(1 - p + B)p}\right) = \ln \frac{(p - B)(1 - p)}{(1 - p + B)p} \leq I(z) \leq \ln \frac{(p + B)(1 - p)}{(1 - p - B)p} = \ln\left(1 + \frac{B}{(1 - p - B)p}\right)$$

with the same probability. The condition $k \geq \frac{32\ln(1/\delta)}{p(1-p)}$ implies $2B \leq p - p^2$, which in turn implies $\frac{B}{(1-p+B)p} \leq \frac{1}{2}$, as well as $\frac{B}{(1-p-B)p} > 0$. We further bound the right-most side through $\ln(1 + x) \leq x$ for all $x \geq 0$, and the left-most side via $\ln(1 - x) \geq -x(\ln 2)$ for $x \in [0, 1/2]$. This gives

$$-\frac{B\ln 2}{(1 - p + B)p} \leq I(z) \leq \frac{B}{(1 - p - B)p} \ ,$$

which, after further overapproximations, allows us to write

$$|I(z)| \leq \frac{B\ln 2}{(1 - p - B)p} \ .$$

Next, the condition $2B \leq p - p^2$ implies $(1 - p - B) \geq (1 - p)/2$ so that, overall,

$$|I(z)| \leq \frac{2B(k, p, \delta)(\ln 2)}{p(1 - p)} = O\left(\sqrt{\frac{\ln(1/\delta)}{p(1 - p)k}}\right)$$

with probability at least $1 - \delta$. This concludes the proof. $\qquad\square$

# C Experimental Setup for Section 4 and Further Experimental Results

**Datasets.** We conduct our experiments on the following datasets:

*KDD2012:* We use the click prediction data from the KDD Cup 2012, Track 2 [3] with the feature processing performed in [23]. The learning task for this data is to predict the click through rate for an advertisement based on a number of features related to the advertisement, the page that it appears on, and the user viewing it. There are 11 categorical features that are each one-hot encoded, resulting in a sparse feature vector with 11 non-zero entries in 54,686,452 dimensions. The label for the example is 1 if the ad was clicked, and 0 otherwise.

*Higgs:* We also use the Higgs dataset [4], which is a binary classification dataset where the goal is to distinguish between a signal process that produces Higgs bosons, and a background process which does not. The dataset has 28 real-valued features, which are a mix of kinematic properties measured by particle detectors and hand-crafted high level features designed by physicists. The dataset consists of 11,000,000 examples produced via Monte Carlo simulation. We use the first 10,000 examples as our testing data, and the remaining examples for training.

**Model Architectures.** Next, we describe the models we use for each dataset. We use the same model architecture regardless of which PET has been used to privatize the labels.

*KDD2012 Model:* The model we use for KDD2012 is a deep embedding network. We reduce the dimension of each example from 54,686,452 dimensions to 500,000 dimensions by hashing feature indices. Each hashed feature index is associated with a learned embedding vector in $\mathbb{R}^{50}$, and the representation vector for an example is the sum of the learned embeddings for each of its non-zero hashed feature indices. This representation vector is passed through two dense layers with 100 and 50 units, respectively, and ReLU activation functions. The final output is a single unit with sigmoid activation that is interpreted as the click probability for each example.

*Higgs Model:* For the Higgs data, we do no feature pre-processing. We use a fully connected feed forward neural network with 4 hidden layers all with 300 units and ReLU activation. The final output is a single unit with sigmoid activation that is interpreted as the probability that the example has the positive class.

**Training Setup.** When training a model on the output of any PET, we always use minibatch gradient descent together with the Adam optimizer [24]. PET-specific training details are presented in the following paragraphs.

**Training with Randomized Response.** In our experiments we found that the bias in randomized response can affect prediction accuracy for the model trained on these labels, so we post-process the RR output to remove the bias introduced (see for example [19], Equation (7) therein).

Let $f : \mathcal{Y} \to \mathbb{R}$ be any function (for example the gradient) of a label, and let $y \in \{0, 1\}$. For plain RR,

$$\mathbb{E}[f(RR(y))] = \frac{e^\epsilon - 1}{e^\epsilon + 1} \cdot f(y) + \frac{1}{e^\epsilon + 1} \cdot (f(0) + f(1)).$$

Thus,

$$f(y) = \mathbb{E}\left[\frac{e^\epsilon + 1}{e^\epsilon - 1} \cdot f(RR(y)) - \frac{1}{e^\epsilon - 1} \cdot (f(0) + f(1))\right].$$

Therefore, we can post-process the output of RR to obtain an unbiased estimate of $f(y)$ for any function $f$. This has no effect on the advantage, however it does help the accuracy substantially. We use this debiasing procedure to obtain an unbiased estimate of the binary cross-entropy loss of the model, and perform minibatch gradient descent on the estimated loss. In all of our figures, we use "RR" to denote this debiased version of the mechanism.

**Training with LLP.** For LLP we use minibatch gradient descent to optimize the Empirical Proportion Risk defined in Section 2. We use the binary cross-entropy loss to compare the predicted label proportion against the observed label proportion. As discussed in Appendix D, since the binary cross-entropy loss is a proper loss, the minimizer of the population-level proportion risk for cross-entropy loss is also a minimizer of the population-level cross-entropy loss, as long as the Bayes optimal classifier can be represented by the model architecture.

**Training with LLP+Lap.** LLP+Lap is parametrized by two parameters, $k \in \{1, 2, \ldots\}$ and $\epsilon > 0$. For each bag $\mathcal{B}_i$ the mechanism discloses $\alpha_i + Z_i$, where $\alpha_i$ is the label proportion in that bag, and $Z_i \sim \text{Lap}(\frac{1}{k\epsilon})$, is an independent zero-mean Laplace random variable with parameter $\frac{1}{k\epsilon}$, that is, with variance $\frac{2}{k^2\epsilon^2}$. Theorem D.2 shows that for any minibatch of data, the expected gradient of the proportion matching risk is unaffected by adding mean-zero noise to the label proportion. Therefore, using Proportion Matching to train with LLP+Lap is similar to training with LLP, except the variance of each gradient step is increased.

**Training with LLP+Geom.** LLP+Geom also has parameters, $k \in \{1, 2, \ldots\}$ and $\epsilon > 0$, but $Z_i$ is replaced by a two-sided Geometric random variable, where $Z_i = \frac{Z_i^+ - Z_i^-}{k}$, and $Z_i^+$ and $Z_i^-$ are geometric distribution with probability of success $1 - e^{-\epsilon}$. The value $\alpha_i + Z_i$ is then clipped to $[0, 1]$. It is not hard to see that, for every given $\epsilon$, LLP+Geom with parameters $k = 1$ and $\epsilon$ coincides with RR with parameter $\epsilon$.

Unlike LLP+Laplace, the clipped noisy label proportion is not an unbiased estimate of the bag's true label proportion. We post-process the output of LLP+Geom to obtain unbiased estimates of the bag's label proportion. In particular, let $\alpha_i$ be the true label proportion for a bag, $\alpha_i^{(\text{noise})} = \alpha_i + Z_i$ be the unclipped noisy proportion, and $\alpha_i^{(\text{clip})} = \text{clip}(\alpha_i^{(\text{noise})}, [0, 1])$ be the clipped noisy proportion. Then, using the memoryless property of the Geometric distribution, we have that

$$\mathbb{E}[\alpha_i^{(\text{noise})} \mid \alpha_i^{(\text{clip})}] = \begin{cases} \alpha_i^{(\text{clip})} & \text{if } \alpha_i^{(\text{clip})} \in (0, 1) \\ -\frac{1}{k}\left(\frac{1}{1-e^{-\epsilon}} - 1\right) & \text{if } \alpha_i^{(\text{clip})} = 0 \\ 1 + \frac{1}{k}\left(\frac{1}{1-e^{-\epsilon}} - 1\right) & \text{if } \alpha_i^{(\text{clip})} = 1. \end{cases}$$

When training with LLP+Geom, we replace the clipped label proportion by $\mathbb{E}[\alpha_i^{(\text{noise})} \mid \alpha_i^{(\text{clip})}]$ before computing the proportion matching loss. Since $\alpha_i^{(\text{noise})}$ is an unbiased estimate of $\alpha_i$, so is the conditional expectation. After this debiasing, Theorem D.2 guarantees that the expected gradient of the proportion matching risk is the same as if the true label proportion had been used. It follows that using Proportion Matching to train with LLP+Geom with proportion debiasing is similar to training with LLP except with higher variance.

**AUC vs Advantage Experimental Setup.** For all PETs, we train models to minimize the binary cross-entropy loss. For RR, we use privacy parameters $\epsilon$ in $\{2^{-4}, 2^{-3}, \ldots, 2^5\}$, for LLP we use bag sizes $k$ in $\{2^0, 2^1, \ldots, 2^9\}$, and for LLP+Laplace and LLP+Geom we use all combinations of $\epsilon$ and $k$ from the same sets. For every PET and every value of their privacy parameters, we train the model with each learning rate in $\{10^{-6}, 5 \cdot 10^{-6}, 10^{-5}, 10^{-4}, 5 \cdot 10^{-4}, 10^{-3}, 5 \cdot 10^{-3}, 10^{-2}\}$. Finally, for each combination of privacy parameter and learning rate, we train the model 10 times (each trial corresponds to different model initialization and data shuffling, and RR noise). For each privacy parameter, we report the mean AUC of the learning rate with the highest mean AUC.

**Estimating Attack Advantage.** The attack advantage of the optimal attacker is estimated and reported for every PET with various parameters. The optimal attacker and its advantage have been characterized for RR as well as for LLP, in Lemma A.1 and Theorem A.17, respectively. Crucially, notice that the advantage can be easily calculated with knowledge of the parameters of each PET (label flipping probability or the bag size), and the class conditional distribution $\eta(x)$ — or at least an accurate estimate of it. We considered two estimates of the class conditionals: one based on Deep Neural Networks (DNN) as described above, and one based of $k$-Nearest Neighbors ($k$NN). The $k$NN estimator can produce accurate estimates for class conditional distributions in the large-scale data regimes, due to its strong consistency properties that has been studied in the 70s and 80s, and that became the part of Machine Learning folklore. DNNs can also produce accurate estimates if the training process and architecture is tuned carefully enough. We found that these two approaches result in very similar results on our large scale benchmark datasets, therefore we relied on the estimate provided by the DNN, and we estimated the attack advantage of the optimal attacker based on the output score of the DNN.

*Remark.* It is known that when trained with event-level (i.e., non-private) labels, DNNs are not guaranteed to give rise to *calibrated* classifiers, which is important for estimating the probabilities involved in the definition of attack advantage, while a $k$NN is typically calibrated. Yet, since we

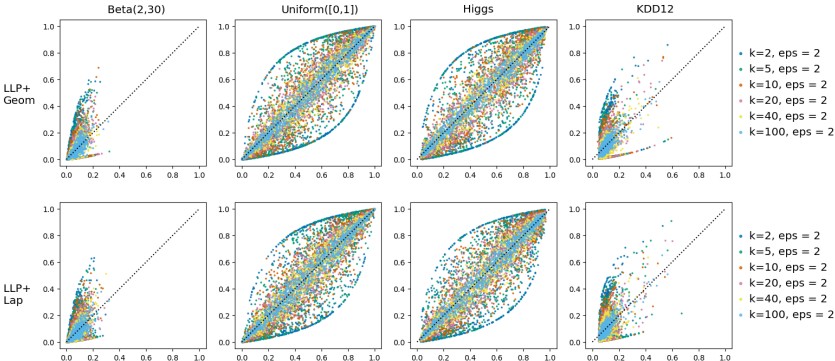

Figure 4: Prior-posterior scatter plots for LLP+Geom and LLP+Lap on two synthetic datasets and the two real-world datasets. The two synthetic datasets have been generated by drawing $\eta(x)$ from a Beta(2,30) distribution and a uniform distribution on $[0, 1]$. The colors of the dots correspond to different parameter values for the PETs. For each bag size $k$ and distribution, we did 1000 independent runs. The further a point is from the $y = x$ dotted line, the more is revealed about its label as a result of the PET.

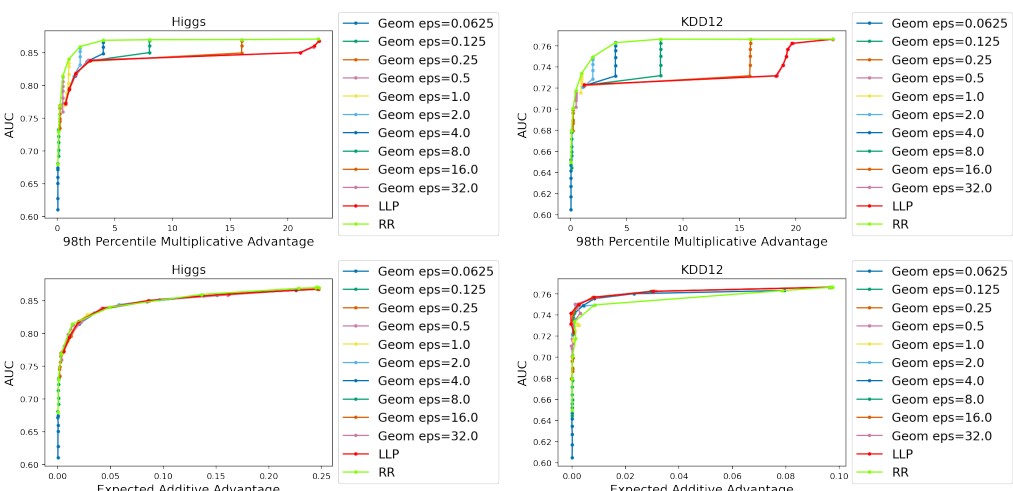

Figure 5: Privacy vs utility tradeoff curves for the various PETs on the Higgs (left) and KDD12 (right) datasets. Utility is measured by AUC on test set, while privacy is either the additive measure (bottom row) or the 98th-percentile of the multiplicative measure (so as to rule out the infinite multiplicative advantage cases that can occur for LLP). Each point corresponds to a setting of the privacy parameter for the PET ($\epsilon$ for RR, $k$ for LLP, and both for LLP+Geom). The $x$-coordinate is the advantage (either additive or multiplicative) value for that PET, while the $y$-coordinate is the test AUC of a model trained from the output of that PET. The AUC of the model trained without a PET roughly corresponds to the top value achieved by these curves. This plot is similar to Figure 3 except that the curves for the LLP+Geom PET correspond to a fixed value of $\epsilon$, rather than a fixed value of $k$.

observed in our initial experiments that the two learning methods resulted in similar overall outcomes, we decided go with DNNs as the underlying classifier for estimating the behavior of the optimal attacker.

**Further experimental results omitted from the main text.** Figure 4 contains scatter plots for LLP+Lap in addition to LLP+Geom for comparison. The two groups of scatter plots look very similar on each dataset. Figure 5 shows the privacy-vs-utility plots (the data is identical to Figure 3) except in this case, the curves for the LLP+Geom PET correspond to a fixed value of $\epsilon$ with the bag size $k$ varying.

**Computational Resources.** We conduct our experiments on a cluster of virtual machines each equipped with a p100 GPU, 16 core CPU, and 16GB of memory. Each training run on Higgs takes approximately 10 minutes, and each training run on KDD12 takes approximately 20 hours.

# D  PROPMATCH with Noisy Label Proportions

In this appendix we argue that the standard PROPMATCH algorithm is still an effective learning algorithm for the LLP+Lap PET (henceforth denoted $\mathcal{M}_{\text{Lap-LLP}}$). Recall that PROPMATCH generally learns by using stochastic gradient descent (SGD) to minimize the empirical proportion matching loss. Our result below shows that the expected value of the gradient of proportion matching loss is the same whether Laplace noise is added to the label proportion or not. In other words, the expected trajectory of SGD for PROPMATCH when using $\mathcal{M}_{\text{Lap-LLP}}$ is the same for all values of $\epsilon$, including $\epsilon = \infty$, which corresponds to the special case of $\mathcal{M}_{\text{LLP}}$. In fact, the main way that decreasing the value of $\epsilon$ (resulting in stronger differential privacy guarantees) affects utility is the increased *variance* of the gradients. The following result characterizes how much the variance increases compared to the case where $\epsilon = \infty$ (or equivalently, compared to PROPMATCH run with $\mathcal{M}_{\text{LLP}}$).

To analyze PROPMATCH, we require a few properties of the loss $\ell : \mathbb{R} \times \mathbb{R} \to \mathbb{R}$ and the noise distribution. First, the loss is *proper*, which means that for every real-valued random variable $A$, the function $f(p) = \mathbb{E}(\ell(p, A))$ has a unique minimizer at $\mu = \mathbb{E}(A)$. We also assume that $\ell(p, y)$ is differentiable in $p$, and that its derivative is affine in $y$, that is $\frac{\partial}{\partial p}\ell(p, y) = a_p y + b_p$, where $a_p, b_p$ may depend on $p$ but not $y$. These conditions are satisfied by the cross entropy (or logistic) loss $\ell(p, y) = -y \log(p) - (1 - y) \log(1 - p)$ (where $p \in [0, 1]$) and the squared error $\ell(p, y) = (p - y)^2$.

Finally, we assume that the noise mechanism operates on a bag of $k$ examples for some $k \in \mathbb{N}$ (possibly $k = 1$). Given the proportion $\alpha$ of positive examples in the bag, it releases $\tilde{\alpha} = \alpha + Z_\alpha$, where $\mathbb{E}[Z_\alpha | \alpha] = 0$ for every fixed $\alpha \in \{0, 1/k, \ldots, 1\}$. This condition is satisfied by LLP (since it adds no noise), LLP+Lap (since Laplace noise is mean 0), and the debiased versions of randomized response and LLP+Geom, described in Appendix C.

Given a class of predictors $\mathcal{H}$ of predictors $h$ that map feature vectors to $[0, 1]$, we can define the problem of minimizing the *population-level noisy proportion-matching loss* for a distribution $\mathcal{D}$, defined via the choice of a bag $\mathcal{B}$ of $k$ labeled examples drawn from $\mathcal{D}$:

$$\min_{h \in \mathcal{H}} \mathbb{E}_{\mathcal{B}, \alpha, \tilde{\alpha}} \left[ \ell \left( \frac{1}{k} \sum_{x \in \mathcal{B}} h(x), \tilde{\alpha} \right) \right]$$

This differs from the loss studied in [13] only in that we allow the label proportion to be perturbed. The PROPMATCH algorithm essentially minimizes this loss.

**Theorem D.1.** *Let $\mathcal{B}$ be a random bag of size $k$, $\ell : \mathbb{R} \times \mathbb{R} \to \mathbb{R}^+ \cup \{\infty\}$ be a loss function, and $\mathcal{M}$ be a mechanism for releasing the label proportions with $\mathcal{M}(\alpha) = \tilde{\alpha} = \alpha + Z_\alpha$, where $\mathbb{E}[Z_\alpha | \alpha] = 0$ for all $\alpha$. Suppose $\mathcal{H}$ contains a predictor $h^*$ that matches the true label probabilities in $\mathcal{D}$ (i.e. $h^*(x) = \mathbb{P}_{\mathcal{D}}(y = 1 | x)$). If $\ell$ is proper (defined above), then $h^*$ is a minimizer of the population-level noisy proportion-matching loss on $\mathcal{D}$. It is essentially unique, in that any other minimizer $h'$ must satisfy $\mathbb{P}_{(x,y) \sim \mathcal{D}}(h'(x) = h^*(x)) = 1$.*

**Theorem D.2.** *Let $\mathcal{B}$ be a random bag of size $k$, $\ell : \mathbb{R} \times \mathbb{R} \to \mathbb{R}^+ \cup \{\infty\}$ be a loss function, and $\mathcal{M}$ be a mechanism for releasing the label proportions with $\mathcal{M}(\alpha) = \tilde{\alpha} = \alpha + Z_\alpha$, where $\mathbb{E}[Z_\alpha | \alpha] = 0$ for all $\alpha$. Suppose that $\frac{\partial}{\partial p}\ell(p, y)$ is affine in $y$.*

*For a predictor $h_\theta \in \mathcal{H}$ (parametrized by $\theta$) denote by $h_\theta(\mathcal{B}) = \frac{1}{k} \sum_{x \in \mathcal{B}} h_\theta(x)$ its average prediction over $\mathcal{B}$. We have*

$$\mathbb{E}\left[ \nabla_\theta \ell\left( h_\theta(\mathcal{B}), \tilde{\alpha} \right) \right] = \mathbb{E}\left[ \nabla_\theta \ell\left( h_\theta(\mathcal{B}), \alpha \right) \right] , \tag{18}$$

*and furthermore*

$$\mathbb{E}\left[ \left\| \nabla_\theta \ell\left( h_\theta(\mathcal{B}), \tilde{\alpha} \right) \right\|_2^2 \right] = \mathbb{E}\left[ \| \nabla_\theta \ell(h_\theta(\mathcal{B}), \alpha) \|^2 \right] + \mathbb{E}\left( Z_\alpha^2 \cdot \left\| \frac{\partial}{\partial \alpha} \nabla_\theta \ell(h_\theta(\mathcal{B}), \alpha) \right\|^2 \right)$$

$$\leq \mathbb{E}\left[ \| \nabla_\theta \ell(h_\theta(\mathcal{B}), \alpha) \|^2 \right] + \left( \max_\alpha \text{Var}(Z_\alpha) \right) \cdot \mathbb{E}\left[ \left\| \frac{\partial}{\partial \alpha} \nabla_\theta \ell(h_\theta(\mathcal{B}), \alpha) \right\|^2 \right] . \tag{19}$$

That is, for nice loss functions like the logistic and square loss, we can write the variance of the noisy LLP gradient as the sum of two terms: the variance of the LLP-based estimate of the proportion-matching loss, and a noise term that scales roughly as the variance of $Z$. For Laplace noise, this

variance is $\frac{2}{k^2\epsilon^2}$. For Geometric noise, the variance is upper bounded above by $\frac{2e^{-\epsilon}}{k^2(1-e^{-\epsilon})^2}$ (which is always smaller than the Laplace variance, most noticeably for large $\epsilon$). For small values of $k$, this upper bound is loose (because clipping and debiasing reduces the overall variance). For example, with randomized response (which is the same as LLP+Geom with $k = 1$), the variance is just $\frac{e^\epsilon}{e^\epsilon+1}$.

Analyzing this formula helps us understand the performance of the noisy variants of LLP. Let us compare LLP+Geom with bag size $k$ to averaging randomized response estimates over $k$ separate examples. Suppose we fix $\epsilon$ and let $k$ increase. There are three effects in play: (1) the noise added for privacy is lower for LLP+Geom (variance $1/k^2$ versus $1/k$); (2) the bias of the gradient estimator is higher for LLP+Geom, since we are estimating the gradient of the proportion-matching loss and not necessarily the true gradient; (3) the variance due to sampling is about the same in both cases, about $1/k$.

We thus get a classic bias-variance tradeoff. As a thought experiment, consider what happens when $\epsilon$ is very small, and almost all the error comes from the added noise. In that case, setting $k$ a bit larger than 1 will help, since the bias introduced will be smaller than the reduction in noise. However, there is a point of diminishing returns, at least when the proportion-matching optimum is different from the true optimum (that is, when the model cannot express the Bayes-optimal predictor). Looking at the experiments with the Higgs data set, we see a muted version of this effect: for small values of $\epsilon$, there is no real difference in the performance of LLP+Geom for $k \in \{1, 2, 4\}$, but larger $k$ introduce a real penalty. For larger values of $\epsilon$ (over 1) on Higgs, and all values of $\epsilon$ on KDD12, the dropoff occurs already for $k = 2$.

This analysis suggests that the advantages of LLP+Geom over RR will be dataset-dependent, and noticeable only for small values of $\epsilon$.

*Proof of Theorem D.1.* This proof is the same as the argument in [12, Thm 3.2]. Fix some bag $\mathcal{B}$ of examples, and take expectation over the sampling of their labels to get a label proportion $\alpha$ and its noisy variant $\tilde{\alpha}$. Because the loss is proper, the unique minimizer of the proportion matching loss is given by the predictions of $h^*$.

$$\arg\min_p \; \mathbb{E}_{\alpha,\tilde{\alpha}} \; \ell(p, \tilde{\alpha}) = \mathbb{E}(\tilde{\alpha}) = \mathbb{E}(\alpha) = \tfrac{1}{k} h^*(x) \,.$$

This is true for every bag $\mathcal{B}$, and thus in expectation over the choice of $\mathcal{B}$. Furthermore, this loss is unique since the indicator functions of bags form a spanning set for the indicators for the support of $\mathcal{D}$. If $h \neq h^*$ with nonzero probability, then the bag-level predictions of $h$ will differ from those of $h^*$ with nonzero probability. $\qquad\square$

*Proof of Theorem D.2.* Let $\ell'(p, y) = \frac{\partial}{\partial p}\ell(p, y) = a_p y + b_p$, where $a_p = \frac{\partial}{\partial y}\ell'(p, y)$. Using the chain rule and the fact that $\ell'$ is affine, we get:

$$\begin{aligned}
&\nabla_\theta \ell\left(h_\theta(\mathcal{B}), \tilde{\alpha}\right) \\
&= \ell'(h_\theta(\mathcal{B}), \tilde{\alpha})\nabla_\theta h_\theta(\mathcal{B}) \\
&= \left(\ell'(h_\theta(\mathcal{B}), \alpha) + Z_\alpha a_{h_\theta(\mathcal{B})}\right)\nabla_\theta h_\theta(\mathcal{B}) \\
&= \nabla_\theta \ell(h_\theta(\mathcal{B}), \alpha) + Z_\alpha a_{h_\theta(\mathcal{B})}\nabla_\theta h_\theta(\mathcal{B}) \\
&= \nabla_\theta \ell(h_\theta(\mathcal{B}), \alpha) + Z_\alpha \cdot \tfrac{\partial}{\partial\alpha}\nabla_\theta \ell(h_\theta(\mathcal{B}), \alpha) \,.
\end{aligned}$$

Since the noise is unbiased for every $\alpha$, the second term above is 0 in expectation for every pair $(\mathcal{B}, \alpha)$. This proves that the estimator is unbiased (Equation (18)). To analyze its variance, note that the unbiasedness of $\tilde{\alpha}$ means that the two terms are decorrelated, and we can sum their convariances:

$$\begin{aligned}
&\text{Covar}\left(\nabla_\theta\ell(h_\theta(\mathcal{B}), \alpha) + Z_\alpha \cdot \tfrac{\partial}{\partial\alpha}\nabla_\theta\ell(h_\theta(\mathcal{B}), \alpha)\right) \\
&= \text{Covar}\left(\nabla_\theta\ell(h_\theta(\mathcal{B}), \alpha)\right) + \text{Covar}\left(Z_\alpha \cdot \tfrac{\partial}{\partial\alpha}\nabla_\theta\ell(h_\theta(\mathcal{B}), \alpha)\right) \\
&\quad + \mathbb{E}\left(\langle\nabla_\theta\ell(h_\theta(\mathcal{B}), \alpha) \,, \, \tfrac{\partial}{\partial\alpha}\nabla_\theta\ell(h_\theta(\mathcal{B}), \alpha))\rangle\right) \\
&= \text{Covar}\left(\nabla_\theta\ell(h_\theta(\mathcal{B}), \alpha)\right) + \text{Covar}\left(Z_\alpha \cdot \tfrac{\partial}{\partial\alpha}\nabla_\theta\ell(h_\theta(\mathcal{B}), \alpha)\right) \,.
\end{aligned}$$

The expected squared norm of a random vector is the trace of its covariance. Thus, the expression above gives us the equality in Equation (18). The final inequality follows again from decorrelation: by conditioning on the pair $(\mathcal{B}, \alpha)$, we can pull out $\mathbb{E}[Z_\alpha^2]$ from the outer product of

$Z_\alpha \cdot \frac{\partial}{\partial \alpha} \nabla_\theta \ell(h_\theta(\mathcal{B}), \alpha)$ with itself. Taking the trace (to get the expected square norm), we can use the fact that $\|\nabla_\theta \ell(h_\theta(\mathcal{B}), \alpha)\|^2$ is nonegative, to replace $\mathbb{E}[Z_\alpha^2]$ with an upper bound that holds for all $\alpha$. This yields the final inequality in Equation (19), as desired. □

We have analyzed PROPMATCH in this section since it is the algorithm used in our experiments. Similar results hold for other LLP algorithms, like EASYLLP [12], though we do not work those out here.

