# OpenReview forum: "Auditing Privacy Mechanisms via Label Inference Attacks"
_NeurIPS.cc/2024/Conference — NeurIPS 2024 spotlight_

### Official Review · Reviewer_mygV · 2024-07-12

**Soundness:** 4
**Presentation:** 3
**Contribution:** 4
**Rating:** 8
**Confidence:** 3

**Summary:**

This paper develops measures for auditing privacy-preserving mechanisms that consume datasets with public features and private labels. The auditing measures compare two attackers whose goal is to infer a private label: a weaker attacker who has access to the dataset’s features and prior knowledge of the correlation between features and labels, and a stronger attacker who additionally has access to the privatized labels output by the mechanism. The advantage of the stronger attacker over the weaker attacker can be measured in two ways: additively or multiplicatively. Upper bounds on the additive and multiplicative measures are obtained for two privacy-preserving mechanisms: randomized response (RR) and learning from label proportions (LLP). These mechanisms are otherwise difficult to compare as RR satisfies label differential privacy, but LLP does not. Experiments on synthetic and real datasets, demonstrate that privacy-utility trade-offs can be evaluated using the proposed measures, and that RR dominates LLP.

**Strengths:**

**Originality:**
While variations of the additive and multiplicative measures have appeared in prior work (as acknowledged by the authors), this paper appears to be the first to apply the multiplicative measure to auditing generally, and the first to apply the additive measure to auditing of non-differentially private mechanisms.

**Significance:**
The proposed measures allow for auditing of non-differentially private mechanisms for label privacy, filling an apparent gap in the literature. The empirical finding about the dominance of randomized response (a differentially private mechanism) over learning from label proportions (a more heuristic mechanism) in terms of privacy and utility is interesting, and may encourage adoption of differential privacy in the field.

**Quality:**
The experiments are thorough, covering a range of datasets and parameter settings for the mechanisms under audit. In addition to computing the proposed audit measures, plots visualizing the prior/posterior update for individual data points are provided, which adds another dimension to the results.

**Clarity:**
The paper is generally clear (see comments below).

**Weaknesses:**

The narrative could be strengthened, perhaps by leaning more on the application to online advertising. At times, I found myself questioning the path taken in the paper. For example, why are the additive and multiplicative measures both needed? It seems the multiplicative measure is superior as it is data-dependent, and a relative comparison is arguably more natural for probabilities. I also wondered why the definitions of expected attack utility and expected attack advantage differ from Wu et al. (2022), who condition on the features $\mathbf{x}$. It would be helpful to explain why conditioning on the features doesn’t make sense here.

Some of the bounds make strong assumptions about the data distribution. For instance, Theorems 3.2 and 3.7 assume the labels and features are uncorrelated, which doesn’t seem to be an interesting case to consider. There are analogues of Theorem 3.2 for general data distributions, but not Theorem 3.7.

The proposed measures are claimed to provide “a more balanced and nuanced picture” (line 40) of privacy/accuracy trade-offs. And the model of the adversary’s uncertainty is claimed to be “reasonable” (line 298). It would be helpful to elaborate on these claims a little more, so that practitioners can better understand any caveats. I understand that the measures are more optimistic because they depend on the adversary’s uncertainty (rather than assuming the worst-case). However the validity of the measures then depends on having a good model of the adversary’s uncertainty. The model adopted in the paper seems to assume the adversary has no side information. It would be great to discuss this assumption in the context of online advertising.


**Minor:**
- Figure 1: It is difficult to distinguish between different coloured points for LLP and LLP+Geom.
- line 224: Incorrect reference to Theorem 3.3.
- line 322: Is the classifier calibrated?
- line 252 and 260: There is an inconsistency in the last argument of the measure: $i$ vs $x_i$.

**Questions:**

- How would you envisage these measures being applied? Would they be used to tune privacy parameters? Compare privacy-preserving mechanisms?
- Why is the model of the adversary’s uncertainty reasonable? Could you expand on this?
- Why does the definition of expected attack utility differ from Wu et al. (2022)?

**Limitations:**

Section 5 discusses limitations including the assumption of iid data, the need to estimate the adversary’s label uncertainty and label semantics. Another limitation of the work is that the measures are only instantiated for binary labels.

---

> ### Author Rebuttal · Authors · 2024-08-06
>
> * Why are additive and multiplicative measures both needed?
>
> The multiplicative measure is generally stronger, in that a small multiplicative advantage implies a small additive one, but not vice versa. We studied both additive and multiplicative measures for a few reasons. While the multiplicative measure is closer to the standard DP interpretation of privacy, the additive measure may be more naturally interpretable in terms of X% increase in inference risk. Besides, the additive measure is more generous to LLP, so the fact that DP mechanisms in practice outperform LLP even when we use the additive measure becomes even more surprising.
>
> * Is the classifier calibrated? (See 823 in appendix)
>
> As explained in LL. 823-836 in the appendix, we had initial experiments with both Deep Neural Network (DNN) architectures and k-Nearest Neighbor (kNN) classifiers. When trained with event-level (that is, non-private) labels, DNNs are not guaranteed to give rise to calibrated classifiers, while a kNN classifier is typically calibrated. Since we observed in our initial experiments that the two resulted in similar overall outcomes, we decided to go with DNNs as the underlying classifier throughout.
>
> * How would you envisage these measures being applied? Would they be used to tune privacy parameters? Compare privacy-preserving mechanisms?
>
> An organization that is considering multiple otherwise incomparable PETs could use our measures to 1) establish which one has the best privacy/accuracy tradeoff for the types of data and downstream tasks they’re concerned with, and 2) set privacy parameters according to their risk tolerance. For example, in our case, Figure 3 shows that RR achieves the best tradeoff for our setting. In addition, the curves can help with choosing epsilon in an informed manner, though of course the decision should consider other contextual factors, and more than one measure of advantage.
>
> * Why is the model of the adversary’s uncertainty reasonable? Side information.
>
> We emphasize that we do model a particular kind of side information. However, we make the assumption that, conditioned on the adversary’s side information and the public features, the labels within a batch are independent of each other. Our model aims to strike a balance between minimizing assumptions (as conservative approaches like DP do) and expressiveness. Alas, very conservative approaches will immediately disqualify deterministic mechanisms. Our goal is a reasonable framework that allows one to put deterministic mechanisms and randomized ones on the same footing.
>
> We use a nonprivately trained predictor to capture adversarial uncertainty. This felt like the right way to get a “population-level” knowledge.
>
> Given that bagging is done randomly, such population-level modeling seems like a good fit for the mechanisms we consider. If bags were not generated uniformly (e.g., in the extreme, if records from users in the same household were bagged together), then distributional assumptions would seem like the wrong approach, and the conservativeness of DP would make more sense.
>
> In the context of online advertising: we imagine that existing predictive models could be used as the source of baseline probabilities. The experiments with the kdd12 data set (which comes from online advertising) provide a simple illustration.
>
> * Why does the definition of expected attack utility differ from Wu et al. (2022)?
>
> Unlike label DP mechanisms analyzed by Wu et al., LLP does not easily admit bounds that hold for all data distributions and are conditioned on arbitrary feature vectors $x$. In particular, in order to obtain a bound that has a clear dependence on the bag size $k$, the conditional distribution $\eta(x)$ must be bounded away from 0 and 1–see Remark A.15 in the appendix. Hence, since we wanted to measure advantage for both label DP and label aggregation, so as to put the two on the same footing, we had to consider a wider spectrum of measures. Specifically, we considered three guarantees in terms of $x$ (from weaker to stronger): (i) *expected* advantage, where the guarantee is only in expectation over the distribution of $x$;. (ii) advantage in high probability over $x$; and (iii) “for all $x$” advantage (that is, conditioning on $x$, as in Wu et al.). In all three cases, though, the guarantees are in expectation over the conditional distribution of $y$ given $x$. For label aggregation (LLP), the latter two guarantees are basically only contained in the appendix (Section A.5 there in -- see, e.g., Thm A.9 and Remark A.15).

---

> > ### Comment · Reviewer_mygV · 2024-08-08
> >
> > Thanks, the authors' response has helped me better appreciate the context/objectives of the paper. I have decided to increase my score for soundness, contribution and the overall rating.

---

### Official Review · Reviewer_Us4Y · 2024-07-12

**Soundness:** 2
**Presentation:** 2
**Contribution:** 2
**Rating:** 3
**Confidence:** 4

**Summary:**

This paper proposes a number of measures of information gained by an adversary from a mechanism.  It presents some theorems and performs a number of experiments.

**Strengths:**

Research on understanding better how much can be inferred when privacy mechanisms are in place is useful.
The text uses mostly correct and understandable English

**Weaknesses:**

The formalization is insufficiently rigorous and often unclear.
The claimed results don't seem to be always sound.
While it is to some extent interesting to define new measures, it is important to understand what we can learn from these measures.  The paper doesn't motivate why the proposed measures are preferable over existing measures.  The experiments are illustrative but don't really answer the question of the value of the measures, and also don't compare to existing alternatives.
Several other measures exist, e.g., there is quite some work using entropy-based measures to quantify how much an adversary can learn, i.e., the information gained by the adversary is the difference between the prior entropy and the posterio entropy.


Some details:

* Often $\mathcal{M}$ is called a mechanism rather than a PET.  Usually, one uses PET to refer to a strategy to enhance privacy, not to a learning algorithm.
* In line 154, a $(\mathbf{x},\mathbf{y})$ is drawn from $\mathcal{D}^m$.  However, $\mathcal{D}^m$ is a product of $m$ datasets, each of the form $\mathcal{X}\times\mathcal{Y}$.  In contrast, $(\mathcal{x},\mathcal{y})$ is a pair of an element from $\mathcal{X}^m$ and an element from $\mathcal{Y}^m$.  These don't match syntactically.  Similar conversions of object seem to happen at several other places in the paper.
* In line 154, I also assume that $\mathcal{A)(\ldots)_i$ means the $i$-th component of the output of $\mathcal{A}$ (rather than for example $\mathcal{A}$ applied to the $i$-th component of $(\mathbf{x},\mathbf{y})$.
* In line 198, Theorem 3.2:
  - it is unclear what is the distribution of $\alpha$ (In fact, $\alpha$ has been used in various contexts before and it seems hard to unambiguously decide the meaning of $\alpha$ here).
  - it is unclear what is the role of the threshold $\beta$
  - $p$ is a probability, therefore the domain of $p$ is $[0,1]$.  The first alternative in the righthandside of the equation gives $p\in[0,1]$, so this first option will always apply.  It is unclear why then there is a second alternative $|p-1/2|\ge \beta$.  Depending on the constants hidden in $\Omega$ this second alternative may not be relevant, or it may is some cases provide a stronger bound.
  - Earlier text suggests that $\alpha_i=\frac{1}{k}\sum_{j=1}^k y_{i,j}.  In the proof, line 528, it seems now that $\alpha=\alpha_i$, but it is unclear what is the value of $i$ for which this holds.  Maybe you mean $\alpha$ is a random variable taking values $\alpha_i$ where $i$ is drawn uniformly from $[k]$ but even that doesn't solve all questions around here.
  - line 528 sets $\Sigma=k\alpha$ ($\Sigma$ is a real value), but doesn't explain what $\Sigma$ means in the conditional probability $P(\ldots | \Sigma)$ in the next line ($\Sigma$ would be an event or boolean condition).  The proof gets hard to understand at that point.
* in Line 208, when describing the behavior when $k$ increases I assume you are considering a scenario in which $n$ is kept constant.
* Theorem 3.3 suggests that if $\eta(x) \in \{0,1\}$, then $EAdv=0$.  This is unexpected:
  - around line 154, there is the definition of $EAU$.  The prior/uninformed attacker doesn't get any information about the distribution $\mathcal{D}$ as $\mathcal{M}_\bot(x,y)=\bot$, so he can only guess classes with probability proportionally to the frequency of these class labels.  In contrast, the informed attacker gets information from a mechanism $\mathcal{M}$.   One would hope the attacker can learn something from $\mathcal{M}$, especially as learning a deterministic function if easier than learning a probabilistic function.  Still, Theorem 3.3 seems to say that the EAdv will be zero.

**Questions:**

--

**Limitations:**

The paper doesn't discuss limitations.

---

> ### Author Rebuttal · Authors · 2024-08-06
>
> The reviewer’s main criticisms focus on: (a) comparison to related work, and (b) clarity of the mathematical setup and notation.
>
> **Comparison to related work**
>
> * “it is important to understand what we can learn from these measures. [...] Other measures exist, like entropy based. The paper doesn't motivate why [...] preferable over existing measures.”
>
> We do discuss related work extensively in Section 3.3. Indeed, our paper explicitly builds on several other works that aim to measure advantage. We are not aware of entropy-based measures being applied in the specific context of label aggregation, or label privacy more generally. Yet, we would be very interested to add comparisons to such works if the reviewer could suggest some specific starting points.
>
> One possibly relevant point is that multiplicative measures such as the one we consider here generally upper bound entropy-based quantities like mutual information (which corresponds to the reviewer’s suggestion of entropy drop from prior to posterior). We can add a general discussion of that point to Section 3.2.
>
> * The paper doesn’t discuss limitations.
>
> This is not accurate, please see Section 5.
>
> **Mathematical setup and notation**
>
> * On our measures and the associated theoretical results.
>
> There seem to be two general misunderstandings regarding our theoretical setting. First, in our definitions of attack utility and attack advantage we are assuming the adversaries (both the informed and the uninformed ones) have full knowledge of the distribution $D$ generating the data. Second, for the PETs we actually analyze, since the data is i.i.d. the attack advantage is either depending on a single example $(x_1,y_1)$ (for Randomized Response) or a single bag of examples (for label aggregation). This is why, for instance, the bounds for label aggregation in Thm 3.2, 3.3 and 3.4 are referring to a single random bag of size $k$ instead of a set of $n$ bags of size $k$, and thus we dropped the bag index $i$ from the notation. Please recall LL. 185-189.
>
> * L. 154, syntactic mismatch.
>
> Yes, this is a minor abuse of notation (and the specific sampling that we use is described in lines 82-84). In general we identify elements of $X^m \times Y^m$ and $(X \times Y)^m$ if they describe the same sequence of labeled examples. We will fix it throughout so as to avoid this notational inconsistency.
>
> * L. 154, is $\mathcal{A}(...)_i$ the i-th component of the output of $\mathcal{A}$ ?
>
> Yes, your interpretation is correct. As described on L. 148, the attacker receives the feature vectors for each example, the output of the PET, and they output a prediction for the label of each example. We will add a clarification to that effect, thanks.
>
> * In line 198, Thm 3.2, ambiguity in the distribution and meaning of $\alpha$.
>
> Thank you for identifying this ambiguity. Theorem 3.2 deals with a single bag containing $k$ examples, and the relevant distributions are: the $k$ pairs $(x_j,y_j)$ are drawn i.i.d. from $D$, and $\alpha$ is set to be the average over the $k$ labels. The theorem focuses on the case of a *single* bag since when using the $M_{LLP}$ mechanism, the bags observed by the attacker are independent and, since $D$ is known, the attacker gains no extra advantage from operating on an entire dataset of bags at once. It is thus sufficient to measure their success on a single bag. We will add some clarifying text to the paper.
>
> * Thm 3.2: role of $\beta$ and two bounds.
>
> Thm 3.2 provides two guarantees. As you point out, one of the guarantees holds without any condition on $p$, while the other requires that $p$ is sufficiently far from $1/2$. In particular, as long as $p$ is bounded away from $1/2$, the expected advantage decreases *exponentially* with $k$, rather than at the much slower rate of $k^{-1/2}$. The parameter $\beta$ in the exponent quantifies how far from $1/2$ that $p$ is. Thus, $\beta$ is a property of distribution $D$. The two bounds we give are generally incomparable, but when $k$ is large and there is a big enough gap $\beta$, the second bound is clearly better.
>
> * L 528-529 meaning of conditioning on $\Sigma$.
>
> Given that $\alpha$ is the proportion of the $k$ examples that had positive labels, $\Sigma = k\alpha$ is the number of positive labels in the bag, so $\Sigma$ is a random variable supported on $\{0, …, k}$. It is valid to condition probabilities on random variables. The notation $P(...| \Sigma)$ is nothing else than a conditional expectation, which is itself a random variable since it is a function of the random variable $\Sigma$.
>
> * L 208, when $k$ increases, is $n$ kept constant?
>
> As said above, because the bags are independent, we only need to consider a single bag when computing advantage. So $n$ is irrelevant to the advantage result in Theorem 3.3 (and subsequent results as well) since the attacker’s ability to predict the labels in one bag does not depend on the number of bags they have.
>
> * Theorem 3.3 unexpectedly suggests that if $\eta(x) \in \{0,1\}$ then EAdv = 0.
>
> As stated on LL. 160-163, both the uninformed and the informed attackers know the data distribution $D$ (one can also infer this from the definitions, since for a fixed $D$, we take the sup over adversaries–including ones that know $D$). Thus, when $\eta(x) = 0$ or $\eta(x) = 1$, both attackers have perfect knowledge of the label regardless of what the mechanism releases, since the label is unambiguously determined by the features (which are assumed to be public). So the inference advantage (conferred by the mechanism) is 0.
>
> * Often $M$ is called a mechanism rather than a PET. Usually, one uses PET to refer to a strategy [...] not to a learning algorithm.
>
> It is true that we use both PET and Mechanism to refer to privacy enhancing technologies. However, we would like to emphasize in all cases, the PETs/Mechanisms we discuss are strategies for enhancing privacy and not learning algorithms. We will make this clearer in the paper.

---

> > ### Comment · Reviewer_Us4Y · 2024-08-10
> >
> > #### Comparison to related work
> >
> > > >    “it is important to understand what we can learn from these measures. [...] Other measures exist, like entropy based. The paper doesn't motivate why [...] preferable over existing measures.”
> >
> > Please, when you quote text of other people, copy it exactly (except for omissions you can indicate with [...]).  Else, please don't use a quotation style and just say "we understand that ... (your own interpretation) ...".
> >
> > > We do discuss related work extensively in Section 3.3.
> >
> > It is important to read the full comment.   This is not only about "comparing to existing work".   The first sentence you quote says "it is important to understand what we can learn from these measures."  Comparison with related work is indeed one way to partially understand the meaning of a new measure, but it is not sufficient on itself.  It is relatively easy to define a new measure which is different from existing measures, but that doesn't necessarily mean that this new measure is interesting.  "What can we learn from the measure?" asks for explanation on the meaning of the output of the measure, and on how this output can be useful to perform tasks readers are familiar with (indeed, ideally in a way which is better than existing measures).
> >
> > > Indeed, our paper explicitly builds on several other works that aim to measure advantage. We are not aware of entropy-based measures being applied in the specific context of label aggregation, or label privacy more generally. Yet, we would be very interested to add comparisons to such works if the reviewer could suggest some specific starting points.
> >
> > My original text doesn't claim that there are papers specifically focusing on entropy-based measures for the specific context for label privacy.  I only said that there exist other measures, and that some of these are entropy-based.  I believe that the entropy-based work which exists is sufficiently general to apply (among others) to the specific context of label privacy.
> > Even if there would not exist a paper whose title is exactly "entropy-based measure in the specific context of label privacy" this shouldn't prevent authors from applying general methods to the specific context of label privacy for the purpose of comparison.
> >
> > I'm not an expert myself in the specific combination of entropy based measures and label privacy, but it is easy to find papers which discuss measuring privacy using mutual information strategies, e.g.,
> >
> > * Konstantinos Chatzikokolakis, Tom Chothia, and Apratim Guha. Statistical measurement of information leakage. In Tools and Algorithms for the Construction and Analysis of Systems, 16th International Conference, TACAS 2010, volume 6015 of Lecture Notes in Computer Science, pages 390–404. Springer, 2010.
> > * Ali Makhdoumi, Salman Salamatian, Nadia Fawaz, and Muriel Médard. From the information bottleneck to the privacy funnel. In 2014 IEEE Information Theory Workshop, ITW 2014, 501–505. IEEE, 2014.
> > * Lejla Batina, Benedikt Gierlichs, Emmanuel Prouff, Matthieu Rivain, François-Xavier Standaert, and Nicolas Veyrat-Charvillon. Mutual information analysis: a comprehensive study. J. Cryptol., 24(2):269–291, 2011.
> >
> > As said, some existing measures are not entropy-based, e.g.,
> >
> > Hanshen Xiao, Srinivas Devadas. PAC privacy: Automatic privacy measurement and
> > control of data processing. In Advances in Cryptology - CRYPTO 2023 - 43rd Annual Inter-
> > national Cryptology Conference, CRYPTO 2023, volume 14082 of Lecture Notes in Computer
> > Science, 611–644. 2023.
> >
> > Most of this work aims at understanding how much the release of some output increases the risk of inferring sensitive information (similar to the key question in lines 144-146).  Hence, while my task is not to provide an exhaustive list of only relevant references, I conclude that there at least a few lines of thinking with goals similar to the current submission which are not mentioned in the current discussion of related work (nor in the comparative experiments or qualitative argumentation why the current proposal would be better).
> >
> > > One possibly relevant point is that multiplicative measures such as the one we consider here generally upper bound entropy-based quantities like mutual information (which corresponds to the reviewer’s suggestion of entropy drop from prior to posterior). We can add a general discussion of that point to Section 3.2.
> >
> > Indeed the specific example of mutual information is an exact, hence hard to compute, hence most authors approximate it.  It may indeed be useful to point out that the proposed new measure is an upper bound of some existing measure (if that would be the case, please provide a proof).
> >
> > > We do discuss related work extensively in Section 3.3.
> >
> > Sec 3.3 extensively discusses a narrow segment of related work.  It discusses "a recent line of work on auditing learning algorithms via membership inference attacks", but virtually no other related work.  It doesn't argue whether the new proposal has advantages.

---

> > > ### Comment · Reviewer_Us4Y · 2024-08-10
> > >
> > > #### Section 5
> > >
> > > While one could indeed argue Section 5 points to some limitations, the text in Section 5 does not seem very accurate, nor does it discuss these interesting points in much depth.
> > >
> > > * Line 410 makes the overly general claim "for the first time allow us to compare their privacy-accuracy trade-off curves".  It is unclear why this wasn't possible with earlier approaches.
> > > * Line 411-412 state that the approach is tailored towards a specific setting, but it doesn't explain how these limitations work and how it would fail in more general settings:
> > >   - i.i.d. data is required, but I guess that if the data is not i.i.d., in this setting this just means that the test data may be from a different distribution than the training data and hence the adversary can even less reliably infer the label from the output?  So the measures the current submission presents would remain valid worst case bounds?
> > >   - the text says "public features together with one sensitive binary feature", but it is unclear why the proposed measures can't be applied in other cases.
> > > * The latter point may be related to line 415 refers to multiple mechanisms.  Composability of measures has always been an important point in privacy measure work, but it doesn't seem to be discussed here.  In fact, the text seems to suggest that there is no easy way to compose, i.e., if we have audited two mechanisms and now want to audit their composition, we can't easily from the measures on the first two mechanisms compute a worst case bound for the measure applied on the composition.  Not having such property would be a significant drawback which would deserve being discussed (with arguments why the proposed measure is nevertheless still better than alternatives).
> > >
> > > #### Terminology
> > >
> > > > > Often M is called a mechanism rather than a PET. Usually, one uses PET to refer to a strategy [...] not to a learning algorithm.
> > >
> > > > It is true that we use both PET and Mechanism to refer to privacy enhancing technologies. However, we would like to emphasize in all cases, the PETs/Mechanisms we discuss are strategies for enhancing privacy and not learning algorithms. We will make this clearer in the paper.
> > >
> > > I disagree.
> > > * Privacy Enhancing Technology (PET) refers to a wide range of strategies to obtain or improve privacy.  Examples include homomorphic encryption, differential privacy, function secret sharing, etc.  One can see from relevant websites such as the PETS symposium and hits of googling "privacy enhancing technology" that a PET is not necessarily a mechanism
> > > * A mechanism is a specific technical term referring to a randomized algorithm with an input and an output.   Literature on differential privacy almost always uses the term "mechanism", literature in cryptology often refers to "randomized algorithm".   It is indeed true that "mechanism" is a more general concept than "randomized learning algorithm", but it still is significantly different than the technologies the word PET refers to.

---

> > > > ### Comment · Reviewer_Us4Y · 2024-08-10
> > > >
> > > > #### Technical details
> > > >
> > > > In general, please consider that the list of technical details provided in my review is only a sample.  Other sections seem to have a similar level of rigor and require being improved too.
> > > >
> > > > > There seem to be two general misunderstandings regarding our theoretical setting. First, ...
> > > >
> > > > I agree with what you explain here, but stress that it is important that the reader too understands this.
> > > >
> > > >
> > > > > >   L. 154, syntactic mismatch.
> > > >
> > > > > Yes, this is a minor abuse of notation (and the specific sampling that we use is described in lines 82-84). In general we identify elements of and if they describe the same sequence of labeled examples. We will fix it throughout so as to avoid this notational inconsistency.
> > > >
> > > > If you abuse notation, it is important to tell this to the reader.  Lines 82-84 do not do that.
> > > >
> > > > > The two bounds we give are generally incomparable, but when k is large and there is a big enough gap $\beta$, the second bound is clearly better.
> > > >
> > > > This may be true, still as my review points out the statement on itself makes things hard to interpret, especially as the big-O notations hiding constant factors make it hard for a user to decide which of both bounds is the best one.   It seems that in this case it would be better to give explicitly the constant factor (or at least a sufficient approximation of it, or a reference to an appendix) so the reader can appreciate the relation.
> > > >
> > > >
> > > > #### Further thoughts
> > > >
> > > >
> > > > Thanks for the rebuttal.  Thinking again about the submission, I now see the following drawbacks which maybe I didn't sufficiently realize or not write in sufficient precision in my original review:
> > > >
> > > > * it seems the proposal has no composability, i.e., in contrast to most existing work, given two mechanisms M1 and M2, EAU(A,(M1,M2),D) is hard to upper bound from EAU(A,M1,D) and EAU(A,M2,D).  This is a significant drawback which doesn't seem to be compensated by other advantages.
> > > > * The evaluation of privacy happens on the same set as the training set, e.g., in the equation after line 154.  In many use cases, the adversary gets some output (e.g., a learnt model) but not the original training data (feature parts).  These use cases can't be handled by the current proposal, it seems to be assumed the adversary also has access to the dataset.  E.g., Line 164 first applies $\mathcal{A}$ to the complete training set $\mathbf{x}$ and then takes the $i$-th component.  It doesn't allow to run $\mathcal{A}$ on some instance $x_i$ not in the training set.  This seems to limit the applicability of the proposal.
> > > > * The approach assumes adversaries have knowledge to the full distribution $\mathcal{D}$ and have the computational power to work with it.  This is different from the common assumption in machine learning where one assumes a "fixed but unknown distribution" for the data.
> > > > * The paper doesn't follow a "worst case approach", which is reasonable.  However, rather than considering average behavior over the randomization of the mechanism, the paper considers an average case over the distribution $\mathcal{D}$, which may have significant implications from the point of learning theory.
> > > >
> > > > So in conclusion, I still feel the authors propose a new measure without sufficiently demonstrating advantages it has compared to the state of the art techniques.  The paper does compare to related work, but only a narrow segment of most similar work.  The approach clearly has some drawbacks, but the paper doesn't discuss their implications in much depth.

---

> > > > > ### Author Response · Authors · 2024-08-13
> > > > > **Response to Reviewer Us4Y's comments**
> > > > >
> > > > > **On related  work, entropy-based measures, etc.**
> > > > >
> > > > > There are several works that study, say, the mutual information between a mechanism’s input and output as a measure of privacy; however, in our setting these measures generally fail to distinguish between information gained as a result of: (1) correlation to the public features, and (2) the mechanism output. Thus, they overestimate the revelation of the mechanism, at times to the extreme. For clarity, consider the extreme situation where $\eta(x) \in {0,1}$ for all $x$. Then, the mutual information between $M(x,y)$ and $y$ fails to capture the fact that the adversary’s inference advantage is 0 for any reasonable measure of advantage in this setting, since the adversary has perfect knowledge of the labels without the mechanism output. Regarding the specific references the reviewer is pointing out, it is true that they have a similar goal as the one in this paper. Yet, those papers do not consider the public features/private label mixed setting we consider here (which is elucidated by the above simple example, and clearly necessary when investigating, for instance, the LLP mechanism), and so it is unclear how to incorporate the public information. This seems to be the case, e.g., for the two references Makhdoumi et al., and Xiao et al. the reviewer is providing. In any event, we will cite such papers in our related work section, thanks for bringing them up.
> > > > >
> > > > > **Privacy vs. Learning**
> > > > >
> > > > > The reviewer makes several comments and raises concerns related to training/testing data and learning formalisms that we believe are orthogonal to the main contributions of our paper, and perhaps indicate a misunderstanding of the goals of our paper. We want to emphasize that our proposed measures try to quantify the risk posed to individuals in a dataset as a result of the output of the PET, whether the released data is used for learning or not. We use machine learning to motivate the experimental section of our work, but our definitions and results make sense in other contexts. For example, if a human analyst uses the PET output to create a report or dashboard, our measures would still be meaningful in that context.
> > > > >
> > > > > In particular, we believe that several drawbacks from the reviewer’s further thoughts are out of scope for our paper:
> > > > >
> > > > > - “The evaluation of privacy happens on the same set as the training set.”
> > > > >
> > > > > Yes, our measures quantify the privacy risk to a dataset due to running a PET on that dataset. Because the data is i.i.d., the only way the PET output can help an adversary infer the labels of data not present in the PET input is through learning statistical correlations between the features and labels. We do not feel that this risk should be attributed to the PET, since those correlations could potentially be learned from other data sources, or the adversary might already know them. Note that Differential Privacy also does not protect against learning correlations in a population (See for example Section 1 of  [Dwork, Roth “The Algorithmic Foundations of Differential Privacy” ‘14] for a full discussion of this concept).
> > > > >
> > > > > - “The approach assumes adversaries have knowledge to the full distribution [...] This is different from [...] "fixed but unknown distribution" for the data.”
> > > > >
> > > > > We do not agree that the common assumptions for learning should also be adopted when modeling an adversary. As explained in LL. 158–163, we assume the adversary knows the distribution $D$ to capture the fact that they may be very well informed about statistical patterns in the data. Our measures capture how much risk can be attributed to the PET itself, beyond what could be inferred from the publicly available data by a well-informed adversary.
> > > > >
> > > > > - “... the paper considers an average case over the distribution D, which may have significant implications from the point of learning theory.”
> > > > >
> > > > > Even though our risk measures average over the distribution $D$, it is unclear to us why this is listed as a drawback, and what are the “significant implications from the point [of view?] of learning theory” the reviewer is alluding to. As we said in the paper, since we want to put on the same footing deterministic and randomized mechanisms, we have to follow an average case (or high probability) analysis over the data distribution. If the mechanism is randomized then we *do* also consider the average behavior over the randomized mechanism (unlike what the reviewer seems to claim) – please have a closer look, e.g., at the display after L. 154.
> > > > >
> > > > > [to continue...]

---

> > > > > > ### Author Response · Authors · 2024-08-13
> > > > > > **Response to Reviewer Us4Y's comments (cont'd)**
> > > > > >
> > > > > > **Further responses**
> > > > > >
> > > > > > - On the use of quotes.
> > > > > >
> > > > > > Apologies. We changed the order of the last two sentences and did some very light paraphrasing when writing "like entropy based" instead of "e.g., there is quite some work using entropy-based measures to ...".  Our paraphrasing came about when editing for length but it was not our intention to change the quotation.
> > > > > >
> > > > > > - ``In general, please consider that the list of technical details provided in my review is only a sample. Other sections seem to have a similar level of rigor and require being improved too.”
> > > > > >
> > > > > > This comment is rather awkward, since we are not put in a position to respond. If the reviewer has more technical issues to flag, then they are compelled to make these issues explicit.
> > > > > >
> > > > > >
> > > > > > - `i.i.d. data is required, but I guess that if the data is not i.i.d., in this setting this just means that the test data may be from a different distribution [...] valid worst case bounds? “
> > > > > >
> > > > > > As discussed above, the test/train distinction the reviewer is speculating about is irrelevant to our setting, since we are interested in measuring the information revealed about the points in the training dataset not the test dataset. We mainly use the i.i.d. assumption to reason about the attacker’s knowledge, specifically that one person’s label does not reveal any additional information about another person’s label.
> > > > > >
> > > > > > - ``The text says "public features together with one sensitive binary feature", but it is unclear why the proposed measures can't be applied in other cases.”
> > > > > >
> > > > > > We don’t fully understand which specific “other cases” the reviewer has in mind. Our measures can clearly be applied, e.g., to multiclass (that is, nonbinary) sensitive features.  Yet, it is not the point of the paper to widen the applicability of the proposed method to the largest possible extent.
> > > > > >
> > > > > > - On the statement of Thm 3.2.
> > > > > >
> > > > > > Thanks, we will consider the reviewer's suggestion here, like splitting the bound into two separate statements and/or giving a specific reference to the appendix.
> > > > > >
> > > > > > - On composability.
> > > > > >
> > > > > > LLP does not satisfy composability guarantees in the worst case (in particular if two computations use bags that differ on one person’s data). Thus, any measure that adequately captures LLP also will not satisfy composability guarantees. This is briefly mentioned in Sect. 5.
> > > > > >
> > > > > >
> > > > > > - ``it seems to be assumed the adversary also has access to the dataset. E.g., Line 164 first applies to the complete training set and then takes the i-th component.”
> > > > > >
> > > > > > No, the adversary only sees the feature parts of the training set. The reason why there is also a probability over the labels $y$ is because the measures also factor in the conditional distribution of $y$ given $x$.
> > > > > >
> > > > > >
> > > > > > -  ``It doesn't allow to run on some instance not in the training set. This seems to limit the applicability of the proposal.”
> > > > > >
> > > > > > As described above, our goal is to measure the risk to a given dataset by publishing the output of the PET run on that data, not to prevent statistical inferences on future data not processed by the PET.

---

> > > > > > ### Comment · Reviewer_Us4Y · 2024-08-14
> > > > > >
> > > > > > > There are several works that study, say, the mutual information between a mechanism’s input and output as a measure of privacy; however, in our setting these measures generally fail to distinguish between information gained as a result of: (1) correlation to the public features, and (2) the mechanism output.
> > > > > >
> > > > > > Such work indeed may exist.  Still, the natural approach is to compare (a) the prior knowledge without the output of the mechanism and (b) the posterior knowledge after seeing the output of the mechanism.  Comparing these two can distinguish between correlation with public features (usually called prior knowledge) and mechanism output (on top of what is expected given the input and public features).
> > > > > >
> > > > > > > Privacy vs. Learning
> > > > > >
> > > > > > It is true that the authors, me, and in fact also some existing literature sometimes use the word "learning" when they mean "mechanism".  I trust in almost all cases this is for reasons of illustration, i.e., a privacy-preserving machine learning algorithm is an important example of a mechanism, and reasoning about the properties of learning algorithms can help us understand the desirable properties for a mechanism.  For the purpose of the current discussion, please be assured that in all these cases I (and I trust also the authors) mean these statements more generally and also applying to mechanisms in general (minus the learning-specific details).
> > > > > >
> > > > > > I insist that none of the following is a PET (at least not the overwhelming majority of papers in the literature):
> > > > > > * a mechanism
> > > > > > * a learning algorithm
> > > > > > * a DP algorithm for probabilistic inference in a Bayesian network
> > > > > > * a DP algorithm for solving a constraint optimization problem
> > > > > >
> > > > > > We could use the word PET in the following phrases:
> > > > > > * The PET used in a DP learning algorithm is "differential privacy"
> > > > > > * The PET used in a DP constraint optimization algorithm is "differential privacy"
> > > > > > * The PET used in an algorithm that performs stochastic gradient descent under homomorphic encryption is "homomorphic encryption"
> > > > > > * The PET used in "an algorithm sending a result encrypted using a one-time pad" is "encryption".
> > > > > >
> > > > > > Here, encryption, hoimomorphic encryption, differential privacy, ... are (privacy enhancing) technologies (PETs), not mechanisms.
> > > > > > Please also not that
> > > > > > * "an algorithm sending a result encrypted using a one-time pad" is not a mechanism, as it is not randomized (given the pad).
> > > > > >
> > > > > > > scope of the paper
> > > > > >
> > > > > > The authors are free to choose the scope of the paper, the reviewers are free to have an opinion what is needed to make a work complete.
> > > > > >
> > > > > > >. the paper considers an average case over the distribution D, which may have significant implications f
> > > > > >
> > > > > > (I omitted here my original "learning theory" as these words made the sentence overly specific)
> > > > > >
> > > > > > We should distinguish two cases:
> > > > > >
> > > > > > (1) The common case (e.g. $(\epsilon,\delta)$-DP) : Whatever is the distribution and the dataset drawn from it, with high probability (over the randomization of the mechanism) the adversary will not be able to infer significant information from the output (over his prior knowledge).
> > > > > >
> > > > > > (2) The case in the current submission:  Whatever is the distribution, with high probability over the dataset drawn from it and the randomization of the mechanism, the adversary will not be able to infer significant information from the output (over his prior knowledge).
> > > > > >
> > > > > > In the latter case, it is possible that we are unlucky and the specific dataset drawn from the known distribution does not provide any privacy when the mechanism is applied.  (This is possible as the offered guarantee only applied on average over all datasets drawn)
> > > > > >
> > > > > > In real-world applications, the current dataset is the only dataset we care about, we don't care about the "average dataset".
> > > > > > The risk that our current dataset is worse than the average case exists, but it is sometimes hard to evaluate how much worse the situation is for the current dataset than for the average dataset.

---

### Official Review · Reviewer_kz5F · 2024-07-13

**Soundness:** 4
**Presentation:** 4
**Contribution:** 3
**Rating:** 9
**Confidence:** 4

**Summary:**

The authors present new auditing tools for privacy mechanisms (differentially private or otherwise) with respect to the threat of label inference. They define measures to measure how much an adversary's posterior belief differs from their prior belief after viewing a dataset processed by a PET method, provide bounds on these measures for different methods theoretically, and provide an empirical assessment of privacy risks over multiple datasets. All of this is accompanied by an in-depth discussion about each definition, bound, and result, along with reflections on what this metric means for privacy and related considerations.

**Strengths:**

* Very well-motivated work, and the motivation of why label inference is a good and more realistic threat to look at vis-a-vis membership inference, etc.
* Very well done literature review that is at once deep and nearly exhaustive and friendly towards newcomers to this area of work.
* Very rigorously defined quantities and notions of interest along with high-level discussions about what those quantities mean.
* The ability to measure privacy guarantees for DP and non-DP methods alike is very welcome and is a significant step in a line of similar work (such as [1] and [2]; please note that I am *not* asking the authors to cite this, but I seek to add further evidence about why this paper is of interest to the community and to argue for its acceptance).
* Well-designed empirical experiments with relevant quantities and plots that capture privacy leakage well. The empirical results are discussed in detail and very carefully.
* Exhaustive theoretical discussion with bounds with respect to the two auditing measures, additive and multiplicative, for different methods (DP and otherwise).

[1] Das, S., Zhu, K., Task, C., Van Hentenryck, P., & Fioretto, F. (2024). Finding ε and δ of Traditional Disclosure Control Systems. Proceedings of the AAAI Conference on Artificial Intelligence, 38(20), 22013-22020. https://doi.org/10.1609/aaai.v38i20.30204
[2] M. Christ, S. Radway and S. M. Bellovin, "Differential Privacy and Swapping: Examining De-Identification’s Impact on Minority Representation and Privacy Preservation in the U.S. Census," 2022 IEEE Symposium on Security and Privacy (SP), San Francisco, CA, USA, 2022, pp. 457-472, doi: 10.1109/SP46214.2022.9833668. keywords: {Couplings;Privacy;Differential privacy;Sociology;Security;Stress;Differential-privacy;census;data-swapping},

**Weaknesses:**

No clear weaknesses. The theory and experiments are solid, every proof is provided, the motivation is well done, and the discussion has great depth and clarity. This is among those rare papers that I cannot point to something to criticise (at least not right away).

The only weakness, to be pedantic, is the lack of code to reproduce the experiments, which I would appreciate the provision of by the authors.

**Questions:**

* Could the authors please provide the code for reproducing the empirical results?

**Limitations:**

Yes, the authors do discuss that and speculate on future directions of work to address them.

---

> ### Author Rebuttal · Authors · 2024-08-06
>
> * Can you publish/provide the code?
>
> We will publish the code, subject to internal organizational approval.

---

### Official Review · Reviewer_zE7q · 2024-07-22

**Soundness:** 3
**Presentation:** 3
**Contribution:** 3
**Rating:** 7
**Confidence:** 4

**Summary:**

The paper introduces new metrics to quantify the amount of label leakage associated with a generic privacy mechanism. It then analyzes two common label privacy mechanisms and quantifies the measures for these two mechanisms. Especially for the mechanism that also has the guarantee of differential privacy, the paper shows that the upper bound of the measures coming from differential privacy can be loose by comparing it to the results analyzed in the paper. In the experiment, the paper empirically evaluates the algorithms with both utility and the proposed privacy measurement.

**Strengths:**

1. The paper is very well-written.
2. The theoretical results of analyzing two label-privacy mechanisms are novel and useful.
    - For the DP mechanism, the paper shows a tighter analysis in terms of the proposed measure.
    - Unlike the results in the literature that work only for the differential privacy mechanism, the paper also shows the theoretical results for the non-DP mechanism.

**Weaknesses:**

As the measure is defined as the maximum advantage of all label inference attack $\mathcal{A}$ and the paper shows how to calculate the measure exactly for two mechanisms, it would be great to see how the existing label inference attack is bounded. This is because the measure is generally hard to compute, e.g. for the mechanisms that are not analyzed in the paper. If an existing label inference attack is shown to be tight to the exact upper bound, this attack can be used as an empirical indicator of the measure too.

**Questions:**

Please see the weaknesses.

**Limitations:**

The paper has well discussed the limitations

---

> ### Author Rebuttal · Authors · 2024-08-06
>
> * As the measure is defined as the maximum advantage of all label inference attack and the paper shows how to calculate the measure exactly for two mechanisms, it would be great to see how the existing label inference attack is bounded. This is because the measure is generally hard to compute, e.g. for the mechanisms that are not analyzed in the paper. If an existing label inference attack is shown to be tight to the exact upper bound, this attack can be used as an empirical indicator of the measure too.
>
> Indeed, if there was an attack that achieved the theoretical optimum, then it could potentially be used to empirically compute the label inference measures on PETs for which this is harder to compute analytically. In practice, this is complicated by the fact that the optimal attack depends on both the data distribution and the PET itself, so an attack that is optimal in one setting may not generalize to other PETs and distributions. It’s unclear how to reconcile this, but at the very least the attack could provide a lower bound on label inference success. In our experiments (for both the informed and uninformed adversary) we either analytically computed the Bayes optimal attack (on the synthetic datasets) or an approximation to the Bayes optimal attack (on the real-world datasets) by training an ML model on non-private data. This approximation to the Bayes optimal attack may turn out to be reasonably tight if we have enough training data.

---

> > ### Comment · Reviewer_zE7q · 2024-08-13
> > **Official Comment by Reviewer zE7q**
> >
> > Thank authors for their responses and they partially addressed my question. I decided to keep my positive score.

---

### Official Review · Reviewer_enWp · 2024-07-26

**Soundness:** 4
**Presentation:** 3
**Contribution:** 4
**Rating:** 8
**Confidence:** 4

**Summary:**

This paper proposes two reconstruction advantage metrics to audit label privatization mechanisms. Unlike differentially private (DP) auditing techniques, which focus on worst-case guarantees, the authors of this work focus on distributional guarantees. Concretely, the authors assume the adversary has knowledge of the data distribution $\mathcal{D}$ over $\mathcal{X} \times \mathcal{Y}$ , either completely or approximately through learning on data. This allows the adversary the leverage correlations in the data, which are unknown to the mechanism, to augment their attack. Consequently, the attack advantage metrics proposed in this work measure the label reidentification risk that can be attributed to the mechanism rather than to correlations between the features and labels inherent to the distribution $\mathcal{D}$. Moreover, these metrics apply to both DP mechanisms and deterministic mechanisms commonly used in practice.

The first metric investigated in the paper is the additive attack advantage, which was introduced in the context of label inference attacks by Wu et al. At a high level, this metric measures the difference between the expected accuracy of two Bayes optimal adversaries: one that can see the output of the mechanism and one that cannot. Notably, the expectation is taken over both the data distribution $\mathcal{D}$ and coin flips in the mechanism. Authors derive various analytical bounds that track how this measure changes under different correlation assumptions and mechanisms (Theorem 3.3, 3.4, 3.5), and demonstrate the privacy-utility trade-off using this metric for various DP mechanisms and a non-DP mechanism (Figure 3).

Since the first metric measures average-case risk, it does not capture unlikely high-disclosure events. The second metric, the multiplicative attack advantage, is introduced to remedy this shortcoming. At a high level, this metric measures the difference of log odds ratio between the adversary's prior and posterior probability of observing two different labels. This metric can be written as $\\log \\frac{\text{Pr}(\mathcal{M}(\mathbf{x}, \mathbf{y} ) = z \mid y_i = 1, \mathbf{x})}{\text{Pr}(\mathcal{M}(\mathbf{x}, \mathbf{y} ) = z \mid y_i = 0, \mathbf{x})}$, and is therefore analogous to the ''log likelihood ratio'' that is commonly characterized in the DP literature. In contrast to the DP literature, the probability in the likelihood ratio here is taken over both the data distribution $\mathcal{D}$ and coin flips in the mechanism. Authors provide a high probability bound for this second metric for a often utilized deterministic mechanism (Theorem 3.7), and demonstrate the privacy-utility trade-off using this metric for various DP mechanisms and a non-DP mechanism (Figure 3). Further notions of attack advantage are explored in Appendix A.5.

An interesting result from this work is the empirical observation that learning with deterministic aggregate labels is harder than learning from randomly perturbed labels. In other words, randomized response had the best privacy-utility trade-off out of all the investigated mechanisms.

**Strengths:**

This paper contains many contributions to the field of auditing DP and non-DP mechanisms via label inference attacks. It builds off of the previously proposed metric of additive attack advantage by:

(1) extending it to non-DP mechanisms

(2) deriving novel bounds of this metric on a non-DP mechanism often used in practice (Theorem 3.2, 3.3, 3.4) and

(3) using this metric to empirically compare the privacy-utility trade-off of DP and non-DP mechanisms on equal footing.

Moreover,  inspired by previous work in the DP community, this paper proposes a metric called multiplicative attack advantage, and repeats the same contributions listed above with this metric. The authors provide experimental evidence that learning with deterministic aggregate labels is harder than learning from randomly perturbed labels. This experimental result is, as far as I am aware, novel. Given these contributions, I recommend this paper for acceptance.

**Weaknesses:**

My main critique of this paper is that, given the prominence that the hybrid LLP+Geom mechanism plays in the experimental results, the paper spends comparatively little time explaining how either the proposed metrics or the posterior probability for this mechanism are computed.

In the Questions section, I go into more detail and ask specific questions regarding this critique, and also give some suggestions to improve organization and readability of the paper.

**Questions:**

$\textbf{Questions:}$

1. Figure 1 plots the prior and posterior distribution for LLP, RR and LLP+Geom. While I have a guess for how these posterior probabilities were calculated (e.g. using the ''likelihood'' $\text{Pr}(\mathcal{M}(\mathbf{x}, \mathbf{y} ) = z \mid y_i = 1, \mathbf{x})$ and the prior $\text{Pr}(y_i = 1 \mid \mathbf{x})$), the paper is not clear how the posterior probabilities were calculated, particularly for LLP+Geom and LLP+Lap. Could the authors please elaborate on how this was done?

$\textbf{Suggestions:}$

1. Given how important the spindles are for the discussion in the experimental section, the remark "Note that a spindle boundary is the set of points having the same multiplicative advantage measure" on line 350 should be elaborated upon. I understand what the authors are trying to say, i.e. that the spindles in Figure 1 trace the contours of $\log \frac{y}{1-y} - \log \frac{x}{1-x} = \pm C$, but this was not obvious upon a first reading.

2. In Theorem 3.2, it was as initially unclear to me why the random variable $\alpha$ appeared without a subscript denoting the ball it belongs to. I think would benefit the reader to remind them that by the iid assumption, the distribution of $\alpha$ over each ball is the same, hence the subscript can be dropped wlog.

$\textbf{Typos and Related Nitpicks}$

1. Line 224 contains what I believe is a typo. It currently reads: "In the bounds of Theorems 3.2, 3.3 and 3.3”, whereas it should read "In the bounds of Theorems 3.2, 3.3 and 3.4”.

2. In the following line 225, the statement "as $p$ and/or $\mu$ gets smaller we should naturally expect a smaller advantage" is misleading because of the previous theorems' dependence on $p$ and $1-p$. A more precise statement would be "as $p$ gets further from 1/2".

3. There is currently an inconsistent use of $\epsilon$ and $\varepsilon$ in the paper. For example, in Definition 2.1 both $\epsilon$ and $\varepsilon$ are used in the same sentence to refer to the privacy parameter.

4. In Definition 3.6, the arguments of $I\_{a,b}$ are $I_{a,b}(\mathcal{M}, \mathcal{D}, \mathbf{x}, z, i)$, but in the following paragraph the arguments $I_{a,b}(\mathcal{M}, \mathcal{D}, \mathbf{x}, z, x_i)$ are used.

5. Lines 266 - 267: “What values of MA hould be considered acceptable? We argue that this probability should be viewed as a probability of system failure and set appropriately small” contains a typo with "hould" and an acronym "MA" that is not defined or (as far as I can tell) mentioned again.

**Limitations:**

Authors adequately address the limitations of their work.

---

> ### Author Rebuttal · Authors · 2024-08-06
>
> * Explain how to compute proposed metrics or posterior distributions.
>
> We will include more details on how we compute the posterior distributions for each mechanism. As suggested, we do apply Bayes’ theorem to compute the posterior:
> $$
> Pr(y_i = 1 |  x, \mathcal{M}(x,y) = z) = \frac{Pr(\mathcal{M}(x,y) = z | x, y_i=1) \cdot Pr(y_i=1 | x )}{Pr(\mathcal{M}(x,y) = z | x)} = \frac{Pr(\mathcal{M}(x,y) = z | x, y_i=1) \cdot Pr(y_i=1 | x_i )}{Pr(\mathcal{M}(x,y) = z | x)} .
> $$
> Now, because in the above $x$ is fixed, this ratio is simply a function of the $\eta(\cdot)$’s, as well as any randomness in $\mathcal{M}$.
> For Randomized Response (RR) with flipping probability $p$ this ratio is $\frac{p\eta(x_i)}{p\eta(x_i) - (1-p)(1-\eta(x_i)}$ if the outcome of $RR(y_i)  = 0$, and $\frac{(1-p)\eta(x_i)}{(1-p)\eta(x_i) - p(1-\eta(x_i)}$ if the outcome is 1.
>
> For label aggregation (LLP), $\mathcal{M}$ is deterministic, so in the numerator $Pr(\mathcal{M}(x,y) = z | x, y_i=1)$ is the probability density at $z$ of a Poisson binomial with flipping probabilities $(\eta(x_1),\ldots, \eta(x_{i-1}), 1, \eta(x_{i+1}), \ldots, \eta(x_k))$. Similarly, in the denominator $Pr(\mathcal{M}(x,y) = z | x)$ is the probability density at $z$ of a Poisson binomial with flipping probabilities $(\eta(x_1), \ldots, \eta(x_k))$.
>
> For LLP + Geom and LLP + Lap, the terms in the ratio are *convolutions* of the posterior for LLP and the Geometric or Laplace distribution, respectively.
>
> Some of these posterior calculations can be found in the appendix, for example, in Line 561 for bags, and in Line 720, Eq (15) and (16), for Randomized Response.
>
> - Other suggestions/typos.
>
> We will implement the suggestions and fix the typos – thanks !

---

> ### Comment · Reviewer_enWp · 2024-08-07
>
> I read through the authors' responses to my own comments and to the other reviewers. In response, I have increased my score to strong accept (8).
>
> My one comment is that, if computing the posterior probabilities in Figures 1/4 for the LLP-Geom/Laplace mechanism involves convolving (either numerically or analytically) a Geometric/Laplace random variable with a PoissonBinomial to obtain the likelihood $\text{Pr}(\mathcal{M}(\mathbf{x},\mathbf{y}) = z \mid y_i=1, \mathbf{x})$, that this convolution step should be explicitly mentioned somewhere in the Appendix.

---

> > ### Author Response · Authors · 2024-08-07
> > **Response to Reviewer enWp's comment**
> >
> > Yes, we agree with that suggestion. We generally did the convolution numerically, and will explain the calculation. We also plan to post the exact code -- Thanks !

---

### Decision · Program_Chairs · 2024-09-25

**Decision:**

Accept (spotlight)

**Comment:**

The submission introduces two auditing metrics (additive and multiplicative attack advantages) to assess the vulnerability of label privatization mechanisms. The authors effectively demonstrate the application of these metrics to (empirically) auditing both differentially private and non-differentially private privacy mechanisms. Most reviewers praised the work, noting its potential for impact in the growing field of privacy auditing. In particular, during the discussion, Reviewer mygV highlighted the paper’s practical contributions and pragmatic approach, as well as its analysis of privacy-utility trade-offs across various mechanisms. Reviewer kz5F also appreciated the rigorous theoretical analysis and empirical validation. Reviewer enWp also strongly supported acceptance, emphasizing the paper's substantial methodological contributions to auditing privacy mechanisms and noting the novelty of the experimental results.

Given the reviewers' comments and my own read of the paper, this submission is a clear acceptance. While Reviewer Us4Y raised concerns regarding the paper's rigor and its placement relative to the broader privacy literature -- engaging in an extensive back-and-forth with the authors -- these concerns were not shared by other reviewers and the AC. I encourage the authors to incorporate the suggestions provided by the reviewers and promised in their rebuttal to the final paper, including releasing their code, providing more details on how to compute posterior distributions (Reviewer enWp), fixing typos, and clarifying ambiguities in some of the mathematical results (Reviewer Us4Y).